# Palaeoecology of ungulates in northern Iberia during the Late Pleistocene through isotopic analysis of teeth

**Mónica Fernández-García[1,2,3] (*), Sarah Pederzani[4], Kate Britton[5], Lucía Agudo-Pérez[1], Andrea Cicero[1], Jeanne Marie Geiling[1], Joan Daura[6], Montserrat Sanz[6], Ana B. Marín-Arroyo[1] (*)**

Grupo de I+D+i EVOADAPTA (Evolución Humana y Adaptaciones durante la Prehistoria), Departamento de Ciencias Históricas, Universidad de Cantabria, 44. 39005 Santander, Spain
Departament de Prehistòria, Arqueologia i Història Antiga, Universitat de València, Av. Blasco Ibañez 28, 46010 Valencia, Spain.
Institut Català de Paleoecologia Humana i Evolució Social (IPHES-CERCA), zona Educacional 4 Edifici W3, Campus Sescelades URV, 43007 Tarragona, Spain.
 Spatio-Temporal Isotope Analytics Lab. Department of Geology & Geophysics. University of Utah.5 Department of Archaeology, University of Aberdeen, Aberdeen AB24 3UF, United Kingdom
Grup de Recerca del Quaternari (GRQ-SERP), Department of History and Archaeology, Universitat de Barcelona, C/Montalegre 6-8, 08001 Barcelona, Spain.

(*) Corresponding authors: anabelen.marin@unican.es, monica.fegar@gmail.com

## Abstract

During the Late Pleistocene, stadial and interstadial fluctuations affected vegetation, fauna, and human groups that were forced to cope with these pronounced spatial-temporal climatic and environmental changes. These changes were especially abrupt during the Marine Isotopic Stage (MIS) 3. Here, we reconstruct the climatic trends in northern Iberia considering the stable isotopic composition of ungulate skeletal tissues found in archaeological deposits dated between 80 to 15 ka cal BP. The carbon and oxygen isotopic composition preserved in the carbonate fraction of tooth enamel provides a reliable and high-resolution proxy of the food and water consumed by these animals, which is indirectly related to the local vegetation, environment, and climate, allowing us to estimate paleotemperatures and rainfall intensity. This study presents new isotope data from 44 bovine, equid, and cervid teeth from five archaeological sites in the Vasco-Cantabrian region (El Castillo, Axlor, Labeko Koba, Aitzbitarte III interior and El Otero,) and one in northeastern Iberia (Canyars), where human evidence is attested from the Mousterian to the Magdalenian. The carbon isotope values reflect animals feeding on diverse C3 plants in open environments, and point to differentiated ecological niches for equids and bovines, especially during the Aurignacian in the Vasco-Cantabrian region. Temperature estimations based on oxygen isotopic compositions and rainfall obtained from carbon isotopic compositions indicate colder and more arid conditions than nowadays for the human occupations from the Late Mousterian to the Aurignacian. The contemporary northeastern Iberia site shows slightly lower temperatures related to an arid period when animals mainly graze in open landscapes. In the Vasco-Cantabrian region, during the MIS2, the Gravettian data reflect a landscape opening, whereas the Magdalenian points to warmer (but still arid) conditions.

**Keywords**: Middle and Upper Palaeolithic; Neanderthal; Homo sapiens, palaeoecology; geochemistry

## 1. Introduction

Understanding local and regional climatic variability during the Late Pleistocene in southern Europe is crucial for assessing the potential impact of climate on the adaptation and decline of Neanderthals and the subsequent expansion and resilience of Anatomically Modern Humans during the Upper Paleolithic (e.g., D'Errico and Sánchez Goñi, 2003; Finlayson and Carrión, 2007; Sepulchre et al., 2007; Staubwasser et al.,

2018). During the Late Pleistocene, the climatic records demonstrate stadial and interstadial continuous
fluctuations during the Marine Isotope Stage 3 (MIS 3, ca. 60-27 ka) and MIS 2 (ca. 27-11 ka). Human
groups had to face those episodes, which affected vegetation and fauna to different extents, depending on
the region. Northern Iberia is a key study area due to the abundance of well-preserved archaeological caves
and rock shelters where, in the last decade, an updated and multidisciplinary approach has been applied to
disentangle how changing environmental conditions affected the subsistence dynamics of Middle and Upper
Paleolithic hominins. Recent chronological, technological, subsistence studies and ecological
reconstructions are revealing a  more complex regional panorama than previously known (e.g., Sánchez
Goñi, 2020; Vidal-Cordasco et al., 2022; 2023; Timmermann, 2020; Klein et al., 2023).
The Vasco-Cantabrian region, located in northwestern Iberia, is subject to the influence of Atlantic climatic
conditions, where recently has been evaluated the impact of the glacial-interglacial oscillations during MIS3
(Vidal-Cordasco et al., 2022). Modelling of traditional environmental proxies (small vertebrates and pollen)
associated to archaeo-paleontological deposits show a progressive shift in the climatic conditions with
decreasing temperatures and rainfall levels detected during the late Mousterian (Fernández-García et al.,
2023). Ecological alterations have been observed in large mammals, such as niche partitioning between
horses and cervids (Jones et al., 2018), a decrease in the available biomass for secondary consumers, and
consequently, a reduction in the ungulate carrying capacity ((Jones et al., 2018; Vidal-Cordasco et al., 2022).
Cold and arid conditions are observed during the Aurignacian and the Gravettian until the onset of MIS2.
Afterwards, during the Last Glacial Maximum (LGM, 23-19 ka), the global climatic deterioration associated
with this glacial phase results in colder and more arid conditions in the region, with a predominance of open
landscapes. However, this region still provided resources for human survival acting as a refugia with more
humid conditions in comparison to the Mediterranean area (Cascalheira et al., 2021; Fagoaga, 2014;
Fernández-García et al., 2023; Garcia-Ibaibarriaga et al., 2019a; Lécuyer et al., 2021; Posth et al., 2023).
By the end of the LGM, a climate amelioration and a moderate expansion of the deciduous forest are
documented from the late Solutrean through the Magdalenian (Garcia-Ibaibarriaga et al., 2019a; Jones et
al., 2021).
In contrast, northeastern Iberia is influenced by the Mediterranean climate. The MIS 3 human settlement in
this region have been linked to cooler temperatures and with higher rainfall, compared to the present, but
with climatic fluctuations less pronounced compared to the Vasco-Cantabrian region (López-García et al.,
2014; Fernández-García et al., 2020; Vidal-Cordasco et al., 2022). Archaeobotanical and small vertebrate
evidence indicate relatively stable climatic conditions, but also suggest the persistence of open forests
during the Middle to Upper Paleolithic transition, as found in northwestern Iberia (Allué et al., 2018; Ochando
et al., 2021). However, certain archaeological records indicate specific climatic episodes, such as increased
aridity and landscape opening during Heinrich Stadials 4 and 5 (e.g., Álvarez-Lao et al., 2017; Daura et al.,
2013; López-García et al., 2022; Rufí et al., 2018).
These multi-proxy studies have significantly expanded our understanding of the environmental evolution in
Iberia, alongside proxies derived from marine core records in Iberia margins (Fourcade et al., 2022; Martrat
et al., 2004; Naughton et al., 2007; Roucoux et al., 2001; Sánchez-Goñi et al., 1999, 2009) and other regional
paleoclimatic records sourced from local natural deposits (e.g., Pérez-Mejías et al., 2019; Moreno et al.,
2010, 2012; González-Sampériz et al., 2020; Ballesteros et al., 2020). However, the availability of proxies
enabling the direct connections between these environmental shifts and human activities remains limited.
In this study, we investigate the palaeoecological and palaeoenvironmental dynamics in northern Iberia
during the late Middle and Upper Paleolithic by measuring the carbon and oxygen isotopic composition of
bioapatite carbonates ($\delta^{13}C_{carb}/\delta^{18}O_{carb}$) preserved in archaeological mammal teeth. These analyses provide
high-resolution snapshots of ecological information from animals accumulated during human occupations at
the caves. Tooth enamel forms incrementally and does not biologically remodel (Kohn, 2004; Passey and
Cerling, 2002), in contrast to other bodily tissues such as bone, which implies that the isotope values
measured on them reflect the animal diet and water sources consumed during its mineralisation, around
one to two years of life for the species included in our study (bovids, equids, cervids)(e.g., Hoppe et al.,
2004; Pederzani and Britton, 2019; Ambrose and Norr, 1993; Luz et al., 1984). The preserved carbon
isotope composition relies on animal dietary choices reflecting mainly the type of plant consumed (C3/C4),
exposition to light and humidity levels. Otherwise, the oxygen isotope composition reflects mainly the
environmental water consumed by animals, directly by drinking or through diet, which reflects isotopic
information derived from water sources as well as changes in climatic conditions. Both indirectly provide
information on the vegetation and climate that allows estimating past temperatures, rainfall, and moisture
on a sub-annual scale, returning isotopic data of the foraging areas where animals were feeding during teeth
formation.
By analysing the stable isotopic composition of 44 ungulate teeth obtained from 15 archaeological levels
directly associated with human occupation, including El Castillo, Axlor, Labeko Koba, Aitzbitarte III interior
and El Otero in northwestern Iberia, and Terrasses de la Riera dels Canyars in northeastern Iberia, this
study presents novel insights into local and regional environmental and climatic trends associated to human
presence during the Late Pleistocene (Fig.1; Fig.2; Appendix A). Specifically, it focuses on the Middle to
Upper Paleolithic transition in both areas and the post-LGM period in the Vasco-Cantabrian region.
The main objectives of this work are: 1) to assess how regional environmental conditions, including changes
in moisture and vegetation cover, but also temperatures and rainfall, are recorded in the stable isotopic
composition of tooth enamel; 2) to characterize animal diet and their ecological niches; 3) to obtain
quantitative temperature data to compare with available proxies; 4) to characterise seasonal patterns of
animals found in the archaeological sites by identifying winter and summer fluctuations.

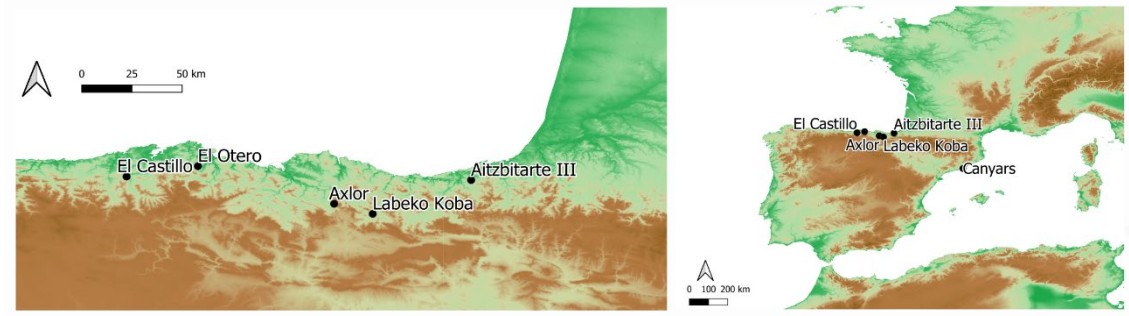

**Figure 1.** Location of the archaeological sites included in this study. From west to east, in the autonomous community of
Cantabria, El Castillo, and El Otero; in the Basque Country, Axlor and Aitzbitarte III interior; in Catalonia, Canyars.

## 2. Archaeological sites and sampled material

This study selected a total of 44 ungulate teeth including 25 bovines (*Bos primigenius, Bison priscus,*
*Bos/Bison* sp.), 14 equids (*Equus* sp. and *Equus ferus*), and five cervids (*Cervus elaphus*) originating from
five archaeological sites in the Vasco-Cantabrian region (El Castillo, El Otero, Axlor, Labeko Koba,
Aitzbitarte III interior) and one in the northeastern area (Terrasses de la Riera dels Canyars, henceforth
Canyars). These teeth were recovered from 15 archaeological levels attributed to the following
technocomplexes: Mousterian (n=14), Transitional Aurignacian (n=10), Châtelperronian (n=2), Aurignacian
(n=12), Gravettian (n=1) and Magdalenian (n=5) (Table 1 and 2). Archaeozoological studies of the
archaeological sites are available (synthesis in Marín-Arroyo and Sanz-Royo, 2022; Daura et al., 2013) and
most prove that faunal remains were accumulated by human acquisition during the different cultural phases.
The isotopic results of equids teeth and other ungulates bone collagen from El Castillo were previously
published by Jones et al. (2019) in combination with the stable isotopes of ungulates from the site, as well
as the combined bioapatite carbonate and phosphate analyses of bovines from Axlor (Pederzani et al.,
2023). A comprehensive description of each archaeological site is provided in Appendix A.

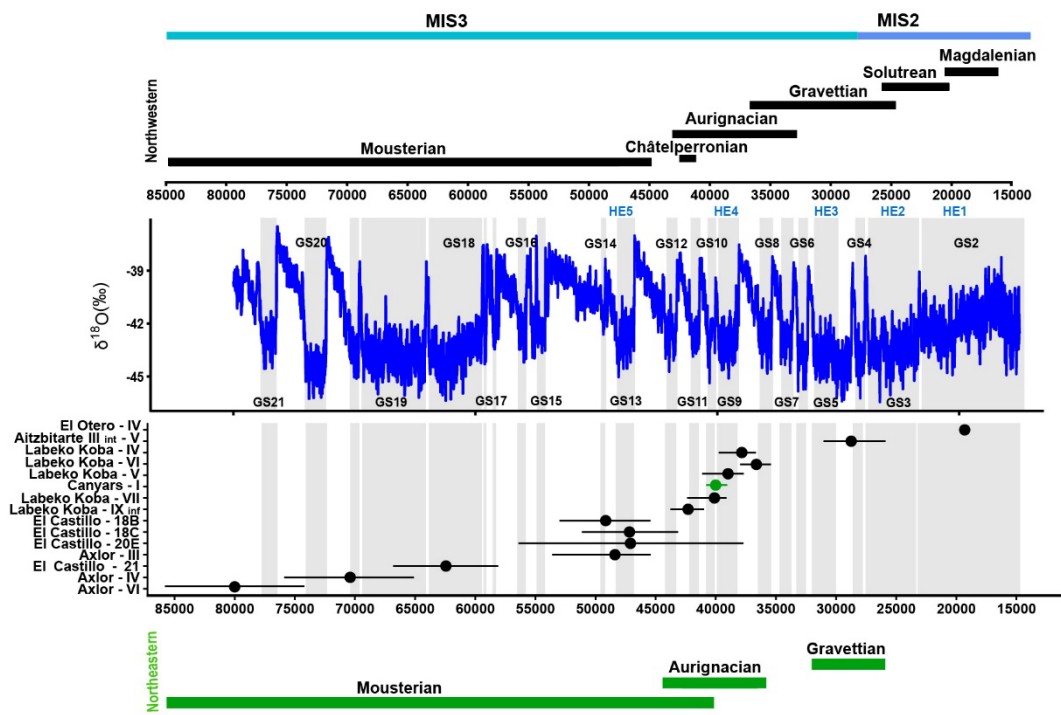


**Figure 2.** Representation of the duration each archaeological level (dots represent the median values, bars represent 95%
confidence intervals for 14C dates and 68% for ESR and OSL dates) related to techno-complexes in both northwestern (in black)
and northeastern Iberia (in green) and the δ18O record from the NGRIP (North Greenland Ice Core Project members, 2004;
Rasmussen et al., 2014). Grey bands indicate Greenland Stadials (GS). Dates from EL Castillo (C14 UF, ESR), El Otero (C14
UF), Axlor (C14 UF, OSL), Labeko Koba (C14 UF), Aitzbitarte III-interior (C14 AMS) and Canyars (C14 UF, ABA, ABOx-SC) are
shown in Appendix B and C.

**3. Methods**
**3.1 Methods: Dating methods**
Individual Bayesian age models were built for Canyars, El Castillo, Labeko Koba and Aitzbitarte III interior
based on radiocarbon dates (AMS UF and non-UF, ABOx-SC and ABA pretreatments on bones and
charcoal remains) using OxCal4.4 software (Ramsey, 2009), considering the INTCAL20 calibration curve
(Reimer et al., 2020) (Appendix C). The Bayesian model enables the modification of the calibrated
Probability Distribution Function (PDF) of individual dates based on the existing relative stratigraphic and
other relative age information. A resolution of 20 years was assumed, being a reasonable balance between
required accuracy and computational costs. An order function in the OxCal was used to calculate the
probability that one PDF predated another, providing information to assess synchronicity and temporal
overlap of individual archaeological levels and cultural phases in each of the four separate sites modelled.
Dates were organised into a 'Sequence,' and chronological information for each level was grouped into a
single 'Phase' with start and end 'boundaries' to bracket each archaeological level. The interval between the
start of each level and its end provided the duration of each level. In all cases, convergence was greater
than 95%. CQL codes, individual Bayesian models and modelled dates per site are reported in Appendix C.
No chronological models were built for El Otero because only a single date was obtained for level IV and El
Castillo levels 20E and 21 (ESR dated) and Axlor levels III, IV and VI (OSL dated) because dates go beyond
the limit of the radiocarbon. To show the duration of these levels in combination with the other sites and
levels, each of these dates was estimated by adding and subtracting the sigma (68% Confidence Interval)
from the uncalibrated date. In this way, we estimated the duration of these levels to be beyond 55 ka cal
BP.

**3.2 Tooth sampling**

All teeth included were sequentially sampled to reconstruct the complete $\delta^{18}O_{carb}$ and $\delta^{13}C_{carb}$ intratooth
profiles based on enamel carbonate bioapatite. Intratooth sequential sampling was applied to the second
and third molars and third and fourth premolars. Bovine and horse teeth sampled exceeded 3-4 cm of crown
height to ensure that at least a one-year isotopic record of animal life was obtained (Britton et al., 2019;
Hoppe et al., 2004). Samples were taken perpendicular to the growth axis on the tooth where the enamel
was best preserved, avoiding, whenever possible taphonomic alterations such as cracks or postdepositional
damages. Samples were performed in the buccal face for the lower teeth and the lingual part for the upper
ones. The outermost enamel surface was abraded to remove the superficial enamel, calculus, cementum,
or concretions adhering to the surface to avoid contaminations. The sequential sampling consisted of
straight strips (ca. 8 x 1.5 x 1 mm) covering the width of the selected lobe, approximately every 2-3 mm,
from the crown to the Enamel-Root-Junction (ERJ). The sample depth covered around 75% of the enamel
depth, and dentine inclusion was avoided. A low-revolution variable-speed manual drill was used, equipped
with 1 mm diamond-coated drill bits of conical and cylindrical shape. About 10-15mg of enamel powder was
collected in each subsample, generating 693 subsamples for IRMS measurements (see complete intratooth
profiles in Appendix D).

| Site | Level - Cultural period | Bovines | Horses | Red deer | Teeth | Subsamples |
|---|---|---|---|---|---|---|
| Axlor | VI - Mousterian | 2 | | | 2 | 32 |
| | IV - Mousterian | 1 | | | 1 | 12 |
| | III - Mousterian | 4 | | | 4 | 62 |
| El Castillo | 21A - Mousterian | 2 | 1 | | 3 | 47 |
| | 20E - Mousterian | 2 | 2 | | 4 | 56 |
| | 18C - Trans. Aurignacian | 4 | | | 4 | 66 |
| | 18B - Trans. Aurignacian | 3 | 2 | 1 | 6 | 93 |
| Labeko Koba | IX inf - Châtelperronian | | 1 | 1 | 2 | 24 |
| | VII - ProtoAurignacian | 3 | | | 3 | 68 |
| | VI - Aurignacian | | 1 | | 1 | 16 |
| | V - Aurignacian | 1 | 1 | | 2 | 39 |
| | IV - Aurignacian | | 1 | | 1 | 16 |
| Canyars | I - Aurignacian | 2 | 3 | | 5 | 76 |
| Aitzbitarte III interior | V - Gravettian | 1 | | | 1 | 18 |
| El Otero | IV - Magdalenian | | 2 | 3 | 5 | 68 |
| **TOTAL** | | **25** | **14** | **5** | **44** | **693** |

**Table 1.** Number of teeth sampled by species, archaeological sites and cultural periods.

**3.3 Sample treatment and stable isotope mass spectrometry**

Several authors have debated the necessity of chemical pre-treatments to remove organic matter and
secondary carbonates from bioapatite carbonates before stable isotopic analysis. Some chemical
treatments can introduce secondary carbonates, increase carbonate content, and alter the original isotopic
signal (Pellegrini and Snoeck, 2016; Snoeck and Pellegrini, 2015). For this reason, in this work, most of the
samples were not pretreated except for the equids and cervids samples from Labeko Koba, El Otero and El
Castillo that were sampled and pretreated in an earlier phase of the project. The absence of pretreatment
can elevate the risk of secondary carbonates (Chesson et al., 2021; France et al., 2020). Nonetheless, any
pretreatment method cannot guarantee their complete removal, and the 'side effects' may compromise the
final isotopic signal to a greater extent. While variations in pretreatment methods exist among samples in
this study, the lack of a universally accepted protocol necessitates careful consideration of any potential
isotopic effects resulting from these differences.
Pretreatment was followed for above-mentioned samples from fourteen teeth, where around 7 mg of
powdered enamel was prepared and pretreated with 3% of sodium hypochlorite (NaOCl) at room
temperature for 24 h (0.1 ml/mg sample) and thoroughly rinsed with deionised water, before a reaction with
0.1M acetic acid for 4 h (0.1 ml/mg sample) (Balasse et al., 2002; equivalent protocol in Jones et al., 2019).
Samples were then thoroughly rinsed, frozen, and freeze-dried. NaOCl is one of the most common agents
used for pretreating carbonates and works as a base that removes organic matter by oxidation. Although it
is considered one of the most efficient agents for removing organic matter, it can induce the absorption of
exogenous carbonates, such as atmospheric $CO_2$ and secondary carbonates (Pellegrini and Snoeck, 2016;
Snoeck and Pellegrini, 2015). It is argued that acetic acid after NaOCl pretreatment can remove exogenous
carbonates absorbed during NaOCl application. However, it is unclear if all newly introduced carbonates are
finally released and which effect they produce on the original isotopic composition. These samples were
analysed in the Godwin Laboratory (Department of Earth Sciences, University of Cambridge). Enamel
powder samples were reacted with 100% orthophosphoric acid for 2 h at 70°C in individual vessels in an
automated Gasbench interfaced with a Thermo Finnigan MAT253 isotope ratio mass spectrometer. Results
were reported in reference to the international standard VPDB and calibrated using the NBS-19 standard
(limestone, $\delta^{13}C$ = +1.95‰ and $\delta^{18}O$ = -2.2‰; Coplen, 2011) for which the precision is better than 0.08‰
for $\delta^{13}C$ and 0.11‰ for $\delta^{18}O$.
For the non-pre-treated samples, carbon and oxygen stable isotopic ratios were measured using continuous
flow-isotope ratio mass spectrometry, specifically a Europa Scientific 20-20 IRMS coupled to a
chromatograph, at the Iso-Analytical laboratory in Cheshire, UK. The samples were weighed into clean
exetainer tubes after being flushed with 99.995% helium. Phosphoric acid was then added to the samples,
and they were allowed to react overnight to ensure the complete conversion of carbonate to $CO_2$, following
the method outlined by Coplen et al. (1983). The reference materials used for VPDB calibration and quality
control of the analysis included IA-R022 (calcium carbonate, $\delta^{13}C$ = -28.63‰, $\delta^{18}O$ = -22.69‰), NBS-18
(carbonatite, $\delta^{13}C$ = -5.01‰, $\delta^{18}O$ = -23.2‰), IA-R066 (chalk, $\delta^{13}C$ = +2.33‰; $\delta^{18}O$ = -1.52). The accepted
values of the in-house standards IA-R022 and IA-R066 were obtained by calibrating against IAEA
international reference materials, NBS-18 and NBS-19, and NBS-18 and IAEA-CO-1 (Carrara marble, $\delta^{13}C$
= 2.5‰, and $\delta^{18}O$ = -2.4‰), respectively. Additionally, in-house standards long-term measured were used:
ILC1 (calcite, $\delta^{13}C$ = 2.13, $\delta^{18}O$ = -3.99‰), and Y-02 (calcite, $\delta^{13}C$ = 1.48, $\delta^{18}O$ = -9.59‰). The analytical
precision of quality control standard replicates was better than 0.09‰ for $\delta^{13}C$ and better than 0.12‰ for
$\delta^{18}O$. The calcium carbonate content test of these samples, ranging between 3.9% and 8.9%, does not
indicate a substantial presence of secondary carbonates, considering Chesson et al. (2021). Additionally,
phosphate results on samples from Axlor showed $\delta^{18}O_{carb}$-$\delta^{18}O_{phos}$ offsets within the expected range for well-
preserved samples (Pederzani et al., 2023).
**3.4 Carbon stable isotopic compositions as environmental and ecological tracers**
To unravel animal diet and compare the different species, in standardised terms, it is necessary to consider
the enrichment factor ($\varepsilon^*$) between $\delta^{13}C$ obtained by the animal on its diet ($\delta^{13}C_{diet}$) and $\delta^{13}C$ recorded on
enamel carbonates ($\delta^{13}C_{carb}$) (Bocherens, 2003; Cerling and Harris, 1999). The $\varepsilon^*$ estimated for large
ruminant mammals results in an offset of around 14.1‰ between diet and dental enamel, commonly applied
to medium-sized herbivores. However, it is well-known that this offset varies between species, considering
animals' different physiological parameters. Recently, a formal model to predict species-specific diet-
consumer isotopic offsets has been proposed, which uses body mass (BM) and digestive physiology as the
main factors that regulate the $\varepsilon^*$ (Tejada-Lara et al., 2018). This model proposes the following prediction
equations for ruminant or foregut fermenters (Equation 1: Eq.1) and hindgut fermenters (Eq. 2):
(Eq. 1) $\varepsilon^* = 2.34 + 0.05$ (BM)          [$r^2 = 0.78$; p-value=0.008]
(Eq. 2) $\varepsilon^* = 2.42 + 0.032$ (BM)          [$r^2 = 0.74$; p-value=0.003]
This work compares species with different digestive physiology, ruminants for bovines and cervids, and non-
ruminants for equids. The $\varepsilon^*$ value was adjusted for each animal to avoid bias from digestive physiology
when comparing these species. The following enrichment factors have been used: 14.6‰ for *Bos taurus*
(Passey et al., 2005a), 13.7‰ for *Equus caballus* (Cerling and Harris, 1999), and 13.2‰ for *Cervus elaphus*
(Merceron et al. (2021) following (Eq. 1) for ruminants with a mean body mass of 125 kg.
In body tissues, carbon isotopic composition is considered a combination of diet (understood as consumed
food), environment openness (and associated exposure to light), and the amount of precipitation. Assuming
that $\delta^{13}C$ of past vegetation is close to $\delta^{13}C_{diet}$ of ungulates, Lécuyer et al. (2021) proposed to estimate Mean
Annual Precipitations (MAP) from $\delta^{13}C_{carb}$, derived from diets based on C3 plants. After transforming $\delta^{13}C_{carb}$
to $\delta^{13}C_{diet}$ using the enrichment factors established above, this work suggested transforming this value to
$\delta^{13}C$ from vegetation ($\delta^{13}C_{leaf}$). However, the isotopic composition of animals' diet may not directly reflect
vegetation cover, but rather the food preference of the animal and this approach should be discussed
alongside other environmental data.
The MAP estimation is based on least square regression developed by Rey et al. (2013) and based on Kohn
(2010) dataset (Eq.4), which requires first to estimate the $\delta^{13}C_{leaf}$ (Eq. 3). The $\delta^{13}C$ values of atmospheric
$CO_2$ ($\delta^{13}C_{atm}$) are fixed in -7‰ (Lécuyer et al., 2021; Leuenberger et al., 1992; Schmitt et al., 2012).
Atmospheric $CO_2$ levels have varied throughout the Late Pleistocene, with $\delta^{13}C_{atm}$ range between -7 to -
6.4‰ (Eggleston et al., 2016), favouring an age-specific correction approach. However, maintaining general
corrections is preferred considering the chronological uncertainty of the studied levels.
(Eq.3) $\delta^{13}C_{leaf}$ (VPDB) = ($\delta^{13}C_{atm}$ - $\delta^{13}C_{diet}$) / [1+( $\delta^{13}C_{diet}$ / 1000)]
(Eq.4) Log1(MAP+300) = 0.092(±0.004) x $\delta^{13}C_{leaf}$ + 1.148(±0.074)

Additionally, Lécuyer et al. (2021) equation also accounts for the $pCO_2$ effect on $\delta^{13}C_{leaf}$ estimation, which
is expected to result in an offset of +1‰ from current levels (considering that $pCO_2$ was lower than that
experienced after the deglaciation period). If this correction was not applied, MAP results could be
underestimated by -150mm. In agreement with Lécuyer et al. (2021) appreciation, these MAP estimations
are a preliminary approximation and should be cross-validated with other environmental proxies. The
associated uncertainties range from ±100 to 200 mm, influencing the interpretation of the final values.
**3.5 Oxygen stable isotope compositions as environmental tracers**
Stable oxygen isotopes from meteoric water (mainly derived from rainfall) strongly correlate with mean air
temperatures in mid to high latitudes (Dansgaard, 1964; Rozanski et al., 1992) on a regional-to-local scale.
Obligate drinkers, like bovines and horses, acquire this water and record its isotopic composition in their
teeth and bones with a fixed but species-specific offset (Pederzani and Britton, 2019). Considering this two-
step relationship, past climatic conditions can be estimated. However, most of the temperature
reconstructions based on $\delta^{18}O$ have considered the $\delta^{18}O$ from the phosphate fraction of bioapatite enamel
($\delta^{18}O_{phos}$) to build linear correlations between tooth enamel and drinking water $\delta^{18}O$ and obtain climatic

information. For this reason, the $\delta^{18}O_{carb}$ values obtained in this work were converted into $\delta^{18}O_{phos}$. To do so, first, to express in VSMOW notation, the $\delta^{18}O_{carb}$ was corrected using the following correlation (Brand et al., 2014; Coplen et al., 1983):

(Eq.5) $\delta^{18}O_{carb}$ (VSMOW)= 1.0309 x $\delta^{18}O_{carb}$ (VPDB) + 30.91

Second, considering the relationship existent in tooth enamel between the carbonate and phosphate fraction (Iacumin et al., 1996; Pellegrini et al., 2011), from a compilation of the existent bibliography of modern animals measurements (Bryant et al., 1996; Pellegrini et al., 2011; Trayler and Kohn, 2017), Pederzani et al. (2023) proposed the following correlation:

(Eq.6) $\delta^{18}O_{phos}$ (VSMOW) = 0.941 x c (VSMOW) - 7.16

Once the isotopic information is expressed in $\delta^{18}O_{phos}$ (VSMOW), we can estimate the $\delta^{18}O$ on meteoric waters ($\delta^{18}O_{mw}$). It is known that different physiological factors will condition how oxygen isotope composition is fixed in each mammalian group. Thus, the correlations are usually species-specific and developed considering the physiology of each animal group. The obligate drinkers heavily rely on consuming large amounts of liquid drinking water, being the relative contribution of water from plants negligible and then minimizing the possible impact of isotopic enrichment through evapotranspiration in plants (Hoppe, 2006; Maloiy, 1973, Pederzani and Britton, 2019). However, certain types of drinking behaviours can impact $\delta^{18}O$, such as systematic consumption of certain highly buffered water sources (rivers or lakes), can significantly attenuate the final signal recorded. The correlation employed by this work relies on recent data compilations (Pederzani et al., 2021b, 2023). In the case of horses (Eq. 7), it has been considered the data combination of Blumenthal et al. (2019); Chillón et al. (1994); Bryant et al., 1994; Delgado Huertas et al., 1995), whereas for bovines (Eq. 8) the data from D'Angela and Longinelli (1990) and Hoppe (2006) have been put together in Eq. 4. To estimate $\delta^{18}O_{mw}$ from red deer remains, we selected D'Angela and Longinelli (1990) correlation (Eq. 9):

(Eq.7) $\delta^{18}O_{mw}$ (VSMOW)= ($\delta^{18}O_{phos}$ (VSMOW) - 22.14) / 0.62

(Eq.8) $\delta^{18}O_{mw}$ (VSMOW)= ($\delta^{18}O_{phos}$ (VSMOW) - 22.36) / 0.78

(Eq.9) $\delta^{18}O_{mw}$ (VSMOW)= ($\delta^{18}O_{phos}$ (VSMOW) - 24.39) / 0.91

Finally, paleotemperatures estimations from $\delta^{18}O_{mw}$ are typically approached using a geographically adjusted linear regression, which can vary from precise adjustments (aimed at reducing errors) to broader geographical adjustments that encompass more variability but are less precise (e.g., Pryor et al., 2014; Skrzypek et al., 2011; Tütken et al., 2007). In this work, temperatures were calculated considering the linear regression model relating $\delta^{18}O_{mw}$ and air temperatures proposed by Pederzani et al. (2021) based on monthly climatic records (monthly mean $\delta^{18}O_{mw}$ and monthly mean air temperatures), from Western, Southern and Central Europe stations from the Global Network of Isotopes in Precipitation (IAEA/ WMO, 2020). Considering current IAEA data sets from northern Iberia, there is a strong positive relationship between $\delta^{18}O_{mw}$ and annual or monthly temperatures (Moreno et al., 2021). However, it is known that Iberia is under a mixed influence between Atlantic and Mediterranean moisture sources that affects the isotopic composition of rainfall (Araguas-Araguas and Diaz Teijeiro, 2005; García-Alix et al., 2021; Moreno et al., 2021). Given uncertainties in past atmospheric circulation patterns and the limited availability of reference stations, it was deemed most appropriate to select an equation that extends beyond the borders of Iberia and incorporates higher variability. Different correlations were for mean annual temperature (Eq. 10), summer (Eq. 11), and winter (Eq. 12) temperatures (T):

(Eq.10) $\delta^{18}O_{mw}$ (VSMOW)= (0.50 x T) - 13.64

(Eq.11) $\delta^{18}O_{mw}$ (VSMOW) = (0.46 x T) - 14.70
(Eq.12) $\delta^{18}O_{mw}$ (VSMOW)= (0.52 x T) - 11.26
Nonetheless, oscillations between glacial and interglacial conditions in the past have influenced global ice
volume and sea level fluctuations (Dansgaard, 1964; Shackleton, 1987), impacting seawater oxygen isotope
composition and the surface hydrological cycle on a worldwide scale, including $\delta^{18}O_{mw}$ (Schrag et al., 2002).
Prior studies have used sea level information to correct $\delta^{18}O_{mw}$ (e.g., Fernández-García et al., 2019; Schrag
et al., 2002). Given the chronological uncertainty in the studied levels, a general correction was applied to
$\delta^{18}O_{mw}$ before temperature estimations, following Fernández-García et al. (2020) approach. Considering the
mean sea level descent for the MIS 3 period (50 meters below present-day sea level)(Chappell and
Shackleton, 1986), this may have contributed to a potential increase in the global $\delta^{18}O_{mw}$ value by ≈0.5‰,
inferring a bias in calculated air temperatures of ≈1ºC.
Due to the uncertainties incurred from converting stable isotope measurements to palaeotemperature, the
final estimations in this work should be considered exploratory and as a method of standardisation to make
results comparable among different sites, species, and other non-isotopic palaeoclimatic records. In these
estimations, the associated error from converting $\delta^{18}O_{phos}$ to MAT is enlarged by the uncertainty derived
from the transformation of $\delta^{18}O_{carb}$ (VPDB) to $\delta^{18}O_{phos}$ (VSMOW) (see Pryor et al., 2014; Skrzypek et al.,
2016 for further discussion). However, Pryor et al. (2014) and Pederzani et al. (2023) concluded that the
impact of this conversion is negligible compared to the error propagation in subsequent calibrations used
for temperature estimations from $\delta^{18}O_{phos}$. These associated errors were quantified following the
methodology outlined by Pryor et al. (2014) (Appendix B).
**3.6 Inverse modelling applied to intratooth profiles**
Intratooth profiles frequently provide a time-averaged signal compared to the input isotopic signal ($\delta^{13}C$/
$\delta^{18}O_{carb}$) during enamel formation (Passey et al., 2005b). This signal attenuation is caused by time-averaging
effects incurred through the extended nature of amelogenesis and tooth formation, and through the sampling
strategy. During mineralisation, the maturation zone, which is time-averaged, often affects a large portion of
the crown height and might affect the temporal resolution of the input signal of the sample taken. To obtain
climatically informative seasonal information on the analysed teeth, the inverse modelling method proposed
by (Passey et al. (2005b) is applied in this work. This method computationally estimates the time-averaging
effects of sampling and tooth formation to obtain the original amplitude of the isotopic input signal more
accurately, thus, to summer and winter extremes (Appendix E). This method considers parameters based
on the amelogenesis trends of each species and sampling geometry, which are critical for a meaningful
interpretation of intratooth isotope profiles. The model also estimates the error derived from the sampling
uncertainty and the mass spectrometer measurements to evaluate the data's reproducibility and precision.
This method was initially developed for continuously growing teeth, taking into account a constant growth
rate within a linear maturation model, with a progressive time-average increment as sampling advances
along the teeth profile. The species studied in this research exhibit non-linear tooth enamel formation,
particularly in later-forming molars (Bendrey et al., 2015; Blumenthal et al., 2014; Kohn, 2004; Passey and
Cerling, 2002; Zazzo et al., 2012). Although the model mentioned above is not ideal, as it does not take into
account non-linear enamel formation and specific growth parameters for the species included are unknown,
it is the best estimation based on the current state of the field and remains widely used (Pederzani et al.,
2021a, b, 2023). Flat and less sinusoidal profiles are less suitable for the application of the model, given its
inherent assumption of an approximately sinusoidal form. Therefore, we chose not to apply this methodology
in the analysis of intratooth $\delta^{13}C$ profiles, and it is recommended to approach the interpretation of model
outcomes for non-sinusoidal $\delta^{18}O$ curves with caution. Further details on the application of this method can
be found in Appendix E.

Following Pederzani et al. (2021b), mean annual temperatures (MAT) were deduced from the average of $\delta^{18}O_{carb}$ values between summer and winter detected in original sinusoidal intratooth profiles (Appendix D). This work shows that comparable results for annual means can be obtained before and after model application, but doing it beforehand avoids the associated errors induced by the inverse model. To maximize data, in non-sinusoidal teeth profiles, MAT was deduced from the average of all points within a tooth. However, this approach is less reliable when complete annual cycles are not recorded. When possible, summer and winter temperature estimations were derived from the obtained $\delta^{18}O_{carb}$ values after inverse modelling application, aiming to identify the corrected seasonal amplitude, which is dampened in the original $\delta^{18}O_{carb}$ signal.

### 3.7 Present-day isotopic and climatic data

Present-day climatic conditions surrounding each site have been considered, allowing an inter-site comparison, essential for compare this study with other regional and global data. Considering current MATs and MAPs, estimated climatic data is expressed in relative terms as MAT and MAP anomalies. Present-day summer and winter temperatures were also considered. Present-day temperatures and precipitation values were obtained from the WorldClim Dataset v2 (Fick and Hijmans, 2017) (Appendix B). This dataset includes the average of bioclimatic variables between 1970-2000 in a set of raster files with a spatial resolution every 2.5 minutes. The exact location of the selected archeo-palaeontological sites was used, using geographical coordinates in the projection on modern climatic maps with QGIS software.

Present-day $\delta^{18}O_{mw}$ values from the analysed sites' areas were obtained using the Online Isotopes in Precipitation Calculator (OIPC Version 3.1 (4/2017); Bowen, 2022) based on datasets collected by the Global Network for Isotopes in Precipitation from the IAEA/WMO (Appendix B).

| Site | Level | Culture | Species | Tooth type | Code | CCE (%) | n | δ13Ccarb VPDB (‰) | min | max | SD | Range | δ18Ocarb VPDB (‰) | min | max | SD | Range |
|---|---|---|---|---|---|---|---|---|---|---|---|---|---|---|---|---|---|
| Axlor | III | Mousterian | *Bos/Bison* sp. | LRM3 | AXL59 | 5.6 | 14 | -8.9 | -9.6 | -8.2 | 1.4 | 0.4 | -6.0 | -7.3 | -5.2 | 0.7 | 2.1 |
| Axlor | III | Mousterian | *Bos/Bison* sp. | LRM2 | AXL60 | 5.5 | 18 | -9.7 | -10.0 | -8.9 | 1.1 | 0.3 | -5.7 | -6.8 | -4.6 | 0.7 | 2.2 |
| Axlor | III | Mousterian | *Bos/Bison* sp. | LRM3 | AXL65 | 6.2 | 13 | -8.9 | -9.3 | -8.1 | 1.2 | 0.4 | -6.0 | -7.2 | -4.6 | 0.8 | 2.6 |
| Axlor | III | Mousterian | *Bos/Bison* sp. | LRM2 | AXL66 | 5.6 | 16 | -8.9 | -9.8 | -8.3 | 1.5 | 0.5 | -4.8 | -6.1 | -3.8 | 0.7 | 2.3 |
| Axlor | IV | Mousterian | *Bos/Bison* sp. | LRM2 | AXL70 | 5.7 | 12 | -9.1 | -9.4 | -8.6 | 0.7 | 0.3 | -5.3 | -7.3 | -3.9 | 1.2 | 3.4 |
| Axlor | VI | Mousterian | *Bos/Bison* sp. | LLM3 | AXL77 | 5.9 | 14 | -9.7 | -10.2 | -9.2 | 1.0 | 0.4 | -6.2 | -7.9 | -5.0 | 0.9 | 2.9 |
| Axlor | VI | Mousterian | *Bos/Bison* sp. | LLM3 | AXL86 | 5.5 | 18 | -9.9 | -10.2 | -9.3 | 0.9 | 0.3 | -5.4 | -6.5 | -3.8 | 0.7 | 2.6 |
| El Castillo | 20E | Mousterian | *Equus* sp. | LRP3/LRP4 | CAS60 | | 14 | -11.9 | -12.5 | -11.5 | 1.0 | 0.3 | -3.3 | -4.1 | -2.4 | 0.4 | 1.6 |
| El Castillo | 20E | Mousterian | *Equus* sp. | LRP3/LRP4 | CAS61 | | 14 | -12.2 | -12.4 | -12.1 | 0.3 | 0.1 | -4.9 | -5.8 | -4.3 | 0.4 | 1.5 |
| El Castillo | 20E | Mousterian | *Bos/Bison* sp. | LLM2 | CAS139 | 6.7 | 16 | -11.6 | -12.2 | -11.2 | 0.9 | 0.3 | -5.6 | -6.3 | -4.9 | 0.5 | 1.4 |
| El Castillo | 20E | Mousterian | *Bos/Bison* sp. | LLM2 | CAS140 | 5.7 | 12 | -11.5 | -11.9 | -11.1 | 0.8 | 0.3 | -5.5 | -6.3 | -4.6 | 0.5 | 1.7 |
| El Castillo | 21A | Mousterian | *Bos/Bison* sp. | LLM3 | CAS141 | 5.7 | 15 | -11.2 | -11.5 | -10.9 | 0.6 | 0.2 | -5.4 | -6.5 | -4.3 | 0.6 | 2.2 |
| El Castillo | 21A | Mousterian | *Bison priscus* | LLM3 | CAS142 | 6.1 | 14 | -11.2 | -11.7 | -10.9 | 0.7 | 0.2 | -5.0 | -5.7 | -4.4 | 0.4 | 1.3 |
| El Castillo | 21A | Mousterian | *Equus* sp. | LLM3 | CAS143 | 6.5 | 17 | -12.6 | -12.9 | -12.5 | 0.4 | 0.1 | -6.2 | -7.2 | -5.4 | 0.5 | 1.8 |
| El Castillo | 18B | Transitional Aurignacian | *Bos/Bison* sp. | ULM2 | CAS132 | 6.2 | 13 | -11.3 | -11.5 | -10.9 | 0.6 | 0.2 | -6.2 | -7.4 | -4.9 | 0.7 | 2.6 |
| El Castillo | 18B | Transitional Aurignacian | *Bos/Bison* sp. | ULM2 | CAS133 | 6.8 | 18 | -10.9 | -11.6 | -10.5 | 1.1 | 0.3 | -5.4 | -6.5 | -4.2 | 0.7 | 2.2 |
| El Castillo | 18B | Transitional Aurignacian | *Bos/Bison* sp. | ULM2 | CAS134 | 6.6 | 18 | -12.4 | -12.8 | -11.6 | 1.2 | 0.3 | -6.3 | -6.3 | -4.5 | 0.5 | 1.8 |
| El Castillo | 18C | Transitional Aurignacian | *Bos/Bison* sp. | LLM3 | CAS135 | 6 | 17 | -11.3 | -11.5 | -11.0 | 0.5 | 0.2 | -6.1 | -6.6 | -5.5 | 0.3 | 1.1 |
| El Castillo | 18C | Transitional Aurignacian | *Bos/Bison* sp. | LLM3 | CAS136 | 5.8 | 17 | -12.0 | -12.5 | -11.7 | 0.9 | 0.2 | -5.8 | -6.7 | -5.0 | 0.6 | 1.7 |
| El Castillo | 18C | Transitional Aurignacian | *Bos/Bison* sp. | LLM3 | CAS137 | 6.6 | 14 | -10.2 | -10.6 | -9.9 | 0.7 | 0.2 | -5.8 | -6.5 | -4.1 | 0.7 | 2.4 |
| El Castillo | 18C | Transitional Aurignacian | *Bos/Bison* sp. | LLM3 | CAS138 | 6.1 | 18 | -11.6 | -11.8 | -11.4 | 0.4 | 0.1 | -5.3 | -5.9 | -4.8 | 0.3 | 1.2 |
| El Castillo | 18B | Transitional Aurignacian | *Cervus elaphus* | ULM2+ULM3 | CAS8 | | 11 | -13.0 | -14.9 | -12.1 | 2.8 | 1.0 | -6.8 | -10.4 | -4.1 | 2.1 | 6.3 |
| El Castillo | 18B | Transitional Aurignacian | *Equus* sp. | ULP3/ULP4 | CAS58 | | 19 | -11.7 | -11.8 | -11.5 | 0.3 | 0.1 | -6.6 | -7.5 | -5.6 | 0.5 | 1.8 |
| El Castillo | 18B | Transitional Aurignacian | *Equus* sp. | LLP3/LLP3 | CAS59 | | 14 | -11.5 | -11.7 | -11.0 | 0.7 | 0.2 | -4.0 | -4.7 | -3.5 | 0.4 | 1.2 |
| Labeko Koba | IX inf | Chatelperronian | *Equus* sp. | URM3 | LAB38 | | 17 | -12.0 | -12.2 | -11.9 | 0.3 | 0.1 | -6.6 | -7.7 | -5.9 | 0.5 | 1.9 |
| Labeko Koba | IX inf | Chatelperronian | *Cervus elaphus* | LLM2 | LAB02 | | 7 | -12.3 | -12.4 | -12.1 | 0.3 | 0.1 | -4.7 | -6.0 | -3.7 | 1.0 | 2.3 |
| Labeko Koba | VI | Aurignacian | *Equus* sp. | URM2 | LAB20 | | 16 | -12.0 | -12.2 | -11.8 | 0.4 | 0.1 | -5.3 | -6.1 | -4.4 | 0.6 | 1.7 |
| Labeko Koba | V | Aurignacian | *Equus* sp. | LRM3 | LAB42 | | 17 | -11.9 | -12.3 | -11.4 | 0.7 | 0.2 | -5.7 | -6.6 | -5.0 | 0.5 | 1.6 |
| Labeko Koba | IV | Aurignacian | *Equus* sp. | LRM2 | LAB36 | | 17 | -11.6 | -11.8 | -11.3 | 0.6 | 0.2 | -5.9 | -6.2 | -5.5 | 0.2 | 0.7 |
| Canyars | I | Aurignacian | *Equus* sp. | URM3 | CAN01 | 7.8 | 12 | -10.0 | -10.4 | -9.5 | 0.9 | 0.3 | -4.8 | -5.3 | -4.3 | 0.3 | 1.1 |
| Canyars | I | Aurignacian | *Equus ferus* | URM3 | CAN02 | 6.2 | 17 | -10.5 | -10.7 | -10.3 | 0.4 | 0.1 | -4.4 | -5.0 | -3.6 | 0.5 | 1.4 |
| Canyars | I | Aurignacian | *Equus ferus* | URP3/URP4 | CAN03 | 6.4 | 17 | -10.7 | -11.2 | -10.2 | 0.8 | 0.2 | -5.3 | -6.0 | -4.0 | 0.4 | 1.4 |
| Labeko Koba | VII | Aurignacian | *Bos primigenius* | LRM3 | LAB53 | 5.2 | 23 | -9.5 | -10.1 | -8.7 | 1.4 | 0.3 | -5.7 | -7.0 | -4.2 | 0.9 | 2.8 |
| Labeko Koba | VII | Aurignacian | *Bos primigenius* | LRM3 | LAB55 | 5.6 | 23 | -10.4 | -11.5 | -9.8 | 1.6 | 0.3 | -5.1 | -7.0 | -2.7 | 1.2 | 4.3 |
| Labeko Koba | VII | Aurignacian | *Bos/Bison* sp. | LRM3 | LAB62 | 6.5 | 21 | -9.7 | -10.2 | -9.1 | 1.2 | 0.3 | -7.2 | -8.1 | -6.2 | 0.6 | 2.0 |
| Labeko Koba | V | Aurignacian | *Bos primigenius* | LRM3 | LAB69 | 5.5 | 21 | -9.3 | -10.2 | -7.3 | 3.0 | 1.0 | -7.2 | -8.8 | -5.5 | 0.9 | 3.3 |
| Canyars | I | Aurignacian | *Bos primigenius* | ULM3 | CAN04 | 6.8 | 14 | -9.3 | -9.8 | -8.7 | 1.1 | 0.3 | -3.6 | -4.2 | -2.6 | 0.6 | 1.6 |
| Canyars | I | Aurignacian | Bos primigenius | ULM3 | CAN05 | 6.6 | 14 | -9.0 | -9.5 | -8.5 | 0.9 | 0.3 | -5.5 | -6.2 | -5.0 | 0.4 | 1.2 |
| Aitzbitarte III | V (int) | Gravettian | *Bos/Bison* sp. | LLM3 | AITI10 | 5.5 | 14 | -9.2 | -9.6 | -8.7 | 0.9 | 0.3 | -5.5 | -6.5 | -4.3 | 0.5 | 2.2 |
| El Otero | IV | Magdalenian | *Cervus elaphus* | LLM2+LLM3 | OTE1 | | 11 | -11.4 | -11.6 | -11.2 | 0.4 | 0.1 | -4.4 | -5.8 | -2.9 | 1.0 | 2.9 |
| El Otero | IV | Magdalenian | *Cervus elaphus* | LLM2+LLM3 | OTE5 | | 10 | -11.3 | -11.5 | -11.0 | 0.5 | 0.2 | -5.1 | -5.7 | -3.8 | 0.6 | 1.9 |
| El Otero | IV | Magdalenian | *Cervus elaphus* | LLM2+LLM3 | OTE6 | | 14 | -11.4 | -11.8 | -10.6 | 1.2 | 0.3 | -4.6 | -5.4 | -4.0 | 0.4 | 1.4 |
| El Otero | IV | Magdalenian | *Equus* sp. | LLP3/LLP4 | OTE11 | | 17 | -11.6 | -11.8 | -11.4 | 0.5 | 0.1 | -5.0 | -6.3 | -3.9 | 0.7 | 2.4 |
| El Otero | IV | Magdalenian | *Equus* sp. | LLP3/LLP4 | OTE12 | | 16 | -11.3 | -11.5 | -10.9 | 0.6 | 0.1 | -3.9 | -4.9 | -3.3 | 0.6 | 1.6 |

**Table 2.** Mean, maximum value (Max), minimum value (Min), and standard deviation (SD) of $\delta^{13}C$ and $\delta^{18}O$ values per archaeological site and level organised by cultural periods. CCE, calcium carbonate equivalent; n, number of intratooth subsamples measured. In tooth type: position (U, upper; L, lower); laterality (R, right; L, left); tooth (M, molar; P, premolar).

## 4. Results

In northwestern Iberia, specifically in the Vasco-Cantabrian region, the mean $\delta^{13}C_{carb}$ values range from -13‰ to -8.9‰, with a mean value of -11‰ (SD = 1.2‰) (Table 2; Table 3). Considering species' different enrichment factors, the $\delta^{13}C_{carb}$ were transformed in $\delta^{13}C_{diet}$, resulting in mean values that extend from -27‰ to -23.5‰ (Fig. 4). It must be considered that average values may reflect slightly different periods or be affected by seasonal bias because different teeth encompass diverse periods, but it has been verified in our teeth that the variations are limited when the seasonal information of the sequential sampling is incorporated (±0.2; Appendix B). The carbon isotopic composition varies between species. The bovines have generally higher mean $\delta^{13}C_{carb}$ (from -12.4‰ to -8.9‰) than the horses (from -12.6‰ to -11.3‰), whereas the red deer fall within the horses' range (from -13‰ to -11.3‰). Average values of $\delta^{18}O_{carb}$ in all Vasco-Cantabrian individuals extend between -7.2‰ and -3.3‰ (mean = -5.5‰; SD = 0.8‰). When transformed to $\delta^{18}O$ expected from meteoric waters ($\delta^{18}O_{mw}$), with species-adapted correlations, the $\delta^{18}O_{mw}$ values range from -10.6‰ to -5.5‰. Less clear patterns in $\delta^{18}O_{carb}$ are observed between bovines and horses, with mean values of -5.7‰ and -5.2‰, respectively. In northeastern Iberia, the site of Canyars, both species have relatively high $\delta^{18}O_{carb}$ values that fall inside the range of variation observed in the Cantabria region, between -5.5‰ and -3.6‰ in bovines and between -4.8‰ and -4.4‰ in horses.

| | | Vasco-Cantabrian region (NW Iberia) | | | | Northeastern Iberia | | | |
|---|---|---|---|---|---|---|---|---|---|
| | | $\delta^{13}C_{carb}$ VPDB (‰) | $\delta^{13}C_{diet}$ VPDB (‰) | $\delta^{18}O_{carb}$ VPDB (‰) | $\delta^{18}O_{mw}$ VSMOW (‰) | $\delta^{13}C_{carb}$ VPDB (‰) | $\delta^{13}C_{diet}$ VPDB (‰) | $\delta^{18}O_{carb}$ VPDB (‰) | $\delta^{18}O_{mw}$ VSMOW (‰) |
| Total | Mean | -11.0 | -25.1 | -5.5 | -8.0 | -9.9 | -24.0 | -4.6 | -7.1 |
| | Max | -8.9 | -23.5 | -3.3 | -5.5 | -9.0 | -23.6 | -3.6 | -5.0 |
| | Min | -13.0 | -27.0 | -7.2 | -10.6 | -10.7 | -24.4 | -5.5 | -7.9 |
| | Range | 4.1 | 3.5 | 3.9 | 5.1 | 1.7 | 0.8 | 1.9 | 2.9 |
| | SD | 1.2 | 0.9 | 0.8 | 1.2 | 0.8 | 0.3 | 0.7 | 1.2 |
| Bovines | Mean | -10.4 | -25.0 | -5.7 | -7.7 | -9.1 | -23.7 | -4.5 | -6.2 |
| | Max | -8.9 | -23.5 | -4.8 | -6.5 | -9.0 | -23.6 | -3.6 | -5.0 |
| | Min | -12.4 | -27.0 | -7.2 | -9.5 | -9.3 | -23.9 | -5.5 | -7.4 |
| | Range | 3.5 | 3.5 | 2.4 | 3.0 | 0.3 | 0.3 | 1.9 | 2.4 |
| | SD | 1.1 | 1.1 | 0.6 | 0.7 | 0.2 | 0.2 | 1.4 | 1.7 |
| Horses | Mean | -11.8 | -25.5 | -5.2 | -8.5 | -10.4 | -24.1 | -4.7 | -7.6 |
| | Max | -11.3 | -25.0 | -3.3 | -5.5 | -10.0 | -23.7 | -4.4 | -7.2 |
| | Min | -12.6 | -26.3 | -6.6 | -10.6 | -10.7 | -24.4 | -4.8 | -7.9 |
| | Range | 1.4 | 1.4 | 3.3 | 5.1 | 0.7 | 0.7 | 0.5 | 0.7 |
| | SD | 0.4 | 0.4 | 1.1 | 1.8 | 0.3 | 0.3 | 0.3 | 0.4 |

**Table 3.** Mean $\delta^{13}C$ from enamel carbonate ($\delta^{13}C_{carb}$) and diet ($\delta^{13}C_{diet}$), and $\delta^{18}O$ from enamel carbonate ($\delta^{18}O_{carb}$) and meteoric waters ($\delta^{18}O_{mw}$), by species on the Vasco-Cantabrian and northeastern Iberia areas. Max: maximum value; Min: minimum value; SD: standard deviation.

### 4.1 Axlor (Mousterian, ca. 80 ka BP - 50 ka cal BP)

A total of seven bovine teeth were included from levels III (n = 4), IV (n = 1), and VI (n = 2) of Axlor cave (Pederzani et al., 2023). The mean $\delta^{13}C_{carb}$ range from -9.9‰ to -8.9‰ ($\delta^{13}C_{diet}$= -24.5‰ to -23.5‰); whereas mean $\delta^{18}O_{carb}$ values are between -6.2‰ and -4.8‰ ($\delta^{18}O_{mw}$ = -8.3‰ and -6.5‰), indicating a range of variation around 1‰ and 1.4‰, respectively (Fig. 3; 4). Considering isotopic compositions by levels, mean $\delta^{13}C_{carb}$ decreases from level III to level IV, whereas mean $\delta^{18}O_{carb}$ remains stable through the sequence (Table 2; Appendix B). A range between 0.3‰ and 0.5‰ is observed in $\delta^{13}C_{carb}$ variation within tooth profiles. Individuals show clear $\delta^{18}O$ sinusoidal profiles, with peaks and troughs and intratooth ranges from 2.1‰ to 3.4‰. The $\delta^{18}O_{mw}$ after inverse modelling intratooth profiles range from -9.1‰ to -7.35‰ (Appendix D; E). Mean Annual Temperatures (MATs) oscillated between 9.1ºC and 12.6ºC (MATAs = -3.1/+0.4ºC) (Table 4). From sinusoidal profiles, summer temperatures were extracted from peaks, resulting

from 15.4ºC to 23.7ºC, and winter temperatures from troughs provided values ranging from -7ºC to 10.8ºC.
Mean Annual Precipitation (MAPs), extracted from $\delta^{13}C_{carb}$, extend between 204mm and 326mm (MAPAs =
-843/-721mm). Based on these estimations, a non-clear climatic trend is observed through these levels.

**4.2 El Castillo (Mousterian and Transitional Aurignacian, 62.5 ka BP – 46.4 ka cal BP)**

From El Castillo, this work includes bovines (n = 11), horses (n = 5), and red deer (n = 1) teeth from the
Mousterian (21 and 20E) and the Transitional Aurignacian levels (18B and 18C). The mean $\delta^{13}C_{carb}$ values
are lower for horses, bovines, and red deer (-13‰ to -10.2‰) than other sites. Between -12.4‰ and -10.2‰
for bovines ($\delta^{13}C_{diet}$ = -24.6‰ to -25.8‰) and between -12.6‰ and -11.5‰ for horses ($\delta^{13}C_{diet}$ = -26.3‰ to
-25.2‰) (Fig. 3). The mean $\delta^{18}O_{carb}$ values extend from -6.8‰ and -3.3‰. Horses and bovines overlap in
their isotopic niche (Fig. 4), mainly due to the notably lower $\delta^{13}C_{carb}$ reported by bovines. The mean $\delta^{13}C_{carb}$
(-13‰) of the single red deer tooth is inside the variation range of bovines and horses but with a lower
$\delta^{18}O_{carb}$ mean value (-6.8‰). Considering these isotopic compositions by levels, bovine mean $\delta^{13}C_{diet}$ values
highly increase the variation range from Mousterian levels (20E and 21A) to Transitional Aurignacian levels
(18C and 18B). In contrast, horses increase mean $\delta^{13}C_{diet}$ values (Fig. 5). Bovine mean $\delta^{18}O_{mw}$ values
decrease from level 21A to level 18B, while horses from 18B have a large intra-level amplitude.
The mean $\delta^{18}O_{carb}$ values from horses have a more significant variation (range = 3.3‰) than bovines (range
= 2.2‰). All individuals show flat $\delta^{13}C_{carb}$ intratooth profiles (<0.4‰), except for red deer (1‰) (Appendix D).
Intratooth $\delta^{18}O_{carb}$ ranges of individuals are around 1-2‰ for horses and 1-3‰ for bovines. Some of the
individuals analyzed do not show non-complete annual cycles. No precise $\delta^{18}O_{carb}$ sinusoidal profiles are
detected in three teeth; the other six have particularly unclear profiles. After modelling, individual $\delta^{18}O_{carb}$
ranges oscillated between 2.7‰ and 7.4‰ (Appendix E). MATs oscillated between 4.6ºC and 12.6ºC
(MATAs = -8.8ºC/-0.9ºC), with mean summer temperatures from around 20.5ºC and mean winter
temperatures around -1.1ºC. MAPs extend between 376mm and 784mm (MAPAs = -656/-248mm) (Table
4). Non-important differences in rainfall estimations based on bovines and equids are noticed, probably
because they feed on similar ecological resources. Diachronic trends are unclear along the sequence but
mean annual and winter temperatures from levels 18C and 18C seem slightly lower. MAPs estimations
oscillated more in the upper levels.

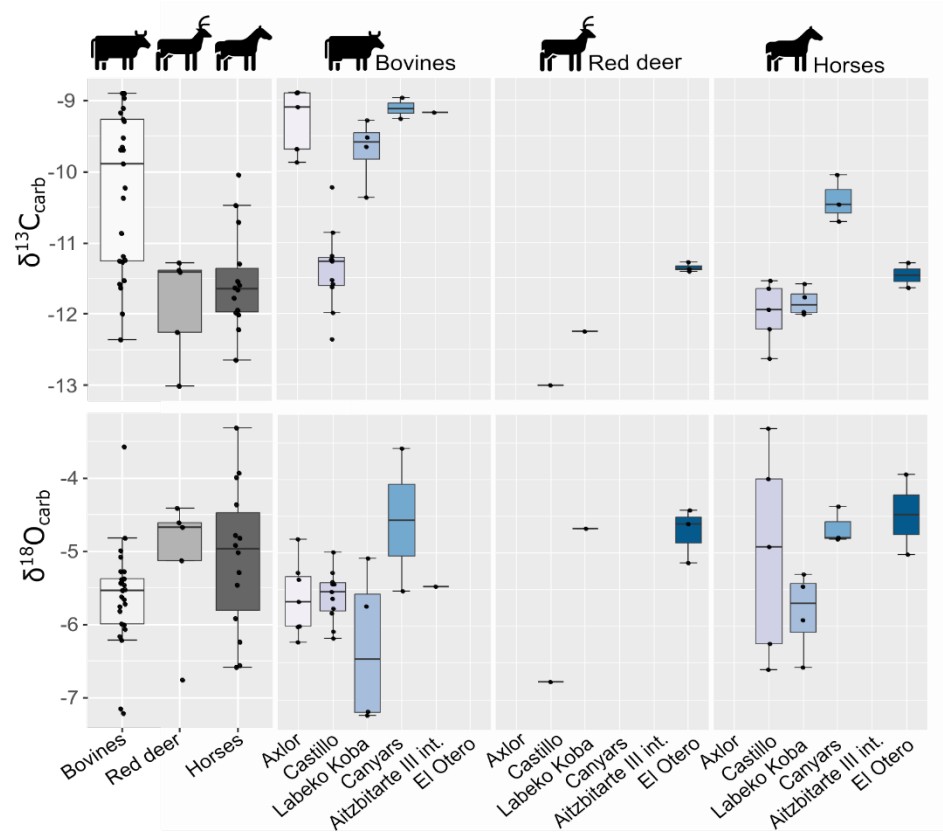


**Figure 3.** Distribution of mean carbon (δ¹³C$_{carb}$) and oxygen (δ¹⁸O$_{carb}$) isotopic values of enamel carbonate by species and archaeological site.

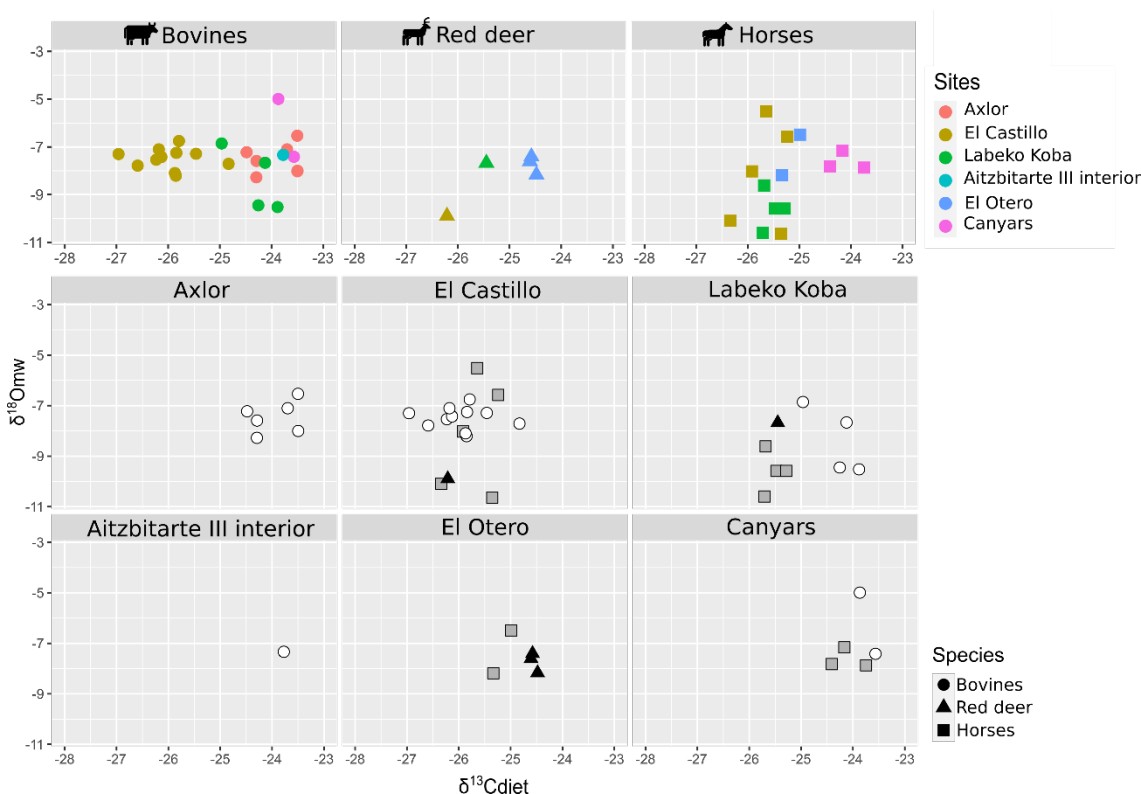


**Figure 4.** Biplot crossing δ¹³C from diet (δ¹³C$_{diet}$) and δ¹⁸O from meteoric waters (δ¹⁸O$_{mw}$) by species and archaeological site.
**4.3 Labeko Koba (Châtelperronian and Aurignacian, 45.1-36.3 ka cal BP)**
This work includes bovines (n = 4), horses (n = 4), and red deer (n = 1) teeth from levels related to
Châtelperronian (IXb inf), ProtoAurignacian (VII), and Aurignacian (VI, V, and IV). Significant differentiation
in mean $\delta^{13}C_{carb}$ between bovines and horses is observed, with higher values between -9.3‰ and -10.4‰
in bovines ($\delta^{13}C_{diet}$ = -25‰ to -23.8‰) than equids, whose values extend from -12‰ to -11.6‰ ($\delta^{13}C_{diet}$ = -
25.8‰ to -25.2‰) (Fig. 3;). These horses' values are within the ranges observed from this species in the
region. Red deer have similar $\delta^{13}C_{carb}$ values to those of horses ($\delta^{13}C_{carb}$ = -12.3‰; $\delta^{13}C_{diet}$ = -25.5‰). Mean
$\delta^{18}O_{carb}$ values are similar between species from -7.2‰ to -4.7‰ ($\delta^{18}O_{mw}$ = -8.5‰ to -6.1‰). However,
bovines have a very high variation within mean $\delta^{18}O_{carb}$ values (2.1‰), also reflected in the intratooth
profiles. These $\delta^{18}O$ values are lower than in other Vasco-Cantabrian sites, especially for two individuals in
levels VII and V (Table 3). Differences in $\delta^{13}C_{diet}$ values between bovines and horses result in isotopic niche
differentiation between both species (Fig. 4). The red deer niche is placed within the horses' niche. The
evolution of niche over time cannot be evaluated by levels due to the limited sample. Considering the isotopic
compositions by levels (Fig. 5), both bovines and horses experienced a slight increase in mean $\delta^{13}C_{diet}$ from
levels IX inf to IV, from Châtelperronian to Aurignacian. Mean $\delta^{18}O_{mw}$ values of bovines decrease from VII
to V, whereas horses increase from IXb inf to VI to decrease from VI to IV.
Variability of $\delta^{13}C_{carb}$ values in intratooth profiles is slightly higher (0.1-0.7‰), especially in bovines (0.3-
0.9‰), with more oscillating profiles than generally flat profiles observed in horses and red deer (Appendix
D; E). Intratooth profiles ranges of $\delta^{18}O_{carb}$ are also larger within bovines (2-4‰) than in horses (1-2‰).
Inverse-modelled individual $\delta^{18}O_{carb}$ ranges oscillated between 5-8‰ and 2-4‰, respectively. Sinusoidal
curves are observed in horses and bovines, but bovine profiles are noisier. The red deer has an extensive
$\delta^{18}O_{carb}$ range (6.3‰) from summer peak to an incomplete winter thought. We detect an inverse relation
between $\delta^{13}C_{carb}$ and $\delta^{18}O_{carb}$ in some points of these individual profiles. MATs oscillated between 5.2ºC and
11.4ºC (MATAs = -5.6/+1.1ºC), with summer temperatures from 14.5ºC to 27.3ºC and winter temperatures
from 1.9ºC to -4.9ºC. MAPs extend between 248mm and 521mm, notably drier than nowadays (MAPAs = -
798/-525mm) (Table 4). Lower rainfall levels and higher seasonal amplitudes are recorded along the
sequence, especially in samples from the ProtoAurignacian level VII. Relevant differences are noticed
between MAPs estimated from bovines and equids, the first providing more arid conditions.

### 478  4.4 Aitzbitarte III interior (Gravettian, 27.9 ka cal BP)

A single bovine individual was analysed from Gravettian level V located in the inner part of the cave. It has
a high mean $\delta^{13}C_{carb}$ (-9.2‰) considering the observed range in bovines from the Vasco-Cantabrian region,
whereas the $\delta^{18}O_{carb}$ mean value (-5.5‰) is inside the common $\delta^{18}O_{carb}$ variation observed (Fig. 3). The
mean $\delta^{13}C_{diet}$ value of -23.8‰ is comparable with Canyars and some individuals from Axlor but different
from Labeko Koba and El Castillo individuals. The individual $\delta^{13}C_{carb}$ fluctuation is slight (0.3‰) (Appendix
D; E). These teeth show not quite sinusoidal profile shape in $\delta^{18}O_{carb}$, with an intratooth range of around
2.2‰. Climatic information is extracted but may be considered cautiously due to the profile shape and the
limited sample size. From the inverse modelled mean $\delta^{18}O_{mw}$ value (-5.4‰), we estimate a MAT of 13ºC
(MATA = -0.4ºC) with a summer temperature of 19.7ºC and winter temperature of -2.9ºC. The MAP
estimation reached 235mm (-1127mm to nowadays) (Table 4).

### 489  4.5 El Otero (Magdalenian, ca. 17.3 ka cal BP)

Two equids and three cervids are included from level IV from El Otero, recently redate and chronologically
related to the Magdalenian (Marín-Arroyo et al., 2018). The mean $\delta^{13}C_{carb}$ values are close, between -11.4‰
and -11.3‰ for red deer ($\delta^{13}C_{diet}$ = -24.4‰ and -24.6‰) and -11.6‰ and -11.3‰ for horse ($\delta^{13}C_{diet}$ = -25.3‰
and -25.3‰) (Fig. 3). These $\delta^{13}C$ values for both species are relatively high concerning other studied
samples, especially for cervids (around +1-2‰). Both species have higher $\delta^{18}O_{carb}$ values concerning the
common range of variation observed in the Vasco-Cantabria region, between -5‰ and -3.9‰ for horses
and between -5.1‰ and -4.4‰ for red deer. When values are transformed to $\delta^{13}C_{diet}$ and $\delta^{18}O_{mw}$, equids

and cervids isotopic niches are separated (Fig. 4). All individuals show low amplitude $\delta^{13}C_{carb}$ intratooth profiles (<0.3‰), but especially equids with an intratooth variation around 0.1‰ (Appendix D; E). Equids and cervids show $\delta^{18}O_{carb}$ sinusoidal profiles, with intratooth ranges between 1.4‰ and 2.4‰. Climatic estimations are proposed only for equids, providing MATs estimations from 8.8ºC to 12.6ºC (MATAs = -4.9/-1ºC) and MAP between 400mm and 456mm (MAPAs = -755/-699mm) (Table 4). A high-temperature seasonality can be seen, with summer temperatures between 19.7ºC and 23.8ºC and winter temperatures from -10.4ºC to -3.1ºC.

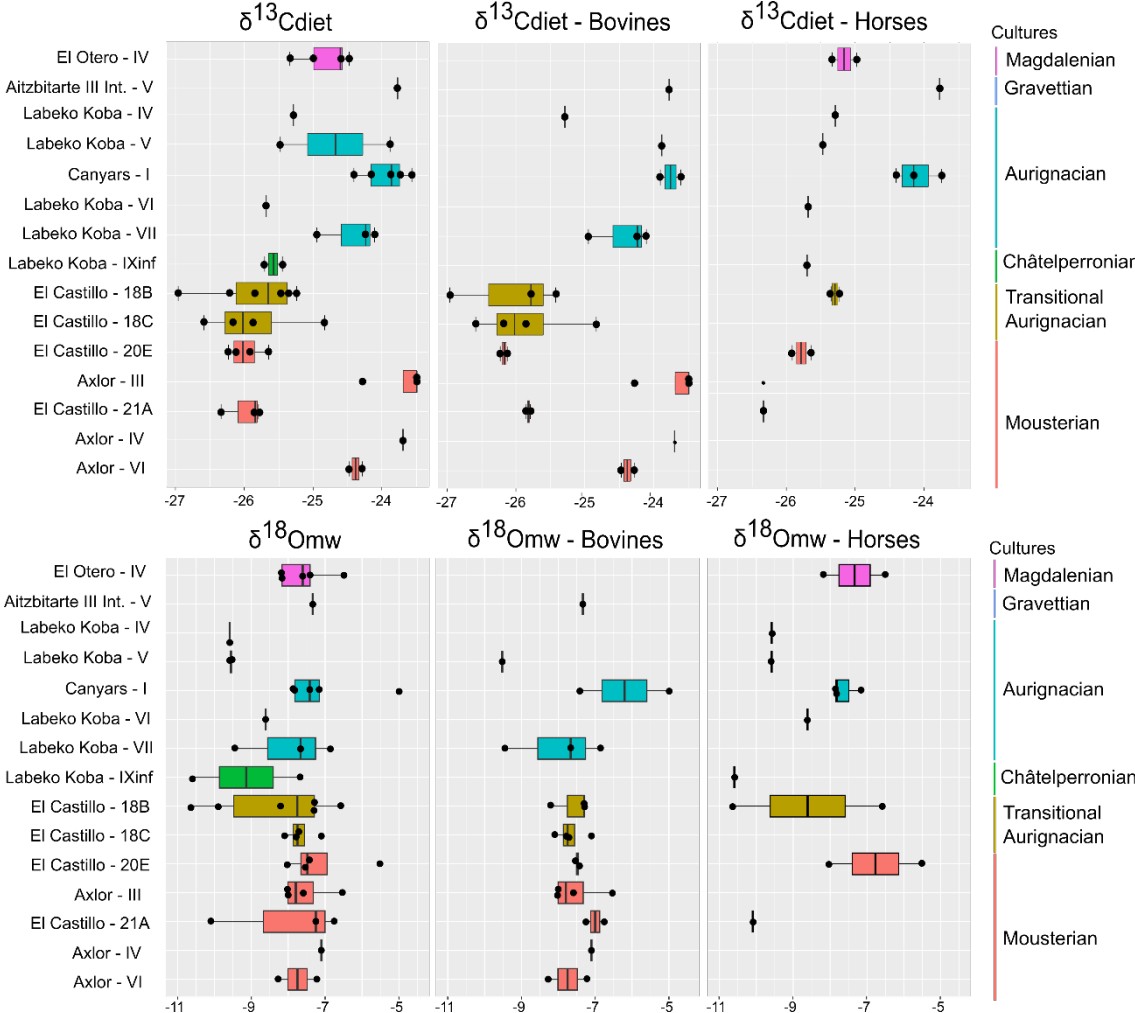

**Figure 5.** Evolution of $\delta^{13}C$ in diet ($\delta^{13}C_{diet}$) and $\delta^{18}O$ in meteoric waters ($\delta^{18}O_{mw}$) by archaeological levels in a diachronic order. From right to left: all species, including cervids, bovines and horses. Colours correspond to different chrono-cultures.

### 4.6 Canyars (Aurignacian, 39.7 ka cal BP)

From the archaeological level I at Canyars, corresponding to the Aurignacian, this work includes bovines (n = 2) and equids (n = 3) teeth. The mean $\delta^{13}C_{carb}$ values for bovines are between -9‰ to -9.3‰ ($\delta^{13}C_{diet}$ = -23.6‰ and -23.8‰), and for horses between -10‰ and -10.7‰ ($\delta^{13}C_{diet}$ = -23.7‰ and -24.4‰) (Fig.3). In this site, the $\delta^{13}C_{carb}$ values for horses are notably higher than in the Vasco-Cantabrian region (around +1-2‰) (Table 3). Both species have relatively high $\delta^{18}O_{carb}$ values, but they fall inside the range of variation observed in the Vasco-Cantabrian region, between -5.5‰ and -3.6‰ in bovines and between -4.8‰ and -4.4‰ in horses. Bovine and equid isotopic niches overlap (Fig. 4), but different responses are seen in mean $\delta^{18}O_{mw}$ values between the two bovines, with one high mean value but close $\delta^{13}C_{diet}$ mean values.

All individuals show flat $\delta^{13}C_{carb}$ intratooth profiles (<0.3‰ variation). Some individuals analysed do not show $\delta^{18}O_{carb}$ sinusoidal profiles, with intratooth profiles moderately flat and ranging from 1.1‰ to 1.6‰. We detect

an inverse relation between $\delta^{13}C_{carb}$ and $\delta^{18}O_{carb}$ in some points of bovine individual isotopic profiles. MATs oscillated between 9.8ºC and 11.9ºC (MATAs = -5.4ºC/-3.3ºC), with summer temperatures from 16.3ºC to 27.5ºC and winter temperatures from -0.5ºC to 1.8ºC (Table 4). MAPs extend between 211mm and 316mm (MAPAs = -431/-326mm). No substantial differences are noticed in the estimations based on bovines and equids because mean $\delta^{13}C$ diet values differed relatively little.

| Site | Sample | Level | Species | MAT (ºC) Estimated | MAT (ºC) Relative | Summer (ºC) Estimated | Summer (ºC) Relative | Winter (ºC) Estimated | Winter (ºC) Relative | Seasonality (ºC) | MAP (mm) Estimated | MAP (mm) Relative |
|---|---|---|---|---|---|---|---|---|---|---|---|---|
| Axlor | AXL59 | III | *Bos/Bison* sp. | 9.4 | -2.8 | 17.6 | -0.3 | -3.9 | -11.0 | 21.5 | 204 | -843 |
| | AXL60 | III | *Bos/Bison* sp. | 10.8 | -1.4 | 22.7 | 4.7 | 4.8 | -2.3 | 17.9 | 300 | -747 |
| | AXL65 | III | *Bos/Bison* sp. | 9.7 | -2.5 | 22.7 | 4.8 | -2.5 | -9.6 | 25.2 | 204 | -843 |
| | AXL66 | III | *Bos/Bison* sp. | 12.6 | 0.4 | 22.8 | 4.8 | -3.2 | -10.3 | 26.0 | 204 | -843 |
| | AXL70 | IV | *Bos/Bison* sp. | 11.1 | -1.1 | 21.9 | 3.9 | -8.0 | -15.1 | 29.9 | 227 | -820 |
| | AXL77 | VI | *Bos/Bison* sp. | 9.1 | -3.1 | 20.4 | 2.5 | -10.9 | -17.9 | 31.3 | 300 | -747 |
| | AXL86 | VI | *Bos/Bison* sp. | 11.1 | -1.1 | 25.9 | 8.0 | 3.1 | -4.0 | 22.8 | 326 | -721 |
| El Castillo | CAS141 | 21A | *Bos/Bison* sp. | 11.7 | -1.7 | 24.2 | 5.6 | -0.8 | -9.9 | 25.1 | 546 | -486 |
| | CAS142 | 21A | *Bison priscus* | 12.6 | -0.9 | 19.6 | 1.0 | 3.1 | -5.9 | 16.5 | 536 | -496 |
| | CAS143 | 21A | *Equus* sp. | 5.7 | -7.8 | 20.7 | 2.1 | -5.6 | -14.7 | 26.3 | 645 | -387 |
| | CAS60 | 20E | *Equus* sp. | | | | | 1.6 | -7.5 | | 510 | -522 |
| | CAS61 | 20E | *Equus* sp. | 9.7 | -3.8 | 25.9 | 7.3 | -4.1 | -13.2 | 30.1 | 561 | -471 |
| | CAS139 | 20E | *Bos/Bison* sp. | 11.2 | -2.3 | 18.8 | 0.2 | 1.8 | -7.3 | 17.0 | 622 | -410 |
| | CAS140 | 20E | *Bos/Bison* sp. | *11.3* | *-2.1* | | | | | | 602 | -430 |
| | CAS135 | 18C | *Bos/Bison* sp. | | | 17.0 | -1.6 | | | | 551 | -481 |
| | CAS136 | 18C | *Bos/Bison* sp. | *10.6* | *-2.9* | | | | | | 699 | -333 |
| | CAS137 | 18C | *Bos/Bison* sp. | | | | | 0.0 | -9.1 | | 376 | -656 |
| | CAS138 | 18C | *Bos/Bison* sp. | 11.8 | -1.7 | 18.3 | -0.3 | 3.1 | -6.0 | 15.3 | 612 | -420 |
| | CAS132 | 18B | *Bos/Bison* sp. | 9.8 | -3.6 | 26.3 | 7.6 | -1.2 | -10.3 | 27.5 | 548 | -484 |
| | CAS133 | 18B | *Bos/Bison* sp. | | | | | -0.1 | -9.2 | | 477 | -555 |
| | CAS134 | 18B | *Bos/Bison* sp. | | | | | 0.8 | -8.3 | | 784 | -248 |
| | CAS58 | 18B | *Equus* sp. | 4.6 | -8.8 | 13.5 | -5.1 | -11.2 | -20.3 | 24.7 | 460 | -572 |
| | CAS59 | 18B | *Equus* sp. | *13.0* | *-0.5* | | | | | | 440 | -592 |
| Labeko Koba | LAB38 | IX inf | *Equus* sp. | 5.2 | -7.4 | 14.5 | -4.1 | -1.8 | -9.1 | 16.2 | 521 | -526 |
| | LAB36 | IV | *Equus* sp. | 7.0 | -5.6 | 16.3 | -2.3 | -2.4 | -9.7 | 18.7 | 448 | -599 |
| | LAB42 | V | *Equus* sp. | 7.6 | -5.0 | | | | -7.3 | | 501 | -546 |
| | LAB69 | V | *Bos primigenius* | 6.3 | -6.3 | 17.3 | -1.2 | -4.9 | -12.2 | 22.2 | 248 | -799 |
| | LAB20 | VI | *Equus* sp. | 9.1 | -3.5 | 15.7 | -2.9 | -0.9 | -8.2 | 16.6 | 517 | -530 |
| | LAB53 | VII | *Bos primigenius* | 11.3 | -1.3 | 27.3 | 8.7 | -2.4 | -9.7 | 29.7 | 278 | -769 |
| | LAB55 | VII | *Bos primigenius* | 11.4 | -1.2 | 26.3 | 7.8 | 1.9 | -5.4 | 24.4 | 397 | -650 |
| | LAB62 | VII | *Bos/Bison* sp. | 7.2 | -5.4 | 20.6 | 2.1 | -2.9 | -10.2 | 23.5 | 295 | -752 |
| Canyars | CAN01 | I | *Equus* sp. | 9.8 | -5.4 | 16.3 | -5.9 | 1.7 | -7.5 | 14.6 | 232 | -410 |
| | CAN02 | I | *Equus ferus* | *11.9* | *-3.3* | | | | | | 284 | -358 |
| | CAN03 | I | *Equus ferus* | 10.4 | -4.7 | 18.6 | -3.6 | -0.5 | -9.7 | 19.1 | 316 | -326 |
| | CAN04 | I | *Bos primigenius* | 17.2 | 2.1 | 27.5 | 5.3 | | | | 247 | -395 |
| | CAN05 | I | *Bos primigenius* | 11.3 | -3.9 | 17.5 | -4.7 | 1.8 | -7.4 | 15.7 | 211 | -431 |
| Aitzbitarte III int | AITI10 | V | *Bos/Bison* sp. | 13.0 | -0.4 | 19.7 | 0.7 | -2.9 | -11.4 | 22.6 | 235 | -1127 |
| Otero | OTE11 | IV | *Equus* sp. | 8.8 | -4.9 | 19.7 | 0.9 | -10.4 | -19.8 | 30.1 | 456 | -699 |
| | OTE12 | IV | *Equus* sp. | 12.6 | -1.0 | 23.8 | 5.0 | -3.1 | -12.5 | 26.8 | 400 | -755 |

**Table 4.** Summary of paleoclimatic estimations, based on $\delta^{18}O$ for temperatures (Mean Annual Temperatures, MAT; summer; winter) and in $\delta^{13}C$ for precipitation (Mean Annual Precipitations, MAP). Summer and winter temperature estimations were obtained from teeth with clear seasonal profiles after modelling, while MAT was averaged between summer and winter before modelling. In profiles with an unclear seasonal shape, MAT was deduced from the original average of all teeth points (values marked in italics). Mean error associated to temperature estimations is 5.1±0.6 (see details in Appendix B). Seasonality is calculated as the temperature difference between summer and winter.

## 5. Discussion

### 5.1 Diet and ecological niches: carbon ratios

Carbon isotopic ratios are valuable indicators for discerning past animal diets, partially influenced by the physiology of the animal. Considering species trends in the studied sites, bovines have generally higher mean $\delta^{13}C_{carb}$ values (from -12.4‰ to-8.9‰) than horses (from -12.6‰ to -11.3‰), whereas the red deer fall within the horses' range (from -13‰ to 11.3‰). In the notheastern site of Canyars, bovines also show higher mean $\delta^{13}C_{carb}$ values (-9‰ to -9.3‰) compared to horses (-10.7‰ to -10‰). These differentiated isotopic ranges for equids and bovines can be potentially linked to feeding behaviour. Still, these species are expected to present different basal $\delta^{13}C_{carb}$ driven by their feeding behaviour and distinct physiological characteristics. Bovines, being ruminants, have been suggested in previous studies to exhibit higher $\delta^{13}C_{carb}$

values due to increased methane production (Cerling and Harris, 1999; Tejada-Lara et al., 2018). Therefore,
transforming $\delta^{13}C_{carb}$ to $\delta^{13}C_{diet}$ values using species-specific equations is crucial to mitigate the species-
specific impact, particularly when comparing ruminants and non-ruminants. Bovines report $\delta^{13}C_{diet}$ values
between -27.5‰ and -23.5‰ and horses between -26‰ and -25‰. These carbon compositions are typical
of animals feeding on C3 plants (commonly accepted range between -34‰ and -23‰), as can be expected
from high-latitude ecosystems during the Pleistocene (Bocherens, 2003; Cerling and Harris, 1999; Drucker,
547    2022).

Environmental factors such as light exposure, water stress, temperature fluctuations, salinity, and
atmospheric $CO_2$ changes can influence variations in $\delta^{13}C$ values in a diet primarily based on C3 plants
(Bocherens, 2003; Kohn, 2010). Typically, $\delta^{13}C_{diet}$ values below -27‰ ($\delta^{13}C_{carb}$ = -13‰) are associated with
animals feeding on C3 vegetation found in closed forested environments, whereas $\delta^{13}C_{diet}$ values between
-27‰ and -23‰ are linked to C3 open landscapes, which could include grasslands and steppe areas
(Bocherens, 2003). The relatively high $\delta^{13}C_{diet}$ observed here points to animals predominantly feeding in
open environments. The canopy effect, characterised by a depletion in $^{13}C$ isotopes due to dense tree cover,
seems unlikely among the analysed samples since none of the individuals reported $\delta^{13}C_{diet}$ below the
standard cut-off of -27‰ (Drucker et al., 2008; Kohn, 2010; van der Merwe, 1991). Therefore, in general
terms, open mosaic landscapes, ranging from light forests to meadows and grasslands, can be inferred for
northwestern Iberia. Given the generally higher $\delta^{13}C_{diet}$ values reported by bovines, it is likely that they were
foraging in more open environments than horses and can be considered predominantly grazers. Particularly,
bovines from El Castillo exhibit distinct feeding behaviour compared to other Vasco-Cantabrian sites, as
evidenced by their lower $\delta^{13}C_{diet}$ values, indicating a potential preference for browsing and feeding in closer
environments, possibly in lightly forested areas. Both extinct aurochs (*Bos primigenius*) and steppe bison
(*Bison priscus*) are usually classified as grass-dominant mix-feeders during the Pleistocene, although it
should be noted that modern European bison (*Bison bonasus*) could include browsing in their diet (Rivals
et al., 2022). For aurochs, a browse-dominated mixed feeding behaviour is also frequently described.
The $\delta^{13}C_{diet}$ range in equids also indicates feeding in open environments, suggesting a general mixed-
feeding pattern for the Vasco-Cantabrian region. However, individuals from northeastern Iberia are likely
grazing in more open environments, as evidenced by their notably higher $\delta^{13}C_{diet}$ values compared to the
Vasco-Cantabrian region (+1-2‰). Evaluating if other factors contribute to lower $\delta^{13}C_{diet}$ values in horses is
critical. In the case of equid from the Vasco-Cantabrian region, it should be considered that they have been
pretreated with a combination of NaClO and acetic acid, which could potentially affect the isotopic values.
Samples after organic removal pretreatment can potentially show either higher or lower $\delta^{13}C$ values and
higher $\delta^{18}O$ values based on previous experiments (Pellegrini and Snoeck, 2016; Snoeck and Pellegrini,
2015), with $\delta^{13}C$ values generally varying below 0.3‰. Based on the observation that horses in the Vasco-
Cantabrian region present lower $\delta^{13}C_{carb}$ values compared to bovines but similar mean $\delta^{18}O_{carb}$ value ranges,
the influence of the pre-treatment on our samples is deemed to be limited.
Furthermore, the high variability in $\delta^{18}O_{carb}$ values at El Castillo and Labeko Koba does not correlate with a
significant variation in $\delta^{13}C_{carb}$ values. Based on dental wear and stable isotopes analysis, Middle and Late
Pleistocene horses (*Equus ferus*) were primarily grazers, although some rare cases have been reported as
mixed feeders or browsers, such as at Igue des Rameaux and Schöningen (Kuitems et al., 2015; Rivals et
al., 2009, 2015; Uzunidis, 2020). Horse populations from northern and eastern Europe were found to be
browsers or mixed feeders, while those from the Mediterranean region tend to be grazers (Rivals et al.,
583    2022).

Finally, the few cervids included in this study exhibit $\delta^{13}C_{diet}$ values that frequently overlap with horses,
indicating a mixed feeding behaviour that varies from more closed environments in El Castillo to more open
habitats in El Otero. During the Pleistocene, the red deer (*Cervus elaphus*) exhibit a flexible, mixed-feeding
behaviour, consuming leaves, shrubs, forbs, grass, and sedges, similar to their present-day counterparts
(Merceron et al., 2021; Rivals et al., 2022). Today, this species inhabits diverse habitats ranging from
steppes to closed temperate forests.

**5.2 Seasonality, mobility and water acquisition: oxygen ratios and intratooth profiles**

Average values of $\delta^{18}O_{carb}$ in Vasco-Cantabrian individuals extend between -7.2‰ and -3.3‰ (Table 3).
Even if no clear species patterns in $\delta^{18}O_{carb}$ are observed, in general, bovines present slightly lower $\delta^{18}O_{carb}$
values from -7.2‰- to 4.8‰ than other species; horses have a significant variation from -6.6‰ to -3.3‰ and
red deer from -6.8‰ to -4.4‰. In Canyars, both species have relatively high $\delta^{18}O_{carb}$ values that fall inside
the variation range observed in the Vasco-Cantabrian region, between -5.5‰ and -3.6‰ in bovines and
between -4.8‰ and -4.4‰ in horses. Each species shows different $\delta^{18}O_{carb}$ intratooth ranges, with bovines
between 1‰ and 3‰, horses mostly around 1.5%, and red deer from 1‰ to 6‰ presenting the higher
ranges (Table 3; Appendix D). After applying inverse modelling to correct the dampening effect (Passey et
al., 2005b), the majority of teeth increase the $\delta^{18}O_{carb}$ intratooth range, between 3‰ and 8‰ for bovines and
2‰ and 7‰ for horses (Appendix E). Most bovines from Axlor and Labeko Koba and horses from El Castillo
and El Otero exhibit well-defined sinusoidal profiles in their $\delta^{18}O_{carb}$ and large intratooth individual ranges,
related to the predominant consumption of water sources that reflect seasonal fluctuations between summer
and winter. Although not all samples consistently follow this pattern, specific intratooth profiles, particularly
those from bovines in El Castillo and Canyars, exhibit sharp profiles with narrow ranges (<1.5‰). This
phenomenon was previously reported in the region in preliminary studies conducted at the sites of El Castillo
(Jones et al., 2019) and in the Magdalenian levels of El Mirón cave (Geiling, 2020).
Non-sinusoidal profiles observed in the data can be attributed to various factors, including sample
techniques and preservation issues and the inherent variability in the original isotopic signal. Factors related
to sampling and methods can be connected to 1) the sampling process (e.g. too deep or too distant sampling
grooves); 2) the imprecision of the mass spectrometer measurements; 3) uncontrolled effects of samples
pretreatments; 4) diagenetic alterations affecting the carbonate fraction. However, it must be noted that
technical reasons, whether related to sampling or pretreatment, do not appear to impact the obtained results
significantly. First, this study reproduces the same intratooth sampling methods that previously yielded
reliable results in similar research (e.g., Pederzani et al., 2023, 2021a). Second, non-significant alterations
in intratooth profiles of pretreated horse samples (El Castillo, Labeko Koba, Otero) are noticed in comparison
to untreated bovid samples (Appendix D). Some bovid samples show these non-sinusoidal profiles equally.
In sites where both species are analysed, no correlation is observed between $\delta^{18}O_{carb}$ and $\delta^{13}C_{carb}$. In tooth
enamel, diagenetic alterations are generally less pronounced than in bone due to its higher mineral content.
However, carbonates within tooth enamel can be more susceptible to diagenesis and recrystallisation
compared to the phosphate fraction, which contains a more extensive reservoir of oxygen and stronger
oxygen bonds (Zazzo et al., 2004; Chenery et al., 2012; Bryant et al., 1996). The carbonate content in our
samples, ranging from 3.9% to 8.9%, is similar to the proportion found in modern tooth enamel, suggesting
no immediate indication of diagenetic alteration. Diagenesis can also be evaluated by comparing the isotopic
values of the carbonate and phosphate fractions in a sample, as there is a predictable difference between
them. However, phosphate fraction measurements were still unavailable in our study, except at Axlor
(Pederzani et al., 2023) where good preservation was attested. Additionally, in the case of diagenetic
alteration, we would expect specimens from the same archaeological levels to be affected similarly, which
is not the case.
Based on these arguments, it is suggested that the non-sinusoidal $\delta^{18}O_{carb}$ signal observed in some
individuals may not be attributed to poor preservation; instead, it likely reflects the original isotopic signature
from water input, which appears to be non-seasonal. Several factors can explain why some teeth do not
reflect an evident seasonal fluctuation, which could be related to animals' mobility, the isotopic composition
of the water sources, and seasonal buffering within those water sources (Pederzani and Britton, 2019). The
main factors considered in our study are 1) the high mobility of the animals analysed among ecosystems
with different isotopic baselines due to large migrations; 2) the inland-coastal or short altitudinal movements
through the region, which lead to the acquisition of water from sources with different isotopic signal; and 3)
the acquisition of water from sources with no clear seasonal signal, such as large bodies of water, rivers,
groundwaters, or meltwaters. At mid-latitudes, the temperature effect is currently the dominant
factor. However, it is crucial to note that past changes in rainfall density (as the "amount effect"; Dansgaard,
1964) cannot be dismissed from having a more significant role then, particularly during glacial and arid
periods. These effects, with their potential to mask temperature oscillations, underscore the urgency and
importance of our research in understanding and predicting climate patterns. Furthermore, variability
between species and within the same species, even within populations living in the same habitat, is also
possible. This can be attributed to multiple factors, from minor differences in foraging and drinking behaviour
to slight metabolic and physiological variations, including body size, metabolic rate, breathing rate, moisture
content of food, and faeces, among others (Hoppe et al., 2004; Kohn, 1996; Magozzi et al., 2019).
Analyses of nitrogen and sulphur stable isotopes on ungulate bone collagen from Axlor, El Castillo and
Labeko Koba (Jones et al., 2018, 2019; Pederzani et al., 2023) have already revealed large variation ranges
linked to the existence of several microenvironments just in a few kilometres within the Vasco-Cantabria
region. Long migrations and long hunting distances cannot solely explain these diverse values because of
the range of species involved and their likely small-scale movements. In our study, the minimal $\delta^{13}C_{carb}$
intratooth variation within individuals (<1‰) indicates limited seasonal changes in their feeding behaviour
that influenced the carbon isotopic composition (Appendix D). Therefore, considering the diverse topography
of the Vasco-Cantabrian, characterized by steep valleys connecting the Cantabrian Cordillera with the
Atlantic Ocean through rivers over short distances (30-50 km), the availability in the past of a wide range of
water sources in small areas seems highly likely. Certain drinking behaviours can influence $\delta^{18}O$, as animals
may acquire water from various sources, with small streams better reflecting seasonal isotopic oscillations
than large lakes or evaporating ponds (see synthesis in Pederzani and Britton, 2019). Systematic
consumption of highly buffered water sources can significantly attenuate the final recorded signal.
Furthermore, rivers in the region frequently contain meltwater from snow during the winter-spring months
and water springs.
**5.3 Regional trends and ecological niches**
This study provides valuable insights despite the limited sample size at each archaeological level. It
establishes a baseline of isotopic values for northern Iberia, allowing for the evaluation of regional trends.
In the northwest, in the Vasco-Cantabrian region, the $\delta^{13}C_{carb}$ values obtained oscillated between -13‰ and
-8.9‰ and between -7.2‰ and -3.3‰ in the case of $\delta^{18}O_{carb}$ values. These values are within the range
expected, considering previous regional studies in ungulates (Carvalho et al., 2022; Jones et al., 2019;
Lécuyer et al., 2021; Pederzani et al., 2023). Although oxygen variability trends are less precise, the main
factor distinguishing the observed changes over time is the variation of carbon isotopic composition among
species and regions. The combination of mean $\delta^{13}C_{diet}$ and $\delta^{18}O_{mw}$ values (Fig. 4; 5) accentuates disparities
in ecological niche overlap between horses and bovines, whereas cervids and horses frequently exhibit
shared ecological niches. The dissimilarities between bovines and horses could be attributed to shifts in
feeding behaviour, which may be accompanied by ecological and environmental changes, either
independently or in parallel.
Comparing the entire dataset and across all sites, the consistently lower $\delta^{13}C_{diet}$ values in horses compared
to bovids throughout time suggest both animals inhabited open landscapes, with bovines exhibiting a grazer
preference while horses show a mix-feeding diet. Only in the Middle-to-Upper Paleolithic transition 18B and
18C levels of El Castillo, an exception is observed with lower $\delta^{13}C_{diet}$ values in bovines, linked to a higher
browser input due to a higher habitat in closer environments, such as open forests, similar to those inhabited
by the horses. This generates a niche overlapping between horses and bovines, most likely reflecting stable
conditions that could support both species in similar ecosystems. Contrarily, in the Châtelperronian and
early Aurignacian levels from Labeko Koba, a clear differentiation between horses and bovines is observed,
mainly in $\delta^{13}C_{diet}$ values, highlighting the occupation of different parts of the landscape by both species. This
spatially-driven niche separation between species could result from resource competition derived from an
unstable climatic period, where species needed to specialise to adapt to the changing conditions. Notable
changes are also observed in the $\delta^{18}O_{carb}$ values from Labeko Koba compared to the older El Castillo and
Axlor sites, with bovines exhibiting a higher fluctuation range and the lowest values in the region. These
trends are consistent with values observed on bone collagen from previous studies in these sites. During
the Middle-to-Upper Paleolithic transition in the region, by comparing horses and red deer, a decrease in
mean $\delta^{13}C$ (from -21‰ to -20‰) and $\delta^{15}N$ values (from 2.5‰ to 6‰) in bone collagen was observed in
contrast to stable red deer mean $\delta^{13}C$ (Fernández-García et al., 2023; Jones et al., 2018, 2019). This
decrease was previously interpreted as niche fractionation, derived from an opening landscape, that drove
equids into low-quality pastures compared to cervids. Pollen evidence in the region suggests a prevalence
of steppe vegetation and low tree cover for the Châtelperronian and Aurignacian (Iriarte-Chiapusso, 2000).
In the same period, Canyars in the northeastern area, higher mean $\delta^{13}C_{diet}$ are observed in both species
(between -23.6‰ and -24.4‰), indicating a preference for more open landscapes by bovines and equids.
The indication of open areas could be linked to the arid climatic conditions associated with the Heinrich
Stadial 4, which coincides with the formation of the studied level. This predominance of open areas coincides
with the presence of typical steppe herbivore species, such as *Equus hydruntinus* and *Coelodonta*
*antiquitatis*, the microfauna and pollen taxa, and the data offered by the use-wear analysis on ungulate
remains identified at the site (Daura et al., 2013; López-García et al., 2022; Rivals et al., 2017).
Aridity is a plausible explanation for the higher niche partitioning observed in Labeko Koba and the higher
$\delta^{13}C_{diet}$ values found in Canyars for both species during the Aurignacian. The $\delta^{13}C_{diet}$ results of bovines from
Aitzbitarte III interior during the Gravettian are consistent with the trend observed in Labeko Koba, where
previous studies have already suggested this time to be notably arid and cold (Arrizabalaga et al., 2010).
Finally, in the Magdalenian level of El Otero, higher $\delta^{13}C_{diet}$ values resemble those observed in Canyars.
However, this time, carbon values are related to niche partitioning between horses and red deer. In contrast,
higher $\delta^{18}O_{mw}$ values might indicate warmer conditions but are still associated with open landscapes in the
Vasco-Cantabrian area.
**5.4 Late Pleistocene climatic evolution in Northern Iberia**
Carbon and oxygen isotopes were used to estimate quantitative parameters related to past temperatures
and precipitation. In the case of oxygen isotopic compositions, an evaluation of environmental water
composition can be addressed before approaching temperature estimations. When transformed to $\delta^{18}O_{mw}$
using species-adapted correlations and correcting bias in sea water $\delta^{18}O_{mw}$, the summer $\delta^{18}O_{mw}$ values
obtained from the modelled teeth range from -8.9‰ to -2.2‰, while the winter values range from -17.1‰ to
-8.9‰. These values can be tentatively compared with the current trends observed in $\delta^{18}O_{mw}$ range recorded
by the IAEA station (IAEA/ WMO, 2022) in Santander (from -3.5‰ in summer to -6.6‰ in winter) and in
Barcelona (from -2.2‰ in summer to -6.3‰ in winter) and the OIPC (Bowen, 2022) estimations for studied
locations (from -1‰ to -9‰) (Appendix B). As observed in the present, Canyars exhibit mean annual $\delta^{18}O_{mw}$
values around -8.2‰, which is lower than the current $\delta^{18}O_{mw}$ estimated for this location (-5.4‰) but higher
than Labeko Koba mean annual $\delta^{18}O_{mw}$ (-9.5‰). This raises the question of whether the baseline $\delta^{18}O_{mw}$
differences between Canyars and the other sites can be attributed to Mediterranean influence rather than
the Atlantic, assuming equivalent air circulation patterns and moisture sources experienced in the past as
in the present (Araguas-Araguas and Diaz Teijeiro, 2005; García-Alix et al., 2021; Moreno et al., 2021).
However, it's important to note that these comparisons must be approached thoughtfully, considering that
moisture fluxes and precipitation trends may have varied significantly during the Pleistocene and the
Holocene (Dansgaard, 1964; Shackleton, 1987).
As indicated by the climate reconstructed here, temperatures were colder, and precipitation levels were
notably lower in the Late Pleistocene period in this region than they are nowadays (Table 4; Appendix B).
From 80 to 50 ka BP, in the Mousterian levels of Axlor, temperatures were slightly colder than today, but
older levels showed higher differences between summer and winter temperatures. Rainfall estimations
exhibit an unusual arid pattern, possibly affected by bovines predominantly feeding in open areas at that
time. This aligns with the impact of basal feeding behaviour on rainfall estimations, as previously advised by
Lécuyer et al. (2021). In this case, it is not possible to isolate the effect of diet from environmental
interference, but previous studies have highlighted stable climatic conditions at the site (Pederzani et al.,
2023). Climatic reconstruction, relying on a compilation of lake sediments from northern Iberia (Moreno et
al., 2012) suggests that from late MIS4 to 60 ka cal BP, cold but relatively humid conditions predominated,
with drier conditions emerging later. Additionally, stalagmites from the Ejulve cave in the Iberian range
indicate a dry climate until 65.5 ka BP, preceding HE6, followed by more humid conditions afterwards (Pérez-
Mejías et al., 2019).
During the late Middle Paleolithic and early Aurignacian occupations, the observed shift in the niche
configuration of species suggests potential climatic perturbations. There is a decreasing trend in
temperatures from the Transitional Aurignacian levels in El Castillo (18C and 18B; ca. 47-46 ka cal BP) to
the Châtelperronian (Xinf; 45.1 ka cal BP) and Early Aurignacian (VII-V; from 40.7 to 36.3 ka cal BP) levels
in Labeko Koba. Lower mean annual and winter temperatures are particularly notable at El Castillo and
Labeko Koba. Labeko Koba levels exhibit high seasonal amplitude, especially at level VII. Additionally, there
is a slight decrease in rainfall and increased fluctuations from the Transitional Aurignacian levels from El
Castillo (18B-18C) to the Aurignacian levels in Labeko Koba (VII-V). Previous studies in the northern Iberian
region underlined an environmental and ecological shift after GS13/HE5, from 48 to 44 ka cal BP, based on
a progressive trend to colder temperatures, aridity increase, and open environmental conditions, matching
with the late Neanderthal occupations, followed by a population hiatus before the arrival of Anatomically
Modern Humans (Fernández-García et al., 2023; Vidal-Cordasco et al., 2022). This episode coincides with
the maximum extent of glaciers in this region, as recorded in Lake Enol and Vega Comeya and an significant
decrease in plant biomass and herbivore abundance around 44 to 38 ka BP (Ballesteros et al., 2020;
Jiménez-Sánchez et al., 2013; Ruiz-Fernández et al., 2022). Moreover, previous isotopic analyses in the
region pointed to some ecological alterations considering perturbations observed in the $\delta^{13}$C and $\delta^{15}$N of
bone collagen (Jones et al., 2018, 2019). This tendency of increased aridity aligns with observations made
in regional lake sediments from northern Iberia between 60 and 23.5 ka cal BP, marked by abrupt climate
changes associated with HE (Moreno et al., 2012). Supporting this, the marine core MD04-2845 in the
northern margin of Iberia reveals a decline in the Atlantic forest and an expansion of steppe and cold grasses
from 47 to 40 ka BP (Fourcade et al., 2022).
When comparing the environmental reconstruction of the Aurignacian period between the Vasco-Cantabrian
(levels V-IV from Labeko Koba) and the northeastern region (Layer I from Canyars), which are synchronous
to HE4 (39 ka BP), this study reveals notably lower rainfall levels for the latter. This is due to the feeding
behaviour observed in animals, mainly in open areas. However, these drier conditions align with the specific
climatic conditions expected for this period and support previous findings revealing aridity and the
predominance of open landscapes (Daura et al., 2013; Rivals et al., 2017). The temperature data indicates
that, at Canyars, colder conditions were experienced, especially during the winter season, compared to the
present. However, in comparison to Labeko Koba, Canyars experienced warmer conditions. As explained
earlier, the Mediterranean basin had consistently higher temperatures, even during colder periods. This is

consistent with the persistence of Mediterranean open forests in the surroundings, as indicated by other studies (López-García et al., 2013; Rivals et al., 2017). Continuous natural records are lacking in the northeastern Iberian margin. However, the inland stalagmite record from Ejulve Cave (Pérez-Mejías et al., 2019) and the sedimentary lacustrine sequence of Cañizar de Villarquemado (González-Sampériz et al., 2020) have identified the most arid intervals during HE5 and HE4. These periods were characterized by steppe vegetation expansions, followed by deciduous woodland expansion. To the south, the Padul sequence agrees with cold and dry conditions alternating with forest recovery (Camuera et al., 2019), as documented in the Alborean Sea (Martrat et al., 2004).

Finally, the sites Aitzbitarte III interior (27.9 ka cal BP) and El Otero (17.3 ka cal BP) provided valuable climatic insights into the Vasco-Cantabrian region during the Upper Paleolithic, specifically during the Gravettian and Magdalenian, respectively. Considering previous research in the region, the climatic trend reported for the Aurignacian, characterised by colder and more arid conditions, was expected to continue or even intensify during the Gravettian (Fernández-García et al., 2023; Garcia-Ibaibarriaga et al., 2019b; Lécuyer et al., 2021). Both sites indicate lower precipitation than today in this area, indicating significant aridity, with ungulates feeding predominantly in open landscapes. However, El Otero's higher mean annual temperatures recorded in the Magdalenian horses respect to other sites within the Vasco-Cantabrian, are consistent with a climatic amelioration following the Last Glacial Maximum (Jones et al., 2021). MIS 2 is marked by the most extreme glacial conditions, as indicated by NGRIP and marine cores in Iberian margins (Martrat et al., 2004; Sánchez Goñi et al., 2002). However, other regional proxies, such as lake sediment and the stalagmite sequence in Pindal Cave (Moreno et al., 2010), suggest a complex and highly variable climate during MIS 2. These proxies identify the coldest and most arid period within MIS 2 as the interval from 18 to 14 ka cal BP rather than the global Last Glacial Maximum (23 to 19 ka cal BP).

## 5. Conclusions

This study provides a detailed analysis of the temporal evolution of the environment and climatic conditions in northern Iberia, spanning from the Middle Paleolithic to the late Upper Paleolithic, this is from the GS21 to the GS2, ranging from 80 ka BP to 17 ka cal BP. In the Vasco-Cantabrian region, the results reveal a heterogeneous open mosaic landscape, ranging from light forest to meadows and grasslands. This landscape reconstruction is primarily inferred by the feeding locations of the studied animals and, consequently, related to the ecosystems where hominins captured them. Despite shifts in niche configuration observed between equids and bovines, both species typically foraging in open areas, with bovines showing a higher preference for grazing. Only in El Castillo, during the late Mousterian and the Transitional Aurignacian levels, bovines show unusually low $\delta^{13}C_{diet}$ related to higher browsing and overlapping with horse isotopic niche. This might indicate a slightly closed mosaic landscape that could sustain both species. In contrast, only horses from Canyars exhibit a preference for grazing behaviour.

Stable climatic conditions are described for Mousterian in Axlor and El Castillo levels from 80 to 50 ka cal BP. However, some elements indicate environmental perturbations initiated during the Transitional Aurignacian levels of El Castillo, around 48-45 ka BP and after HE5/GS13. After GS12 (44.2-43.3 ka BP), horses and bovines are potentially occupying different ecological niches during the Châtelperronian and early Aurignacian levels of Labeko Koba, pointing to a species' environmental specialisation, which can be a consequence of competition for food resources during an unstable ecological period. The climatic estimations indicate a temperature shift during this period, with a slight decrease in temperatures and evidence of fluctuations in rainfall. Previous environmental studies on the region have underlined ecological stress and increasing aridity from around 42.5 ka cal BP, which may relate to a broader ecosystem decline. When comparing the environmental conditions during the Aurignacian period in the northeast (Canyars) and the northwest (Labeko Koba), the first had higher baseline temperatures but also experienced higher aridity. Animals continued to feed on open landscapes during the Gravettian and Magdalenian levels in the Vasco-

Cantabrian region, represented by Aitzbitarte III interior and El Otero. However, there is evidence of a
temperature recovery after the LGM at the El Otero.
The results presented here, derived from the first extensive sampling in the Vasco-Cantabrian, establish the
basis of future stable isotopic studies on faunal tooth enamel in Iberia. Despite the uncertainties inherent in
this work, both $\delta^{18}O$ and $\delta^{13}C$ contributed to the regional climatic characterisation, including the estimation
of temperatures and precipitations, as well as the seasonality range between summer and winter. The
potential influence of pretreatment effects and uncontrolled diagenetic alterations on the enamel carbonate
fraction has been assessed. However, complementary diagenetic tests, using new techniques like $\delta^{18}O_{phos}$
and FTIR analyses are advised in further works to gain more insights into sample preservation. Ongoing
sulphur, hydrogen and strontium studies will provide additional information on the mobility patterns of
animals hunted by Late Pleistocene hominins and, therefore, will help better understand the ecological and
environmental context occupied by Neanderthal and modern humans and their landscape use in this
particular region. Finally, a more comprehensive characterisation of the baseline oxygen values would also
enhance the environmental interpretation of the existing data.
**Appendices**
Appendices A, C, D and E are presented after bibliography. Raw data is presented in Appendix B, available
at https://github.com/ERC-Subsilience/Ungulate_enamel-carbonate
**Code availability**
R code used to perform plots, temperature and error calculations, Bayesian models code and inverse
models in this manuscript can be accessed at GitHub (https://github.com/ERC-
Subsilience/Ungulate_enamel-carbonate).
**Data availability**
The available datasets used for this article are provided in the supplementary materials (Appendix A-E).
**Author contribution**
A.B.M.-A. got the funding and designed the research. A.B.M.-A and M.F.-G. get the permissions for sampling
in the regional museums. M.F.-G., K.B, and S.P. defined the analysis strategy. M.F.-G. analysed the data
and wrote the manuscript with critical inputs from A.B.M.-A., K.B, and S.P. J.M.G., L.A., M.F.-G., and A.C.
M.F.-G., L.A., J.M.G., and A.C. achieved the teeth sampling and lab sample preparation. J.D. and M.S. are
responsible for the excavations in Canyars and contribute to the discussion. All the authors revised and
commented on the manuscript.
**Competing interests**
The contact author has declared that none of the authors has any competing interests.
**Acknowledgements**
We acknowledge the Museo de Arqueología y Prehistoria de Cantabria (MUPAC), the Consejería de
Educación, Cultura y Deporte del Gobierno de Cantabria, the Museo de Arqueología de Bizkaia (Arkeologi
Museoa) and the Centro de Colecciones Patrimoniales de la Diputación Foral de Gipuzkoa (Gordailua) –
Provincial Government of Guipuzkoa's Heritage Collection Centre for the access to the archaeological
collections. We do appreciate the work achieved by H. Reade during the initial sampling, pretreatment and
analyses of samples undertaken at the University of Cantabria and Cambridge. We want to thank the two
anonymous referees for their valuable comments, which significantly improved the quality of the paper.
**Financial support**
Funding for Vasco-Cantabria research was obtained from the Spanish Ministry of Science and Innovation
(PID2021-125818NB-I00, HAR2017-84997-P and HAR2012-33956), the European Research Council under
the European Union's Horizon 2020 Research and Innovation Programme (grant agreement number
818299; SUBSILIENCE project) and Proyecto Puente by Consejería de Educación, Cultura y Deporte del
Gobierno de Cantabria. Research for Canyars was funded by the Spanish Ministry of Science and
Innovation (PID2020-113960GB-100), Departament de Cultura de la Generalitat de Catalunya
(CLT/2022/ARQ001SOLC/128) and AGAUR (SGR2021-00337). M.F.-G. is supported by the APOSTD
postdoctoral fellowship (CIAPOS/2022/081/AEI/10.13039/501100011033), funded by the Generalitat
Valenciana and the European Social Fund. S.P. was supported by a German Academy of Sciences
Leopoldina postdoctoral fellowship (LPDS 2021-13) during this project. M.S. benefited from financial support
from a Ramon y Cajal postdoctoral grant (RYC2021-032999-I) funded by the Spanish Ministry of Science
and Innovation and the European Union-NextGenerationEU.

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

'"""29iomin29iomineralization

**Appendix A. Sites description**

**A1. Vasco-Cantabrian sites**

**Axlor (Dima, Vizcaya, País Vasco)**

Axlor is a rock-shelter located in Dima (43.2706; -1.8905), with a continuous Middle Paleolithic sequence from the MIS5 to the MIS3 (DeMuro et al., 2023; Pederzani et al., 2023; Marín-Arroyo et al., 2018). It is placed on the southwestern slope of the Dima Valley, with an elevation of approximately 320 m above sea level (a.s.l.), at 33 km straight from the present-day coastline, next to one of the lowest mountain passes linking the Cantabrian basins and the Alavese Plateau. The site was discovered in 1932 and initial excavations were performed by Barandiarán (1967-1974). J. M. Barandiarán undertook the excavations between 1967 and 1974, identifying eight Mousterian levels (I-VIII) (Barandiarán, 1980).

From 2000 to 2008, new excavations by González-Urquijo, Ibáñez-Estévez and Rios-Garaizar were achieved and, since 2019, these are ongoing by González-Urquijo and Lazuén. Due to the lack of chronology during Barandiarán excavations, among other aspects, work was focused on obtaining a detailed stratigraphy on the new excavation areas to correlate it with Barandiarán's levels (González-Urquijo & Ibáñez-Estévez, 2021; González Urquijo et al., 2005). The new stratigraphic sequence is roughly equivalent to the previous one, but with additional levels not previously identified or excavated by Barandiarán. Some of these levels were deposited before Level VIII (Gómez-Olivencia et al., 2018; 2020). The Middle Paleolithic sequence extends from layers VIII to III (or from N to B-C). Levallois production is predominant in the lower levels (VI to VIII), while Quina Mousterian technocomplex does in the upper ones (from III to V) (Rios-Garaizar, 2012, 2017). Recent chronological data by radiocarbon (Pederzani et al., 2023; Marín-Arroyo et al., 2018) and OSL (Demuro et al., 2023) methods confirm that a sequence Axlor levels VI, VIII, and VIII probably accumulated during MIS5d–a (109–82 ka), while levels D to B probably were formed during the period encompassing the start of MIS 4 (71–57 ka) through to the beginning or middle of MIS 3 (57–29 ka) and upper Level III to 46,200 ±3,000 BP, which calibrates between 45,350 cal BP and beyond the calibration curve at > 55,000 cal BP.

The archaeozoological study indicates an anthropic origin of the faunal assemblage with scarce carnivore activity documented (Altuna, 1989; Castaños, 2005; Gómez-Olivencia et al., 2018). In lower layers, the most abundant taxa are *Cervus elaphus* (VIII) and *Capra pyrenaica* (VII), while in upper layers III-V, C*ervus elaphus* is substituted by *Bos primigenious/Bison priscus* and *Equus sp*. The material included in this work comes from the faunal collection of the Barandiarán excavation currently curated at the Bizkaia Museum of Archaeology (Bilbao), where teeth were sampled, and the stable isotope analyses on enamel phosphate were included in Pederzani et al. (2023).

**El Castillo (Puente Viesgo, Cantabria)**

El Castillo cave is located in Puente Viesgo (43.2924; -3.9656), with an elevation of approximately 195m a.s.l., at 17 km straight from the present-day coastline. The cave belongs to the karstic system that was formed in the Monte Castillo, which dominates the Pas Valley. The site was discovered in 1903 by H. Alcalde del Río. H. Obermaier carried out the first excavation seasons between 1910 and 1914 when many of the archaeological remains were recovered, mainly from the cave hall. These interventions were done under the supervision of the "Institut de Paléontologie Humaine" (IPH) and Prince Albert I of Monaco. From 1980 to 2011, V. Cabrera and F. Bernaldo de Quirós underwent new excavations focusing on the cave entrance, on the Middle to Upper Paleolithic transitional levels, mainly 16, 18 and 20 (Cabrera-Valdes, 1984). The site has yielded an important stratigraphic sequence, composed by 26 sedimentological units (1-26) related to

different anthropic occupational units, often separated by archaeologically sterile units: Eneolithic (2), Azilian (4), Magdalenian (6 and 8), Solutrean (10), Aurignacian (12, 14, 16 and 18), Mousterian (20, 21 and 22) and Acheulean (24) (Cabrera-Valdés, 1984).

Unit 21 is mostly sterile (Cabrera Valdés, 1984; Martín-Perea et al., 2023), and ESR dated it, yielding a mean date of 69,000 ± 9,200 years BP (Rink et al., 1997). However, Martín-Perea et al. (2023) suggested some dating uncertainty from interpreting the initial stratigraphic nomenclature. They suggest that the ESR dates provided for level 21 by Rink et al. (1997) were erroneously attributed to this unit and it might correspond to 20E, indicating that below that subunit, the chronology is older than 70,000 years BP (Martín-Perea et al., 2023). The Mousterian Unit 20 cave is divided into several subunits (Martín-Perea et al., 2023). In Unit 20, a cave roof collapse took place, transforming the cave system into an open rock shelter. This unit contains abundant archaeological and paleontological remains. Lithic industry consists of sidescrapers, denticulates, notches and cleavers, the majority on quartzite and presents both unifacial, bifacial discoid debitage and Levallois debitage. Unit 20E was attributed to Quina Mousterian by Sánchez-Fernández and Bernaldo De Quiros (2009) and contains a Neanderthal tooth (Garralda, 2005). Considering the geochronological uncertainties for dates on 20E related to Rink et al. (1997), we have decided to rely solely on ESR date of 47,000 ± 9400 BP provided by Liberda et al. (2010) for this level. Unit 20C presents clear evidence of the Mousterian lithic industry and radiocarbon dates of 48,700±3,400 uncal BP (OxA-22204) and 49,400±3,700 uncal BP (OxA-22205) (Wood et al., 2018) and mean ESR date of 42,700 ±9900 BP (Liberda et al., 2010). Level 19 is archaeologically sterile and separates Unit 20 from Unit 18 (Wood et al., 2018).

Unit 18 is divided into 18A (archaeologically sterile), 18B, and 18C. Levels 18B and 18C were classified as Transitional Aurignacian, representing a gradual transformation from the Mousterian to the Aurignacian, which is unique to El Castillo cave (Cabrera et al., 2001; Maíllo and Bernaldo de Quirós, 2010; Wood et al., 2018). These levels' dates and cultural attribution have been the subject of much debate (e.g. Zilhao and D'Errico, 2003; Wood et al., 2018). According to Wood et al. (2018), the last dates of these levels range between 42,000±1,500 uncal BP (OxA-22203) and 46,000±2,400 uncal BP (OxA-21973), which is much earlier than the start of the Aurignacian period in the Cantabrian region (Marín-Arroyo et al., 2018; Vidal-Cordasco et al., 2022). The lithic assemblage of Unit 18 appears to be dominated by Discoid/Levallois technology (Bernaldo de Quirós and Maíllo-Fernández, 2009) but with a high percentage of "Upper Paleolithic" pieces. Additionally, punctual bone industry and pieces with incisions and engravings were discovered in Unit 18 (Cabrera-Valdés et al., 2001). Three deciduous tooth crowns attributed to Neanderthals were found in Unit 18B (Garralda et al., 2022). Above, Unit 17 is sterile but contains scarce lithic and faunal materials, while Level 16 was attributed to the Proto-Aurignacian, with dates of 38,600±1,000 uncal BP (OxA-22200) (Wood et al., 2018).

According to Luret et al. (2020), there was a shift in hunting practices between the Late Mousterian (unit 20) and the Transitional Aurignacian (unit 18). During the Late Mousterian, hunting strategies were less specialized, and the species hunted included red deer, horses, and bovines. However, in Unit 18, a specialization in red deer hunting is observed. However, the explanation of this shift has been proposed as a response to a cultural choice or induced by climatic changes. However, recent taphonomic studies by Sanz-Royo et al. (2023) on the old collections of Aurignacian Delta level reveal a more significant role of carnivores than shown by Luret et al. (2020). The material included in this work comes from the faunal collection recovered during the Cabrera-Valdés and Bernaldo de Quirós excavations curated at Museo de Prehistoria y Arqueología de Cantabria (MUPAC, Santander).

**Labeko Koba (Arrastre, Guipúzcoa, País Vasco)**

Labeko Koba is a cave in the Kurtzetxiki Hill (43.0619; -2.4833), at 246 m a.s.l. and 29 km straight from the present-day Atlantic coast. In 1987 and 1988, the site was discovered due to the construction of the Arrasate ring road, and a savage excavation was carried out (Arrizabalaga, 2000a). Unfortunately, the site was destroyed after that. The stratigraphic sequence identified nine different levels. The lower Level IX was attributed to the Châtelperronian, based on the presence of three Châtelperron points. Although there is a lack of human remains in few Cantabrian Châtelperronian sites, recent research has suggested that this techno-complex was produced by Neanderthals (Maroto et al., 2012; Rios-Garaizar et al., 2022). Level VII marks the beginning of the Aurignacian sequence, likely Proto-Aurignacian, with a lithic assemblage dominated by Dufour bladelets (Arrizabalaga, 2000a). Levels VI, V, and IV contain lithic assemblages that suggested an Early Aurignacian attribution (Arrizabalaga, 2000b; Arrizabalaga et al., 2009). This site is significant because it is one of the few sites with Châtelperronian assemblages and with both Proto-Aurignacian and Early Aurignacian separated (Arrizabalaga et al., 2009).

Initial radiocarbon dates were inconsistent with the stratigraphy of the site and much more recent than expected for the Early Upper Paleolithic (Arrizabalaga, 2000a). This incoherence was determined to be affected by taphonomic alterations (Wood et al., 2014). Later radiocarbon dates undertaken with an ultrafiltration pre-treatment provided a new regional framework for the regional Early Upper Paleolithic (Wood et al., 2014). The Châtelperronian layer IX inf is dated to 38,100±900 uncal BP (OxA-22562) and 37,400±800 uncal BP (OxA-22560). The Proto-Aurignacian levels cover a period from 36,850±800 uncal BP (OxA-21766) to 35,250±650 uncal BP (OxA-21793). The three Early Aurignacian levels are dated to 35,100±600 uncal BP (OxA-21778) for level VI, ~ 34,000 uncal BP (OxA-21767 and OxA-21779) for level V, and ~ 33,000 BP (OxA-21768 and OxA-21780) for level IV (Arrizabalaga et al., 2009).

Taphonomic studies indicate an alternation in the use of the cave between carnivores and humans, the latter during short occupation periods (Villaluenda et al., 2012; Ríos-Garaizar et al., 2012; Arrizabalaga et al., 2010). Labeko Koba is considered to have functioned as a natural trap where carnivores, mainly hyenas, access animal carcasses. At least in the base of Labeko Koba IX, carnivore activity was higher, and they would have consumed the same prey as humans (Villaluenga et al., 2012). The presence of humans is linked to strategic use as a campsite associated with a small assemblage of lithic artifacts. The most consumed species by Châtelperronian groups were red deer, followed by the consumption of large bovids, equids, and woolly rhinoceros. During the Aurignacian period, there was some stability in human occupations, although they still alternated with carnivore occupations (Arrizabalaga et al., 2010). Cold-adapted fauna such as reindeer and woolly rhinoceros were identified in association with the Châtelperronian. Reindeer and the woolly mammoth and arctic fox were still present during the Aurignacian levels. The original sampling of the teeth studied by this work was performed in the San Sebastian Heritage Collection headquarters, where the Guipuzcoa archaeological materials were deposited at that time.

**Aitzbitarte III interior (Rentería, Guipúzcoa, País Vasco)**

Aitzbitarte III is an archaeological site located within the Landarbaso karstic system comprising nine caves (43.270; -1.8905). The cave is situated 220 m.a.s.l. and is 10 km away from the present-day coastline. Initial archaeological interventions were carried out at the end of the 19th century by P.M. de Soraluce (Altuna, 2011). Recent excavations were initially conducted in the deep zone inside the cave between 1986 and 1993, where the studied tooth was recovered, and later focused on the cave entrance between 1994 and 2002, by J. Altuna, K. Mariezkurrena, and J. Ríos-Garaizar (Altuna et al., 2011; 2017).

While the cave's entrance area contains a sequence comprising possible Mousterian and Evolved Aurignacian and Gravettian levels (Altuna et al., 2011; 2013), the stratigraphy in the inner cave presents eight levels: level VIII (some tools with Mousterian features), VII (sterile), VIb, VIa and V (Middle Gravettian

technocomplex with abundance of Noailles burins), IV-II (disturbed archaeological levels) and I (surface)
(Altuna et al., 2017). Levels V have dates of 24,910 uncal BP (I-15208) and 23,230 uncal BP (Ua-2243);
whereas level VI extends from 23,830 ± 345 uncal BP (Ua-2628) and 25,380± 430 uncal BP (Ua-2244)
(Altuna, 1992; Altuna et al., 2017), with a possible outlier dated at 21,130 uncal BP (Ua-1917).
The Gravettian occupation in the inner part of the cave was initially thought to be more recent than the one
in the cave entrance. However, it was not easy to correlate the two excavation areas due to different
sedimentation rates. The abundant human occupations took place during a singular cold phase in the Middle
Gravettian with a specialized paleoeconomy focused on the hunting of *Bos primigenius* and *Bison priscus*
(85% in level VI and 68% in level V), which is unusual in the Cantabrian region mostly focused on red deer
and ibex. Other ungulates present are *Cervus elaphus* and *Rupicapra rupicapra*, and to a lesser extent
*Capra pyrenaica, Capreolus capreolus, Rangifer tarandus*, and *Equus ferus* (Altuna et al., 2017; Altuna &
Mariezkurrena, 2020). There is a scarce representation of carnivores. The tooth studied was sampled at the
Gordailua Center for Heritage Collections of the Provincial Council of Gipuzkoa.

**El Otero (Secadura, Voto, Cantabria)**
El Otero cave is located in Secadura (Voto) (43.3565; -3.5360), at 129 m.s.a.l and 12 km from the present-
day coastline, near the Matienzo valley in a coastal plain environment covered by meadows and gentle hills.
The discovery was made in 1908 by Lorenzo Sierra. The site was excavated in 1963 by J. Gonzalez
Echegaray and M.A. García Guinea, in two different sectors (Sala I and Sala II) with an equivalent
stratigraphic sequence (González Echegaray, 1966). Nine levels were identified in Sala I, from level IX to
level I. Levels IX and VIII were initially related to the "Aurignacian-Mousterian, based on lithics assemblages
with a combination of both technocomplex features. The overlying levels VI-IV were separated by a
speleothem crust (level VII) and were initially related to Aurignacian, due to the presence of end-scrappers,
bone points, blades, or burins on truncation (Freeman, 1964; Rios-Garaizar, 2013). Also, perforated deer,
ibex, and fox teeth were found in levels V and IV. This site lacked chronological dating methods, until a
selection of material from levels VI, V and IV revealed a difference in chrono-cultural attribution (Marín-
Arroyo et al., 2018). Radiocarbon results yielded younger dates for such a cultural attribution and showed
significant stratigraphic inconsistency. Level VI gave a result of 12,415±55 uncal BP (OxA-32585), two dates
in Level V are 12,340±55 (OxA-32509) and 10,585±50 uncal BP (OxA-32510), and a date in Level IV is
15,990±80 uncal BP (OxA-32508). All these results fall into the range of the Late Upper Paleolithic
(Magdalenian-Azilian initially identified in levels III-I), eliminating attribution of these levels to the Aurignacian
despite the presence of apparently characteristic artefacts. Further assessments of archaeological materials
will be needed.
Red deer dominate the assemblage, except for level IV where horses are more abundant. Wild boar, roe
deer, and ibex are also present, but large bovids are relatively rare (González Echegaray, 1966). Level IV
is the richest and most anthropogenic level, with evidence of butchering in red deer (captured in winter and
early summer) and chamois (in autumn). The formation of this level involved humans and carnivores, and
although certain data may suggest an anthropogenic predominance, the limited sample analyzed
taphonomically and the pre-selection of preserved pieces do not allow for a definitive conclusion (Yravedra
& Gómez-Castanedo, 2010). The material included in this work is curated at the Museo de Prehistoria y
Arqueología de Cantabria (MUPAC, Santander).

**A2. Northeastern Iberia sites**
**Terrasses de la Riera dels Canyars (Gavà, Barcelona, Cataluña)**
Terrasses de la Riera dels Canyars (henceforth, Canyars) is an open-air site located near Gavà (Barcelona)
(41.2961;1.9797), at 28 m.s.a.l and 3 km straight from the present-day coastline. The site lies on a fluvial
terrace at the confluence of Riera dels Canyars, a torrential stream between Garraf Massif, Llobregat delta
and Riera de Can Llong (Daura et al., 2013). Archaeo-paleontological remains were discovered during
quarries activities in 2005 and was complete excavated on 2007 by the *Grup de Recerca del Quaternari*
(Daura and Sanz, 2006; Daura et al., 2013). This intervention determined nine lithological units. The
paleontological and archaeological remains come exclusively from one unit, the middle luthitic unit (MLU),
and specifically from layer I. The MLU is composed of coarse sandy clays and gravels, filling a paleochannel
network named lower detrital unit (LDU) (Daura et al., 2013). Five radiocarbon dates were obtained on
charcoals from layer I, which yield statistically consistent ages from 33,800 ±350 uncal BP to 34,900 ±340
uncal BP, which results in mean age of 39,710 cal BP (from 40,890 to 38,530 cal BP) (Daura et al., 2013;
this work).
The layer I of the site has yielded a rich faunal assemblage, consisting of over 5,000 remains. Among the
herbivores, the most common species found are *Equus ferus*, *Bos primigenius, Equus hydruntinus*, and
*Cervus elaphus* (Daura et al., 2013; Sanz-Royo et al., 2020). *Capra* sp. and *Sus scrofa are* also present,
although in lower frequencies. The carnivores found at the site are also noteworthy, with *Crocuta crocuta*
and *Lynx pardinus* being the most frequent. Presence of cold-adapted fauna associated to stepped
environments is recorded, such as cf. *Mammuthus* sp., *Coelodonta antiquitatis*, and *Equus hydruntinus*.
Small mammal analysis, pollen, and use-wear analysis have provided further evidence that a steppe-
dominated landscape surrounded the Canyars site, supporting a correlation with the Heinrich Stadial 4, in
coherence with the chronology obtained for the layer (López-García et al. 2013; 2023; Rivals et al., 2017).
However, the presence of woodland is also attested by forest taxa within charcoal and pollen assemblages
(Daura et al., 2013).
Taphonomic study is ongoing. But several evidences point that hyenas have played an important role in the
accumulation of the faunal assemblage (Daura et al., 2013; Jimenez et al. 2019). However, sporadic human
presence is documented by few human modifications found in faunal remains (cutmarks and fire alterations).
Although the paucity of the lithic assemblage in the site, it shows a clear attribution to Upper Palaeolithic
technocomplex, most likely the Early Aurignacian (Daura et al., 2013). Recently, it was documented a
perforated bone fragment, which has been identified as a perforated board for leather production (Doyon et
al., 2023). All teeth included in this work were sampled in *Laboratori de la Guixera* (Ajuntament de
Casteldefels) where the material is stored.

**References Appendix A**
Altuna, J., Mariezkurrena, K., de la Peña, P., Rios-Garaizar, J. 2011. Ocupaciones Humanas En La Cueva de Aitzbitarte III (Renteria,
País Vasco) Sector Entrada: 33.000-18.000 BP. Servicio Central de Publicaciones del Gobierno Vasco; EKOB: 11–21.
Altuna, J., Mariezkurrena, K., de la Peña, P., Rios-Garaizar, J. 2013. Los niveles gravetienses de la cueva de Aitzbitarte III
(Gipuzkoa). Industrias y faunas asociadas, in: de las Heras, C., Lasheras, J.A., Arrizabalaga, Á., de la Rasilla, M. editors.
Pensando El Gravetiense: Nuevos Datos Para La Región Cantábrica En Su Contexto Peninsular Y Pirenaico.
Monografías Del Museo Nacional Y Centro de Investigación de Altamira, 23. Madrid: Ministerio de Educación, Cultura;
pp. 184–204.
Altuna, J. & Mariezkurrena, K. 2020. Estrategias de caza en el Paleolítico superior de la Región Cantábrica. El caso de Aitzbitarte
II (zona profunda de la cueva). Sagvntvm-Extra 21, Homenaje al Profesor Manuel Pérez Ripoll: 219-225.
Altuna, J., Mariezkurrena, K., Ríos Garaizar, J., & San Emeterio Gómez, A. 2017. Ocupaciones Humanas en Aitzbitarte III (País
Vasco) 26.000 - 13.000 BP (zona profunda de la cueva). Servicio Central de Publicaciones del Gobierno Vasco. EKOB;
8: 348pp.
Arrizabalaga, A., 2000a. El yacimiento arqueológico de Labeko Koba (Arrasate, País Vasco). Entorno. Crónica de las
investigaciones. Estratigrafía y estructuras. Cronología absoluta. In: Arrizabalaga, A., Altuna, J. (Eds.), Labeko Koba
(País Vasco). Hienas y Humanos en los Albores del Paleolítico Superior, Munibe (Antropologia-Arkeologia) 52. Sociedad
de Ciencias Aranzadi, San Sebastián-Donostia, pp. 15-72.
Arrizabalaga, A., 2000b. Los tecnocomplejos líticos del yacimiento arqueológico de Labeko Koba (Arrasate, País Vasco). In:
Arrizabalaga, A., Altuna, J. (Eds.), Labeko Koba (País Vasco). Hienas y Humanos en los Albores del Paleolítico Superior,
Munibe (Antropologia-Arkeologia) 52. Sociedad de Ciencias Aranzadi, San Sebastián-Donostia, pp. 193-343.
Arrizabalaga, A., Iriarte, E., Ríos-Garaizar, J., 2009. The Early Aurignacian in the Basque Country. Quaternary International, 207:
25–36.
Arrizabalaga, A., Iriarte, M.J. & Villaluenga, A. 2010. Labeko Koba y Lezetxiki (País Vasco). Dos yacimientos, una problemática
común. Zona Arqueológica, 13: 322-334.
Barandiarán JM. 1980. Excavaciones en Axlor. 1967- 1974. En: Barandiarán, J. M.: Obras Completas. Tomo XVII; pp. 127-384.
Bernaldo de Quirós, F., Maíllo-Fernández, J.-M. 2009. Middle to Upper Palaeolithic at Cantabrian Spain. In: Camps M, Chauhan
PR (eds) A sourcebook of Palaeolithic transitions: methods, theories and interpretations. Springer, New York, pp. 341–
1446     359.
Cabrera-Valdes, V. 1984. El Yacimiento de la cueva de «El Castillo» (Puente Viesgo, Santander). Bibliotheca Praehistorica Hispana
22, C.S.I.C., 485 p.
Cabrera-Valdes, V., Maillo-Fernandez, J.M., Lloret, M., Bernaldo De Quiros, F. 2001. La transition vers le Paléolithique supérieur
dans la grotte du Castillo (Cantabrie, Espagne) la couche 18. L'Anthropologie 105, pp. 505–532.
Daura, J., Sanz, M. (2006). Informe de la troballa del jaciment arqueològic "Terrasses dels Canyars" (Castelldefels-Gavà).
Notificació de la descoberta i propostes d'actuació. Grup de Recerca del Quaternari, SERP, UB. Servei d'Arqueologia i
Paleontologia, Departament de Cultura i Mitjans de Comunicació, Generalitat de Catalunya. Unpublished Archaeological
Report.
Daura, J., Sanz, M., García, N., Allué, E., Vaquero, M., Fierro, E., Carrión, J. S., López-García, J. M., Blain, H. A., Sánchez-Marco,
A., Valls, C., Albert, R. M., Fornós, J. J., Julià, R., Fullola, J. M., Zilhão, J. 2013. Terrasses de la Riera dels Canyars
(Gavà, Barcelona): The landscape of Heinrich stadial 4 north of the "Ebro frontier" and implications for modern human
dispersal into Iberia. Quaternary Science Reviews, 60, 26–48.
Demuro, M., Arnold, L., González-Urquijo, J., Lazuen, T., Frochoso, M. 2023. Chronological constraint of Neanderthal cultural and
environmental changes in southwestern Europe: MIS 5–MIS 3 dating of the Axlor site (Biscay, Spain). Journal of
Quaternary Research
Doyon, L., Faure, T., Sanz, M., Daura, J., Cassard, L., D'Errico, F., 2023. A 39,600-year-old leather punch board from Canyars,
Gavà, Spain. Scientific Advances, 9. https://doi.org/10.1126/sciadv.adg0834
Freeman, L.G. 1964. Mousterian Developments in Cantabrian Spain. Ph.D. thesis. Dept. of Anthropology, University of Chicago,
Chicago.
Garralda, M.D. 2005. Los Neandertales en la Península Ibérica:The Neandertals from the Iberian Peninsula. Munibe (Antropologia-
Arkeologia) 57, Homenaje a Jesús Altuna. pp. 289–314.
Garralda, M.D., Madrigal, T., Zapata, J., & Rosell, J. 2022. Neanderthal deciduous tooth crowns from the Early Upper Paleolithic at
El Castillo Cave (Cantabria, Spain). Archaeological and Anthropological Sciences.
Gómez-Olivencia, A., Arceredillo, D., Álvarez-Lao, D.J., Garate, D., San Pedro, Z., Castaños, P., Rios-Garaizar, J., 2014. New
evidence for the presence of reindeer (Rangifer tarandus) on the Iberian Peninsula in the Pleistocene: an
archaeopalaeontological and chronological reassessment. Boreas 43, 286–308.
Gómez-Olivencia, A., Sala, N., Núñez-Lahuerta, C., Sanchis, A., Arlegi, M., Rios-Garaizar, J., 2018. First data of Neandertal bird
and carnivore exploitation in the Cantabrian Region (Axlor; Barandiaran excavations; Dima, Biscay, Northern Iberian
Peninsula). Scienti. Rep. 8, 10551.
González Echegaray, J.G. 1966. Cueva del Otero. Excavaciones Arqueológicas en España, 53. Madrid: Ministerio de Educación
Nacional Dirección General de Bellas Artes Servicio Nacional de Excavaciones.
González-Urquijo, J.E., Ibáñez-Estévez, J.J. 2001. Abrigo de Axlor (Dima). Arkeoikuska: Investigación arqueológica 2001; 2002:
90–93.
González Urquijo, J.E., Ibáñez Estévez, J.J., Rios-Garaizar, J., Bourguignon, L., Castaños Ugarte, P., Tarriño Vinagre, A. 2005.
Excavaciones recientes en Axlor. Movilidad y planificación de actividades en grupos de neandertales. In: Montes Barquín
R, Lasheras Corruchaga JA, editors. Actas de La Reunión Científica: Neandertales Cantábricos. Estado de La Cuestión.
Monografías Del Museo Nacional Y Centro de Investigación de Altamira No 20. Madrid: Ministerio de Cultura; 2005. pp.
527–539.
Jimenez, I. J., Sanz, M., Daura, J., Gaspar, I. D., García, N. 2019. Ontogenetic dental patterns in Pleistocene hyenas (Crocuta
crocuta Erxleben, 1777) and their palaeobiological implications. International Journal of Osteoarchaeology, 29, 808–821.
Liberda, J.J., Thompson, J.W., Rink, W.J., Bernaldo de Quirós, F., Jayaraman, R., Selvaretinam, K., Chancellor-Maddison, K.,
Volterra, V., 2010. ESR dating of tooth enamel in Mousterian layer 20, El Castillo, Spain. Geoarchaeology n/a-n/a.
López-García, J.M., Blain, H.A., Fagoaga, A., Bandera, C.S., Sanz, M., Daura, J., 2022. Environment and climate during the
Neanderthal-AMH presence in the Garraf Massif mountain range (northeastern Iberia) from the late Middle Pleistocene
to Late Pleistocene inferred from small-vertebrate assemblages. Quaternary Science Reviews, 288.
López-García, J. M., Blain, H.-A., Bennàsar, M., Sanz, M., Daura, J. 2013. Heinrich event 4 characterized by terrestrial proxies in
southwestern Europe. Climate of the Past, 9: 1053–1064.
Luret, M., Blasco, R., Arsuaga, J.L., Baquedano, E., Pérez-González, A., Sala, N., & Aranburu, A. 2020. A multi-proxy approach to
the chronology of the earliest Aurignacian at the El Castillo Cave (Spain). Journal of Archaeological Science: Reports,
1496 33: 102339.
Maroto, J., Vaquero, M., Arrizabalaga, Á., Baena, J., Baquedano, E., Jordá, J., Julià, R., Montes, R., Van Der Plicht, J., Rasines,
P., Wood, R., 2012. Current issues in late Middle Palaeolithic chronology: New assessments from Northern Iberia.
Quaternary International, 247: 15–25.
Marín-Arroyo, A.B., Rios-Garaizar, J., Straus, L.G., Jones, J.R., de la Rasilla, M., González Morales, M.R., Richards, M., Altuna, J.,
Mariezkurrena, K., Ocio, D., 2018. Chronological reassessment of the Middle to Upper Paleolithic transition and Early
Upper Paleolithic cultures in Cantabrian Spain. PLoS One 13: 1–20.
Martín-Perea, D.M., Maíllo-Fernández, J., Marín, J., Arroyo, X., Asiaín, R., 2023. A step back to move forward: a geological re-
evaluation of the El Castillo Cave Middle Palaeolithic lithostratigraphic units (Cantabria, northern Iberia). Journal of
Quaternary Science, 38: 221–234.
Pederzani, S., Britton, K., Jones, J.R., Agudo Pérez, L., Geiling, J.M., Marín-Arroyo, A.B., 2023. Late Pleistocene Neanderthal
exploitation of stable and mosaic ecosystems in northern Iberia shown by multi-isotope evidence. Quaternary Research:
1–25.
Rink, W.J., Schwarcz, H.P., Lee, H.K., Cabrera Valdés, V., Bernaldo de Quirós, F., Hoyos, M. 1997. ESR dating of Mousterian
levels at El Castillo Cave, Cantabria, Spain. Journal of Archaeological Science, 24 (7): 593-600.
Rios-Garaizar J. 2012.Industria lítica y sociedad en la Transición del Paleolítico Medio al Superior en torno al Golfo de Bizkaia.
Santander: PUbliCan - Ediciones de la Universidad de Cantabria.
Rios-Garaizar, J. 2017. A new chronological and technological synthesis for Late Middle Paleolithic of the Eastern Cantabrian
Region. Quaternary International, 433: 50-63.
Rios-Garaizar, J., Arrizabalaga, A. & Villaluenga, A. 2012. Haltes de chasse du Châtelperronien de la Péninsule Ibérique: Labeko
Koba et Ekain (Pays Basque Péninsulaire). L'Anthropologie, 116: 532–549.
Rios-Garaizar, J., de la Peña, P., Maillo-Fernández, J.M. 2013. El final del Auriñaciense y el comienzo del Gravetiense en la región
cantábrica: una visión tecno-tipológica. In: de las Heras C., Lasheras J.A., Arrizabalaga Á., de la Rasilla M. (Eds.),
Pensando El Gravetiense: Nuevos Datos Para La Región Cantábrica En Su Contexto Peninsular Y Pirenaico.
Monografías Del Museo Nacional Y Centro de Investigación de Altamira, 23. Madrid: Ministerio de Educación, Cultura;
pp. 369–382.
Rios-Garaizar, J., Iriarte, E., Arnold, L.J., Sánchez-Romero, L., Marín-Arroyo, A.B., San Emeterio, A., Gómez-Olivencia, A., Pérez-
Garrido, C., Demuro, M., Campaña, I., Bourguignon, L., Benito-Calvo, A., Iriarte, M.J., Aranburu, A., Arranz-Otaegi, A.,
Garate, D., Silva-Gago, M., Lahaye, C., Ortega, I. 2022. The intrusive nature of the Châtelperronian in the Iberian
Peninsula. PLoS One 17, e0265219.
Rivals, F., Uzunidis, A., Sanz, M., Daura, J., 2017. Faunal dietary response to the Heinrich Event 4 in southwestern Europe.
Palaeogeogr. Palaeoclimatol. Palaeoecol. 473, 123–130.
Sanz-Royo, A., Sanz, M., Daura, J. (2020). Upper Pleistocene equids from Terrasses de la Riera dels Canyars (NE Iberian
Peninsula): The presence of Equus ferus and Equus hydruntinus based on dental criteria and their implications for
palaeontological identification and palaeoenvironmental reconstruction. Quaternary International, 566–567, 78–90.
Sanz-Royo, A., Terlato, G., Marín-Arroyo, A.B., 2024. Taphonomic data from the transitional Aurignacian of El Castillo cave (Spain)
reveals the role of carnivores at the Aurignacian Delta level. Quaternary Science Advances, 13: 100147.
https://doi.org/10.1016/j.qsa.2023.100147
Vidal-Cordasco, M., Ocio, D., Hickler, T., Marín-Arroyo, A.B., 2022. Ecosystem productivity affected the spatiotemporal
disappearance of Neanderthals in Iberia. Nat. Ecol. Evol. 6, 1644–1657.
Villaluenga, A., Arrizabalaga, A. & Rios-Garaizar, J. 2012. Multidisciplinary approach to two Châtelperronian series: lower IX layer
of Labeko Koba and X Level of Ekain (Basque country, Spain). Journal of Taphonomy, 10: 525–548.
Wood, R.E., Arrizabalaga, A., Camps, M., Fallon, S., Iriarte-Chiapusso, M.J., Jones, R., Maroto, J., De la Rasilla, M., Santamaría,
D., Soler, J., Soler, N., Villaluenga, A., Higham, T.F.G. 2014. The chronology of the earliest Upper Palaeolithic in northern
Iberia: New insights from L'Arbreda, Labeko Koba and La Viña. Journl of Human Evolution, 69: 91–109.
https://doi.org/10.1016/j.jhevol.2013.12.017
Wood, R., Bernaldo de Quirós, F., Maíllo-Fernández, J.M., Tejero, J.M., Neira, A., Higham, T. 2018. El Castillo (Cantabria, northern
Iberia) and the Transitional Aurignacian: Using radiocarbon dating to assess site taphonomy. Quaternary International,
474: 56–70.
Yravedra, J., & Gómez-Castanedo, A. 2010. Estudio zooarqueológico y tafonómico del yacimiento del Otero (Secadura, Voto,
Cantabria). Espacio, Tiempo y Forma. Serie I, Nueva época. Prehistoria y Arqueología, 3: 21-38
Zilhao, J., DEerrico, F. 2003 The chronology of the Aurignacian and Transitional technocomplexes. Where do we stand? In Zilhão,
J. et d'Errico, F. eds., The chronology of the Aurignacian and of the transitional technocomplexes Dating, stratigraphies,
cultural implications Proceedings of Symposium 61 of the XIVth Congress of the UISPP, pp. 313–349.

 **Appendix C - Individual Bayesian Models**

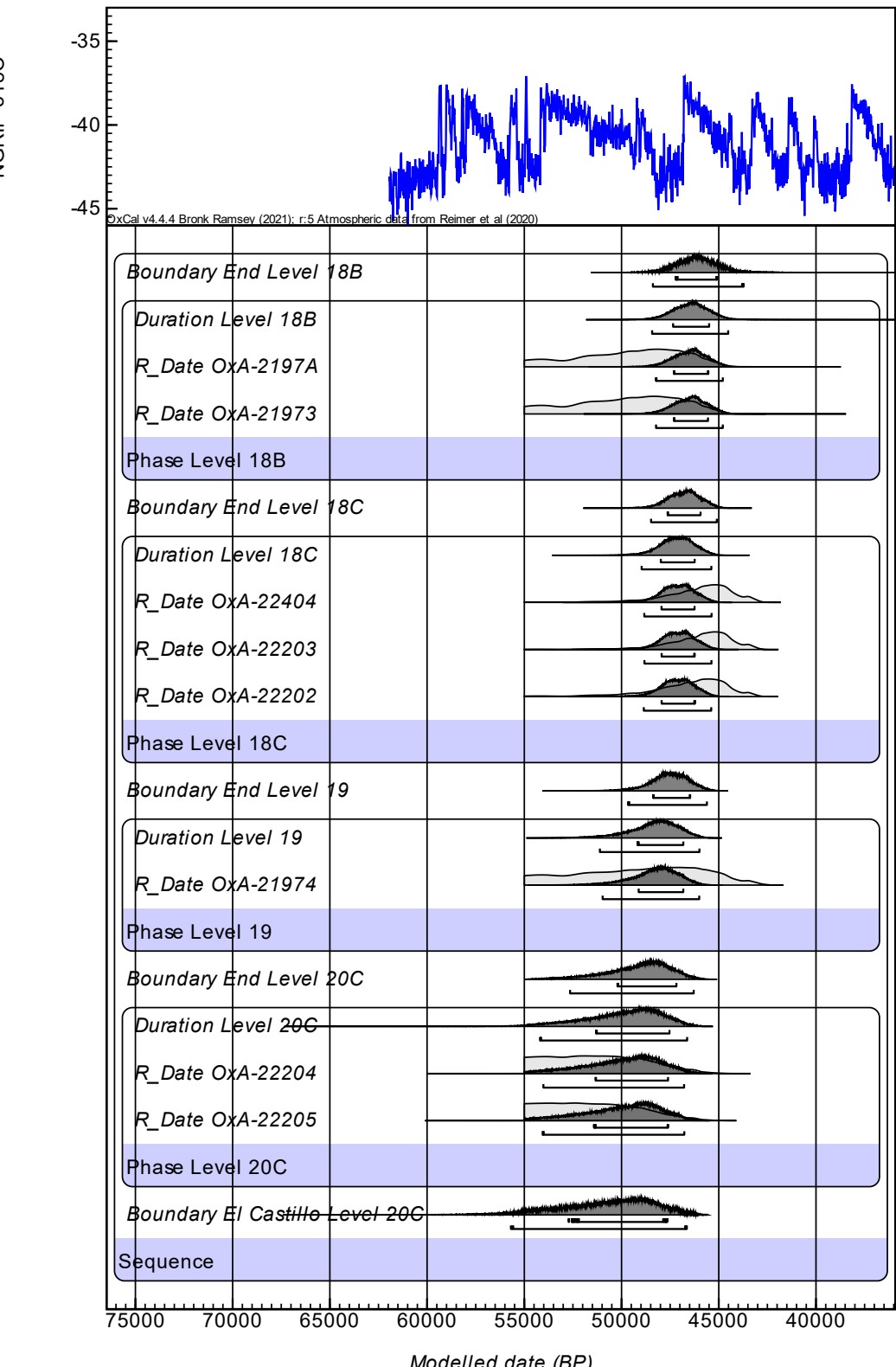


**Figure C1.** Radiocarbon dates from El Castillo modelled in OxCal4.4 against INTCAL20.




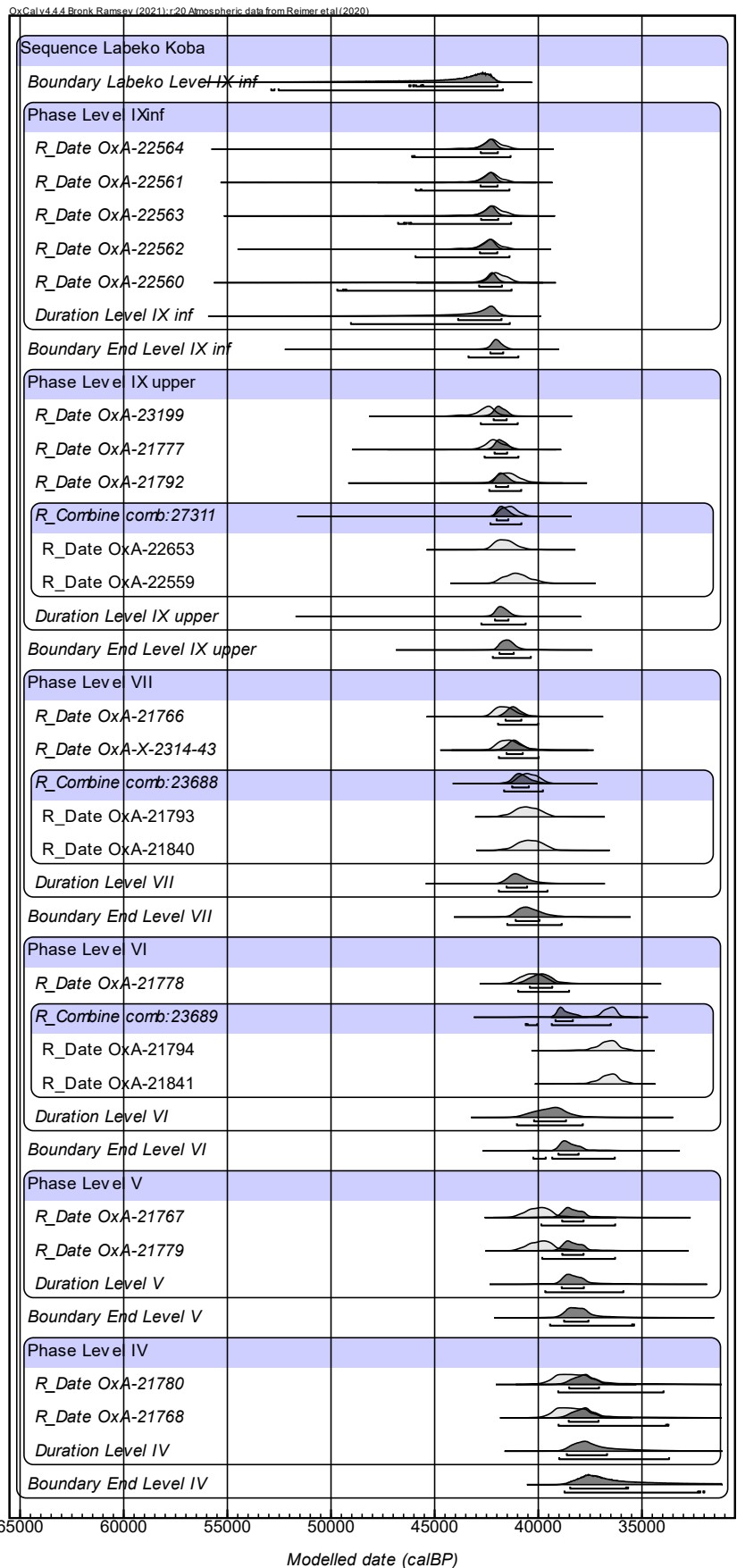

**Figure C2.** Radiocarbon dates from Labeko Koba modelled in OxCal4.4 against INTCAL20.


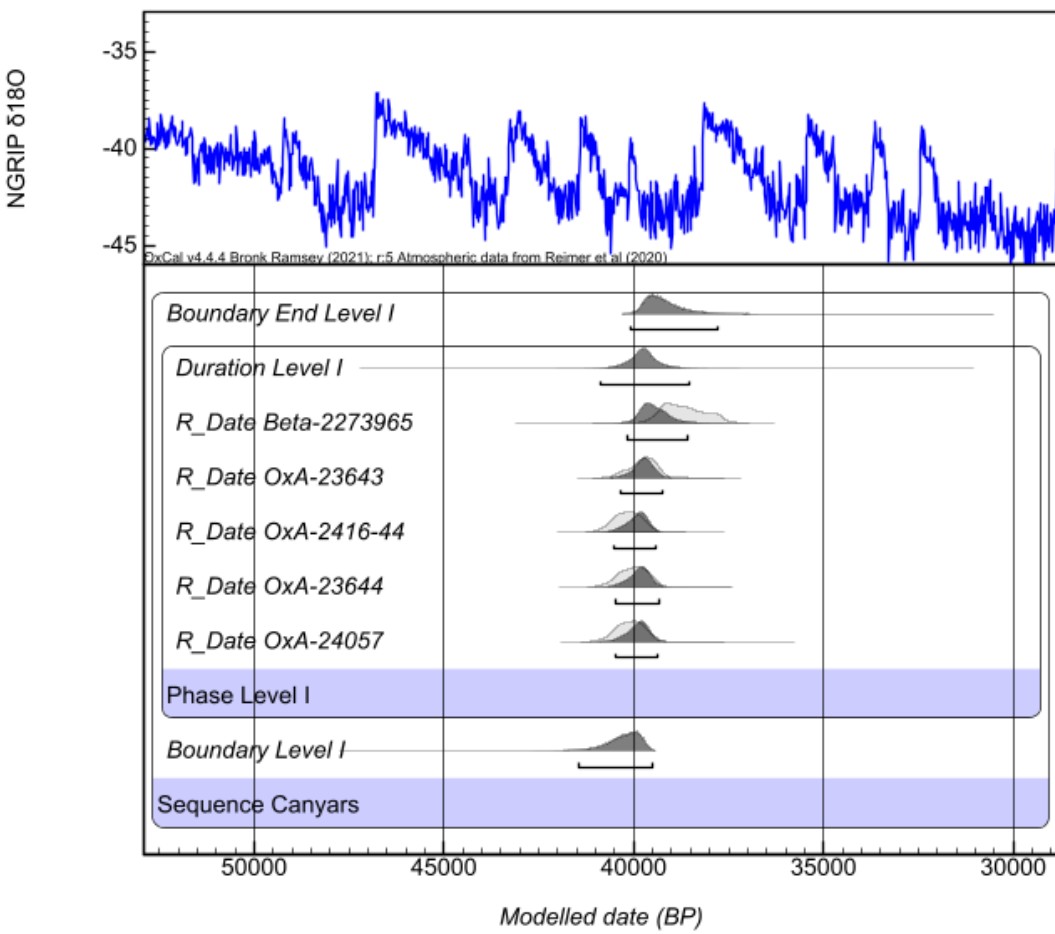


**Figure C3.** Radiocarbon dates from Canyars modelled in OxCal4.4 against INTCAL20.

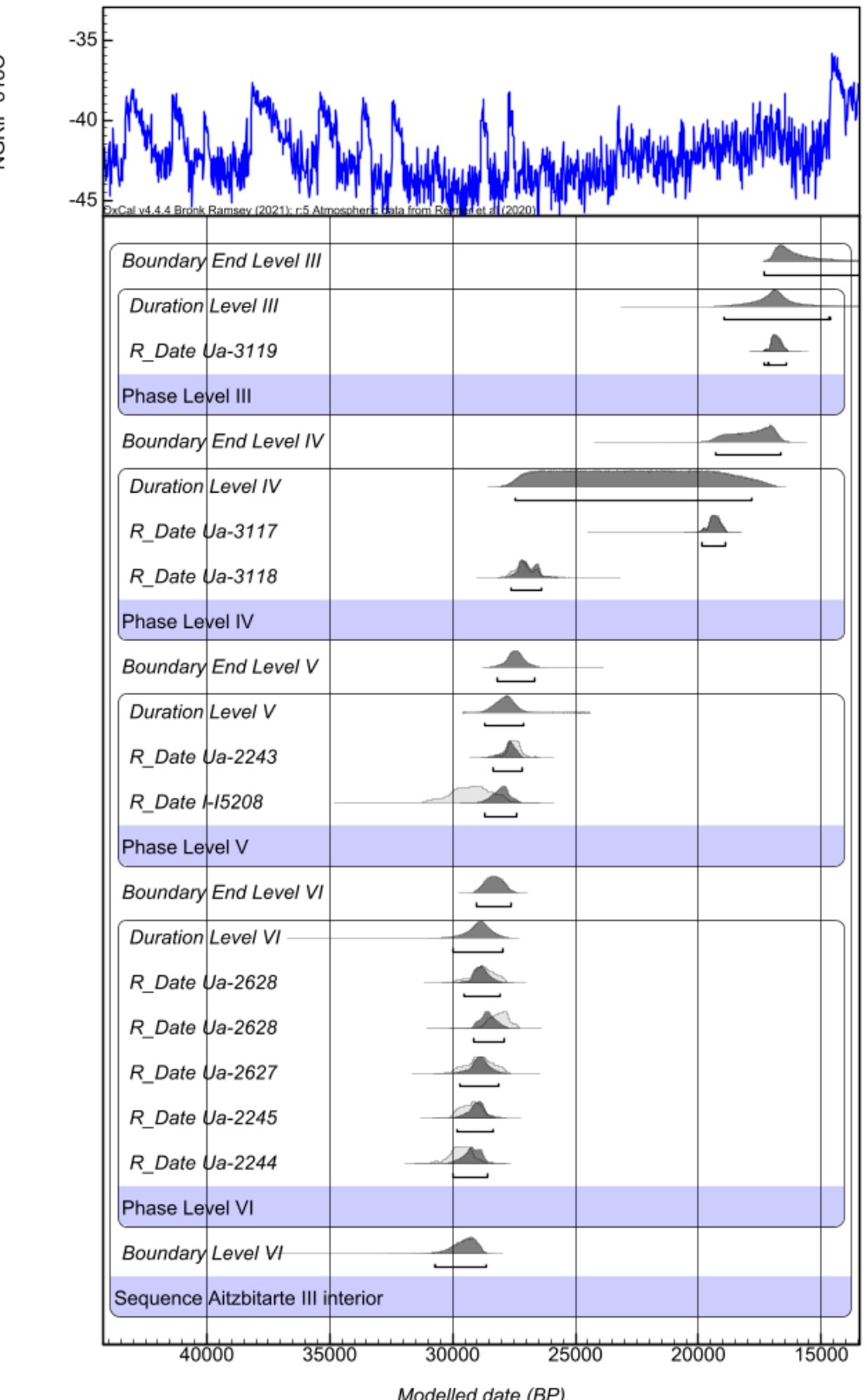

Figure C4. Radiocarbon dates from Aitzbitarte III-interior modelled in OxCal4.4 against INTCAL20.

## Results of Bayesian Models

| El Castillo | Unmodelled (BP) | | | Modelled (BP) | | | Indices Amodel 78.8, Aoverall 82.4 | | | |
|---|---|---|---|---|---|---|---|---|---|---|
| | from | to | % | from | to | % | A | L | P | C |
| Boundary End Level 18B | | | | 48383 | 43733 | 95.449.974 | | | | 97.1 |
| **Duration Level 18B** | | | | **48438** | **44536** | **95.449.974** | | | | **99.8** |
| R_Date OxA-2197A | ... | 45427 | 95.449.973 | 48235 | 44793 | 95.449.974 | 98.1 | | 95.2 | 99.8 |
| R_Date OxA-21973 | ... | 45655 | 95.449.973 | 48240 | 44793 | 95.449.974 | 91.9 | | 95.2 | 99.8 |
| Phase Level 18B | | | | | | | | | | |
| Boundary End Level 18C | | | | 48470 | 45117 | 95.449.974 | | | | 99.8 |
| **Duration Level 18C** | | | | **48977** | **45382** | **95.449.974** | | | | **99.9** |
| R_Date OxA-22404 | 49976 | 42918 | 95.449.974 | 48833 | 45383 | 95.449.974 | 82.2 | | 95.3 | 99.8 |
| R_Date OxA-22203 | 49451 | 42999 | 95.449.974 | 48819 | 45381 | 95.449.974 | 76.1 | | 95.2 | 99.8 |
| R_Date OxA-22202 | 51146 | 43039 | 95.449.974 | 48861 | 45386 | 95.449.974 | 101.2 | | 95.4 | 99.8 |
| Phase Level 18C | | | | | | | | | | |
| Boundary End Level 19 | | | | 49629 | 45623 | 95.449.974 | | | | 99.7 |
| Duration Level 19 | | | | 51060 | 45997 | 95.449.974 | | | | 99.7 |
| R_Date OxA-21974 | ... | 44367 | 95.449.974 | 50965 | 45998 | 95.449.974 | 120.2 | | 95.3 | 99.8 |
| Phase Level 19 | | | | | | | | | | |
| Boundary End Level 20C | | | | 52583 | 46286 | 95.449.974 | | | | 99.5 |
| **Duration Level 20C** | | | | **54134** | **46593** | **95.449.974** | | | | **99.3** |
| R_Date OxA-22204 | ... | 47048 | 95.449.974 | 53958 | 46713 | 95.449.974 | 94 | | 95.3 | 99.3 |
| R_Date OxA-22205 | ... | 47348 | 95.449.974 | 53965 | 46715 | 95.449.974 | 86.9 | | 95.3 | 99.3 |
| Phase Level 20C | | | | | | | | | | |
| Boundary El Castillo Level 20C | | | | 55552 | 46609 | 95.449.974 | | | | 95.3 |
| Sequence | | | | | | | | | | |
| U(0 | 68.268.949 | 3.99E-17 | 4 | 68.268.949 | 5.38E-17 | 3.776 | | 100 | | |
| T(5) | -2.65 | 2.65 | 95.449.974 | | | | | | | 99.9 |
| Outlier_Model General | | | | -2684 | 2502 | 95.449.974 | | | | 100 |

**Table C1.** Radiocarbon dates from El Castillo modelled in OxCal4.4 against INTCAL20.

| Aitzbitarte III Interior | Unmodelled (BP) | | | Modelled (BP) | | | Indices Amodel 78.8, Aoverall 82.4 | | | |
|---|---|---|---|---|---|---|---|---|---|---|
| | from | to | % | from | to | % | A | L | P | C |
| Boundary End Level III | | | | 17300 | 12910 | 9.544.997 | | | | 98 |
| Duration Level III | | | | 18960 | 14630 | 9.544.997 | | | | 99.6 |
| R_Date Ua-3119 | 17270 | 16390 | 9.544.997 | 17300 | 16430 | 9.544.997 | 100.8 | | 95.8 | 99.8 |
| Phase Level III | | | | | | | | | | |
| Boundary End Level IV | | | | 19320 | 16640 | 9.544.997 | | | | 99.3 |
| Duration Level IV | | | | 27430 | 17820 | 9.544.997 | | | | 98.9 |
| R_Date Ua-3117 | 19830 | 18900 | 9.544.997 | 19840 | 18910 | 9.544.997 | 99.9 | | 95.3 | 99.6 |
| R_Date Ua-3118 | 27700 | 26430 | 9.544.997 | 27600 | 26360 | 9.544.997 | 98.1 | | 95.2 | 99.5 |
| Phase Level IV | | | | | | | | | | |
| Boundary End Level V | | | | 28210 | 26680 | 9.544.997 | | | | 99.7 |
| **Duration Level V** | | | | **28680** | **27130** | **9.544.997** | | | | **99.9** |
| R_Date Ua-2243 | 28260 | 26610 | 9.544.997 | 28370 | 27190 | 9.544.997 | 88.8 | | 95.4 | 99.8 |
| R_Date I-I5208 | 30830 | 27760 | 9.544.997 | 28710 | 27370 | 9.544.997 | 57.7 | | 94.8 | 99.8 |
| Phase Level V | | | | | | | | | | |
| Boundary End Level VI | | | | 29010 | 27630 | 9.544.997 | | | | 99.7 |
| Duration Level VI | | | | 29990 | 27930 | 9.544.997 | | | | 99.8 |
| R_Date Ua-2628 | 29760 | 27840 | 9.544.997 | 29570 | 28080 | 9.544.997 | 118.2 | | 96 | 99.8 |
| R_Date Ua-2628 | 28760 | 27360 | 9.544.997 | 29150 | 27920 | 9.544.997 | 67 | | 94.3 | 99.8 |
| R_Date Ua-2627 | 29920 | 27870 | 9.544.997 | 29680 | 28110 | 9.544.997 | 120.5 | | 96 | 99.8 |
| R_Date Ua-2245 | 30070 | 28280 | 9.544.997 | 29820 | 28360 | 9.544.997 | 108 | | 95.9 | 99.8 |
| R_Date Ua-2244 | 30720 | 28760 | 9.544.997 | 30010 | 28570 | 9.544.997 | 77.7 | | 94.9 | 99.7 |
| Phase Level VI | | | | | | | | | | |
| Boundary Level VI | | | | 30730 | 28650 | 9.544.997 | | | | 96 |
| Sequence | | | | | | | | | | |
| U(0,4) | 3.99E-17 | 4 | 9.544.997 | 5.38E-17 | 3.772 | 9.544.997 | 100 | | | 99 |
| T(5) | -2.65 | 2.65 | 9.544.997 | | | | | | | 95.5 |
| Outlier_Model General | | | | -1420 | 1280 | 9.544.997 | | | | 99.9 |

**Table C2.** Radiocarbon dates from Labeko Koba modelled in OxCal4.4 against INTCAL20.

| Canyars | Unmodelled (BP) | | | Modelled (BP) | | | Indices Amodel 78.8, Aoverall 82.4 | | | |
|---|---|---|---|---|---|---|---|---|---|---|
| Boundary End Level I | | | | 40090 | 37770 | 95.45 | | | | 95.3 |
| **Duration Level I** | | | | **40890** | **38530** | **95.45** | | | | **99.7** |
| R_Date Beta-2273965 | 39630 | 37570 | 9.544.997 | 40190 | 38560 | 95.45 | 63.2 | | 93.4 | 99.6 |
| R_Date OxA-23643 | 40520 | 39140 | 9.544.997 | 40330 | 39240 | 95.45 | 114.2 | | 96.1 | 99.8 |
| R_Date OxA-2416-44 | 40880 | 39450 | 9.544.997 | 40540 | 39400 | 95.45 | 99.2 | | 96 | 99.8 |
| R_Date OxA-23644 | 40740 | 39300 | 9.544.997 | 40470 | 39340 | 95.45 | 110.5 | | 96 | 99.8 |
| R_Date OxA-24057 | 40790 | 39390 | 9.544.997 | 40490 | 39380 | 95.45 | 104.3 | | 96 | 99.8 |
| Phase Level I | | | | | | | | | | |
| Boundary Level I | | | | 41450 | 39500 | 95.45 | | | | 96.6 |
| Sequence Canyars | | | | | | | | | | |
| U(0,4) | 3.99E-17 | 4 | 9.544.997 | 5.38E-17 | 3.82 | 95.45 | 100 | | | 100 |
| T(5) | -2.65 | 2.65 | 9.544.997 | | | | | | | 99.4 |
| Outlier_Model General | | | | -800 | 1480 | 95.45 | | | | 99.9 |

**Table C3.** Radiocarbon dates from Canyars modelled in OxCal4.4 against INTCAL20.

| Labeko Koba | Unmodelled (BP) | | | Modelled (BP) | | | Indices Amodel 78.8, Aoverall 82.4 | | | |
|---|---|---|---|---|---|---|---|---|---|---|
| | from | to | % | from | to | % | A | L | P | C |
| Boundary End Level IV | | | | 38710 | 32030 | 9.544.997 | | | | 98.4 |
| **Duration Level IV** | | | | **39000** | **33710** | **9.544.997** | | | | **99.8** |
| R_Date OxA-21768 | 39700 | 37030 | 9.544.997 | 39050 | 33820 | 9.544.997 | 75.5 | | 80 | 99.8 |
| R_Date OxA-21780 | 39780 | 36910 | 9.544.997 | 39050 | 33960 | 9.544.997 | 81.3 | | 82.3 | 99.8 |
| Phase Level IV | | | | | | | | | | |
| Boundary End Level V | | | | 39470 | 35440 | 9.544.997 | | | | 99.8 |
| **Duration Level V** | | | | **39730** | **35950** | **9.544.997** | | | | **99.8** |
| R_Date OxA-21779 | 41170 | 38260 | 9.544.997 | 39830 | 36330 | 9.544.997 | 21 | | 87.2 | 99.8 |
| R_Date OxA-21767 | 41230 | 38500 | 9.544.997 | 39860 | 36340 | 9.544.997 | 15.5 | | 85.5 | 99.8 |
| Phase Level V | | | | | | | | | | |
| Boundary End Level VI | | | | 40240 | 36360 | 9.544.997 | | | | 99.8 |
| **Duration Level VI** | | | | **41030** | **37860** | **9.544.997** | | | | **99.9** |
| R_Date OxA-21841 | 37710 | 35420 | 9.544.997 | | | | | | | |
| R_Date OxA-21794 | 38040 | 35460 | 9.544.997 | | | | | | | |
| R_Combine comb:23689 | 37350 | 35900 | 9.544.997 | 40620 | 36500 | 9.544.997 | 4.3 | | | 99.8 |
| R_Date OxA-21778 | 41390 | 39190 | 9.544.997 | 40970 | 38550 | 9.544.997 | 90 | | 94.4 | 99.9 |
| Phase Level VI | | | | | | | | | | |
| Boundary End Level VII | | | | 41490 | 38890 | 9.544.997 | | | | 99.9 |
| **Duration Level VII** | | | | **41910** | **39570** | **9.544.997** | | | | **99.9** |
| R_Date OxA-21840 | 41610 | 39250 | 9.544.997 | | | | | | | |
| R_Date OxA-21793 | 41720 | 39390 | 9.544.997 | | | | | | | |
| R_Combine comb:23688 | 41290 | 39570 | 9.544.997 | 41650 | 39780 | 9.544.997 | 87.3 | | | 99.9 |
| R_Date OxA-X-2314-43 | 42350 | 40260 | 9.544.997 | 41900 | 40000 | 9.544.997 | 96.5 | | 95.4 | 99.9 |
| R_Date OxA-21766 | 42520 | 40530 | 9.544.997 | 41950 | 40020 | 9.544.997 | 80.3 | | 94.6 | 99.9 |
| Phase Level VII | | | | | | | | | | |
| Boundary End Level IX upper | | | | 42190 | 40360 | 9.544.997 | | | | 99.9 |
| **Duration Level IX upper** | | | | **42750** | **40580** | **9.544.997** | | | | **99.9** |
| R_Date OxA-22559 | 42090 | 39850 | 9.544.997 | | | | | | | |
| R_Date OxA-22653 | 42520 | 40530 | 9.544.997 | | | | | | | |
| R_Combine comb:27311 | 42120 | 40600 | 9.544.997 | 42330 | 40800 | 9.544.997 | 95 | | | 99.9 |
| R_Date OxA-21792 | 42370 | 40330 | 9.544.997 | 42380 | 40820 | 9.544.997 | 113.4 | | 95.7 | 99.9 |
| R_Date OxA-21777 | 43160 | 40960 | 9.544.997 | 42600 | 40950 | 9.544.997 | 99.5 | | 95.6 | 99.9 |
| R_Date OxA-23199 | 43980 | 41490 | 9.544.997 | 42800 | 40990 | 9.544.997 | 52.4 | | 92.8 | 99.9 |
| Phase Level IX upper | | | | | | | | | | |
| Boundary End Level IX inf | | | | 43420 | 40970 | 9.544.997 | | | | 99.9 |
| **Duration Level IX inf** | | | | **48940** | **41340** | **9.544.997** | | | | **99.8** |
| R_Date OxA-22560 | 42780 | 40980 | 9.544.997 | 49670 | 41300 | 9.544.997 | 75.3 | | 76 | 99.8 |
| R_Date OxA-22562 | 43830 | 41220 | 9.544.997 | 45860 | 41380 | 9.544.997 | 102.8 | | 90.9 | 99.8 |
| R_Date OxA-22563 | 43250 | 41010 | 9.544.997 | 46280 | 41300 | 9.544.997 | 99.1 | | 89.7 | 99.8 |
| R_Date OxA-22561 | 43790 | 41130 | 9.544.997 | 45920 | 41340 | 9.544.997 | 102.3 | | 90.7 | 99.8 |
| R_Date OxA-22564 | 43370 | 41050 | 9.544.997 | 46060 | 41320 | 9.544.997 | 101 | | 90.2 | 99.8 |
| Phase Level IXinf | | | | | | | | | | |
| Boundary Labeko Level IX inf | | | | 52660 | 41740 | 9.544.997 | | | | 96.6 |
| Sequence Labeko Koba | | | | | | | | | | |
| N(0,2) | -4 | 4 | 9.544.997 | | | | | | | 99.4 |
| Outlier_Model SSimple | | | | ... | 840 | 9.544.997 | | | | 97.5 |
| U(0,4) | 3.99E-17 | 4 | 9.544.997 | 5.38E-17 | 3.932 | 9.544.997 | 100 | | | 98.3 |
| T(5) | -2.65 | 2.65 | 9.544.997 | | | | | | | 97.5 |
| Outlier_Model General | | | | -6130 | 9280 | 9.544.997 | | | | 99.4 |

**Table C4.** Radiocarbon dates from Aitzbitarte III-interior modelled in OxCal4.4 against INTCAL20.

## Appendix D. Intratooth curve plots

Original curves derived from enamel intratooth sampling on enamel carbonate. Provided by sites. In blue, oxygen stable isotope composition (δ18O), and, in brown, carbon stable isotope composition (δ13C). In the x-axis, the distance from Enamel Rooth Junction (ERJ). Notice that the y-axis can experience some variations between sites.

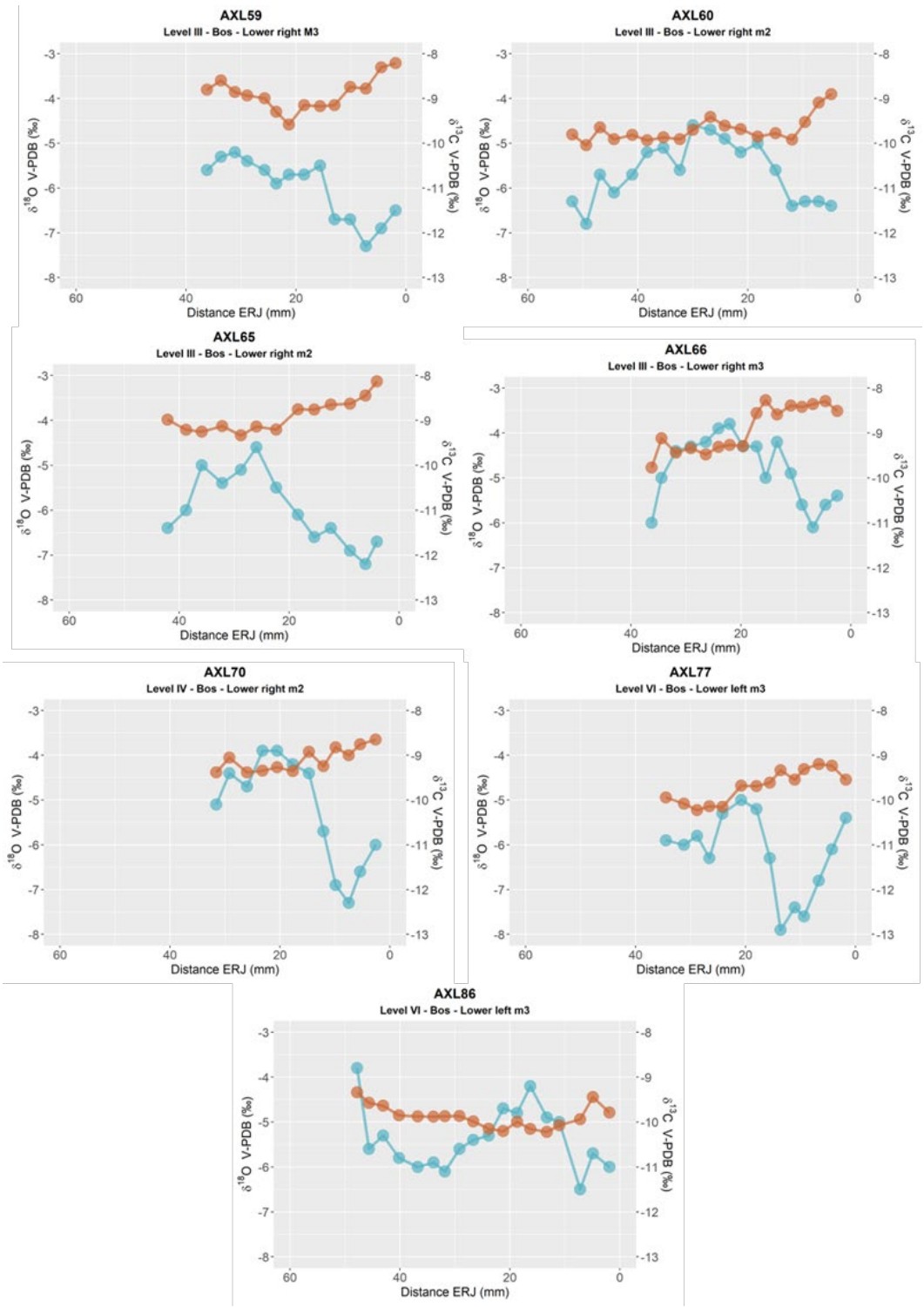

**Figure D1.** Intratooth plots of oxygen (δ18O) and carbon (δ13C) isotope composition from teeth from Axlor, considering distance from enamel root junction (ERC).

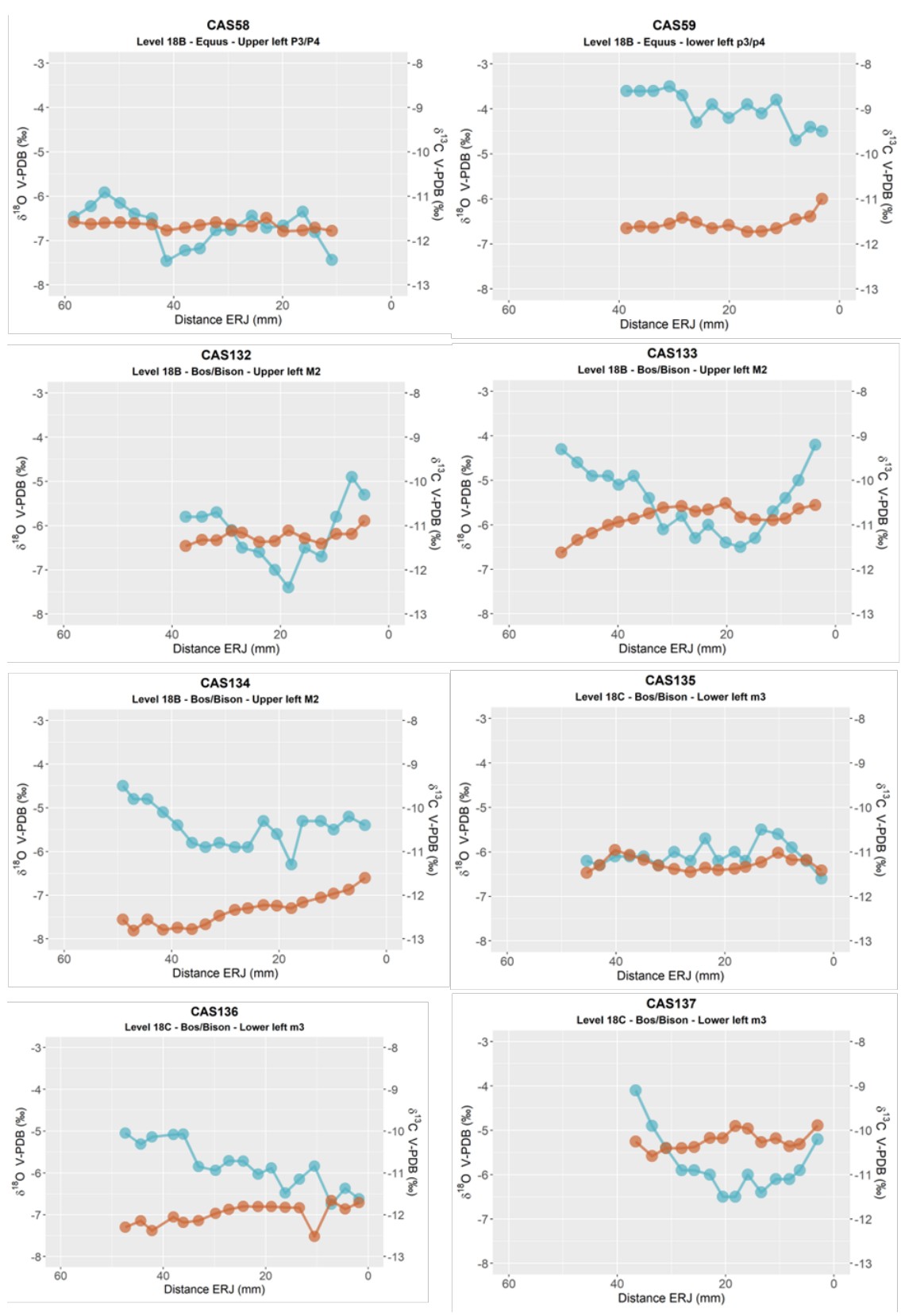

**Figure D2.** Intratooth plots of oxygen (δ18O) and carbon (δ13C) isotope composition from teeth from El Castillo, considering the sample's distance from the enamel root junction (ERC).

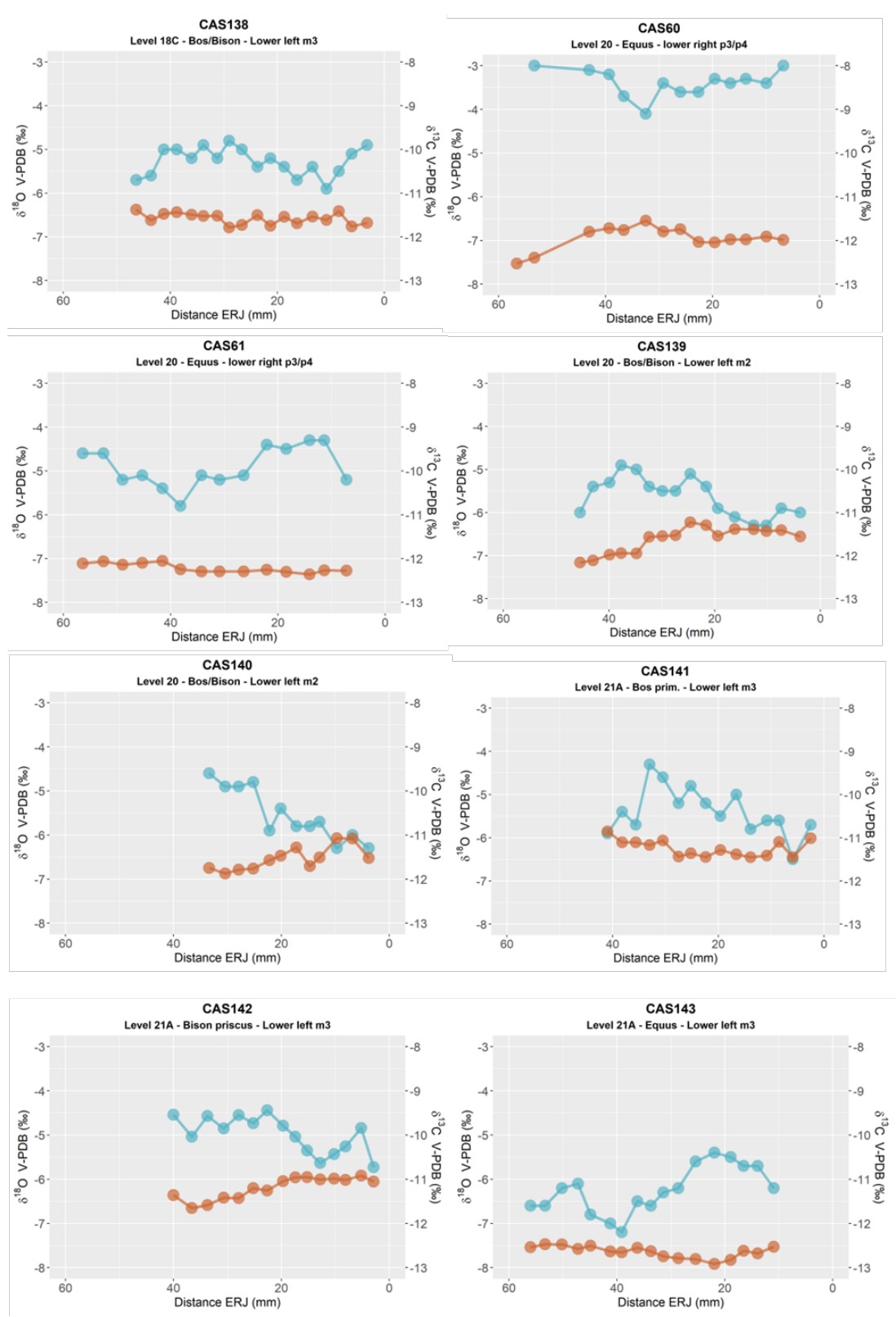

**Figure D3.** Intratooth plots of oxygen (δ$^{18}$O) and carbon (δ$^{13}$C) isotope composition from teeth from El Castillo, considering the sample's distance from the enamel root junction (ERC).

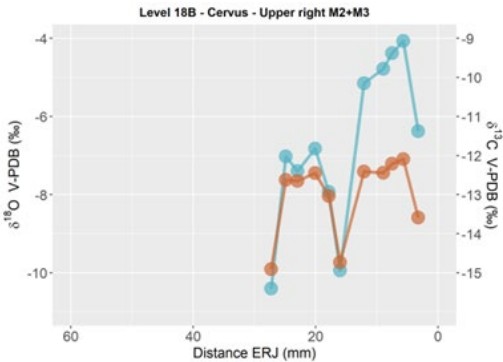

**Figure D4.** Intratooth plots of oxygen (δ18O) and carbon (δ13C) isotope composition from teeth from El Castillo, considering the sample's distance from the enamel root junction (ERC).

**Figure D5.** Intratooth plots of oxygen (δ18O) and carbon (δ13C) isotope composition from teeth from Labeko Koba, considering the sample's distance from the enamel root junction (ERC).

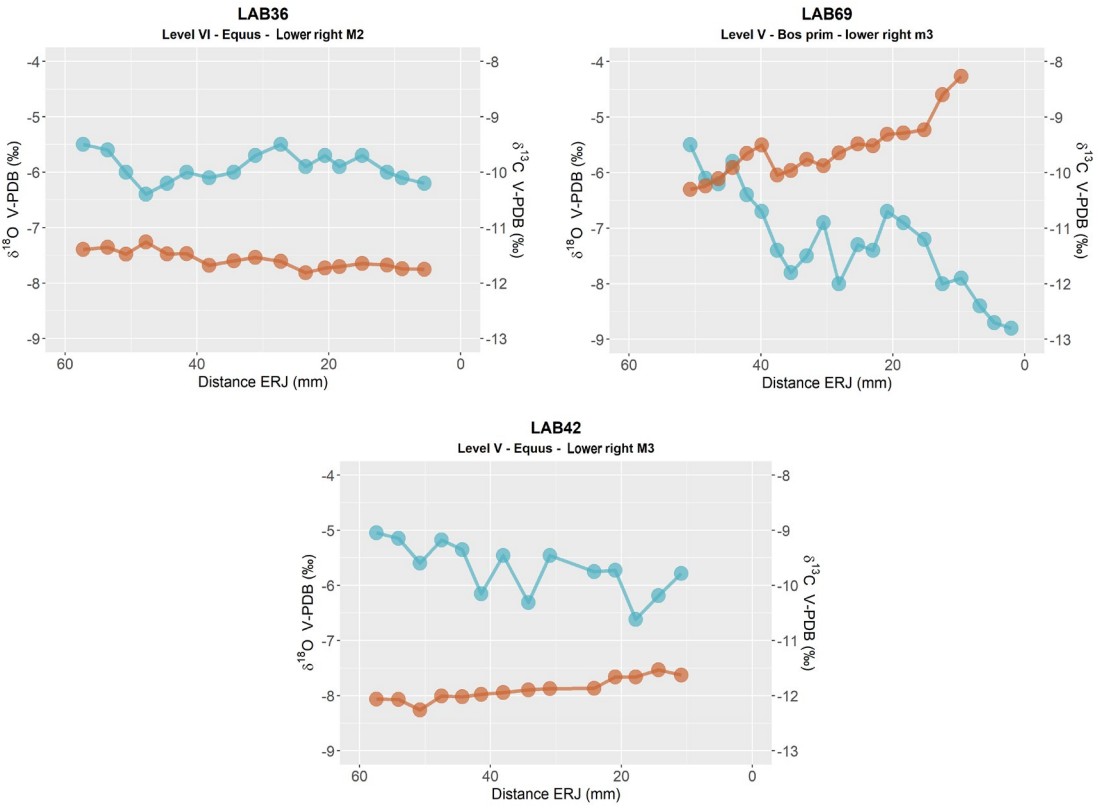

**Figure D6.** Intratooth plots of oxygen (δ18O) and carbon (δ13C) isotope composition from teeth from Labeko Koba, considering the sample's distance from the enamel root junction (ERC).

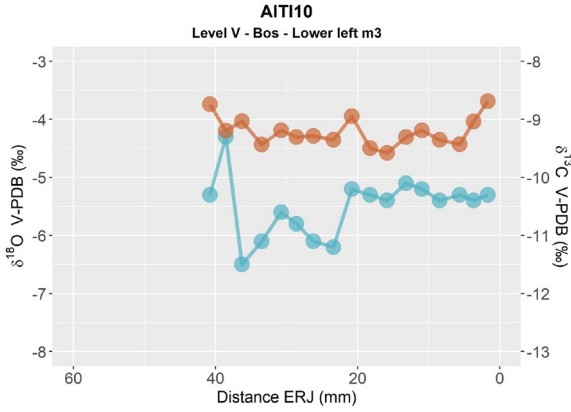

**Figure D7.** Intratooth plots of oxygen (δ18O) and carbon (δ13C) isotope composition from teeth from Aitzbitarte III interior, considering the sample's distance from the enamel root junction (ERC).

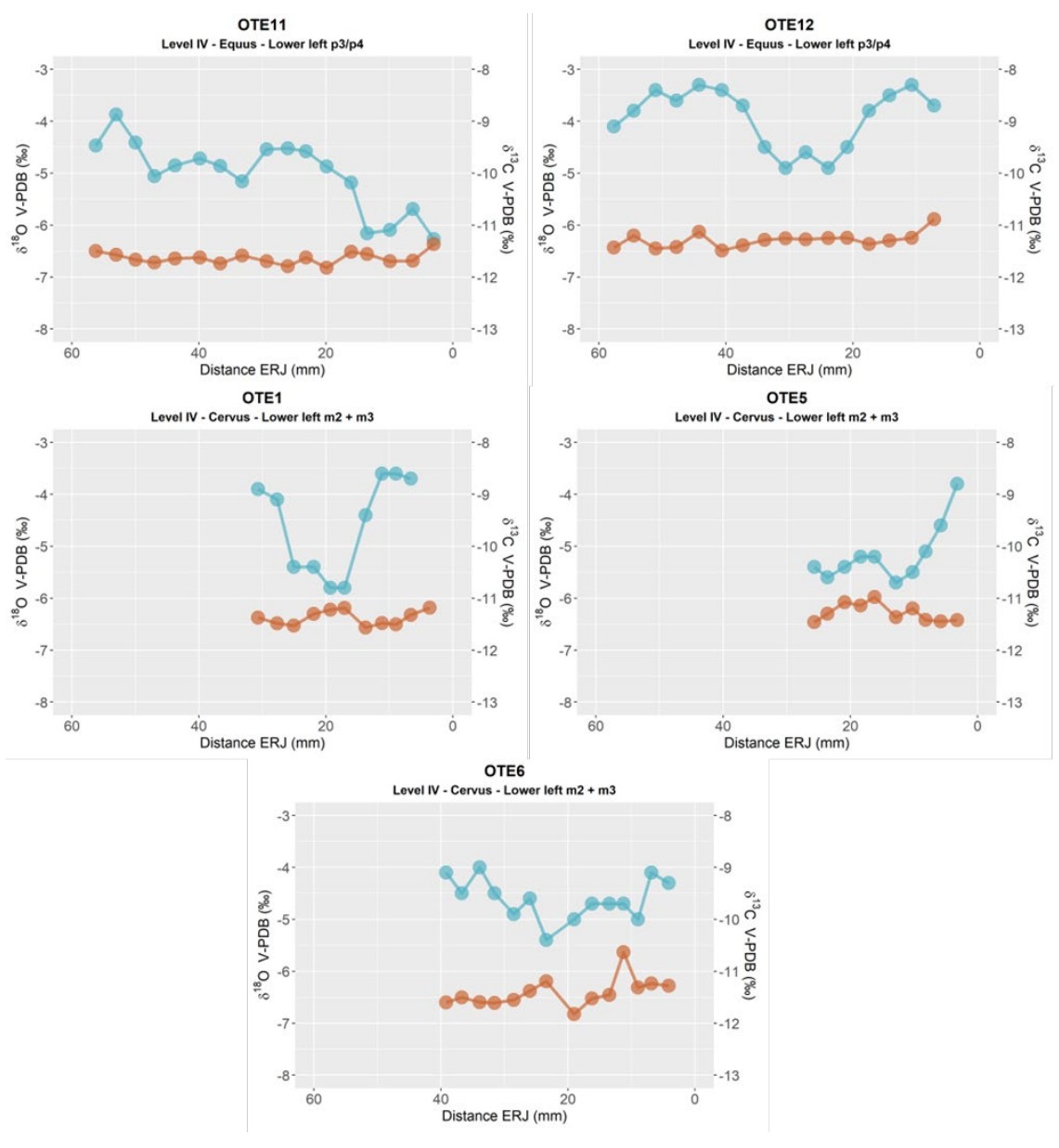

**Figure D8.** Intratooth plots of oxygen (δ¹⁸O) and carbon (δ¹³C) isotope composition from teeth from El Otero, considering the sample's distance from the enamel root junction (ERC).

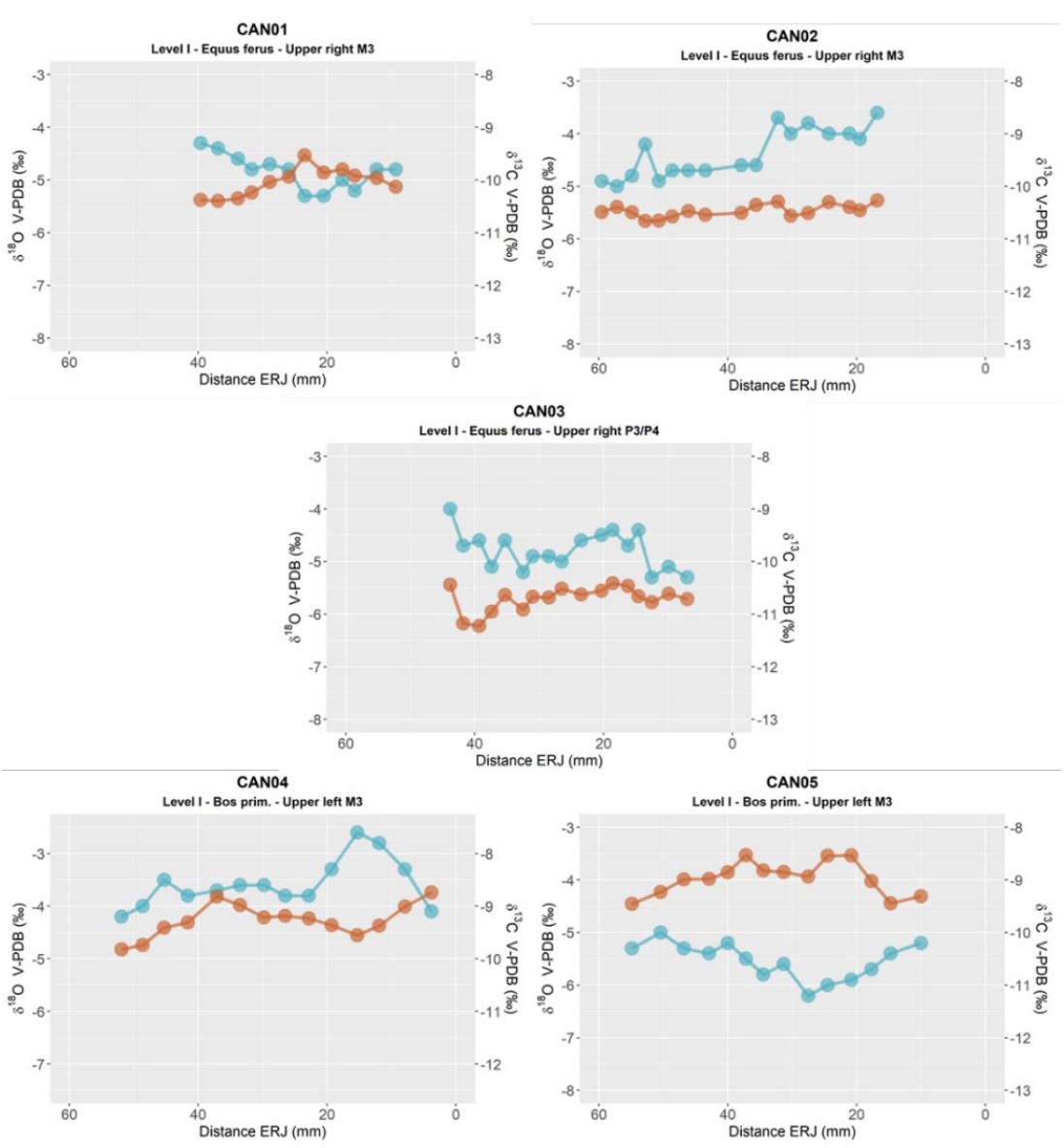

**Figure D9.** Intratooth plots of oxygen (δ¹⁸O) and carbon (δ¹³C) isotope composition from teeth from Canyars considering the sample's distance from the enamel root junction (ERC).

**Appendix E. Inverse Modelling: Methodological Details and Models**

The intratooth $\delta^{18}O$ profiles presented in this study were obtained through the application of inverse modelling, using an adapted version of the code published in reference (Passey et al., 2005b). This modeling approach allowed for the correction of the damping effect and the reconstruction of the original $\delta^{18}O$ input time series. The model reproduces the temporal delay between $\delta^{18}O$ changes in the animal's input and their manifestation in tooth enamel, exhibiting a consistent x-direction delay in the modelled $\delta^{18}O$ curve relative to the enamel $\delta^{18}O$ input time series. The model utilizes different species-specific parameters related to enamel formation, which vary between bovines and equids. These parameters have been established based on previous studies (Bendrey et al., 2015; Zazzo et al., 2012; Passey and Cerling, 2002; Kohn, 2004; Blumenthal et al., 2014). For *Bos/Bison* sp., the initial mineral content of enamel is fixed at 25%, the enamel appositional length is set at 1.5 mm, and the maturation length is 25 mm. For *Equus* sp., the initial mineral content of enamel is fixed at 22%, the enamel appositional length is set at 6 mm, and the maturation length is 28 mm.

In addition, the model requires other variables related to sampling geometry, as well as error estimates derived from mass spectrometer measurements. The distance between samples varies for each tooth, but as a general trend, the sampling depth on the tooth enamel surface in the samples of this study represents approximately 70% of the total enamel depth. The standard deviation of the measurements obtained from the mass spectrometer was typically set at 0.12%, taking into account the uncertainty associated with the standards. Finally, the models require a damping factor that determines the cumulative damping along the isotopic profile by adjusting the measured error (Emeas) to the prediction error (Epred). In the teeth analysed in this study, the damping factor ranged from 0.001 to 0.1.

The most likely model solutions were selected, and summer and winter values were extracted from the $\delta^{18}O$ profiles, considering the original peaks and troughs identified in the unmodelled $\delta^{18}O$ profile. This approach was adopted to prevent the introduction of artificial peaks that the model may produce, particularly in teeth without a distinct sinusoidal shape. Flat and less sinusoidal profile are less suitable for the application of the model, given its inherent assumption of an approximately sinusoidal form. Non-sinusoidal curves can lead to complex interpretations in the model outcomes. Consequently, this methodology was not applied to analysed intratooth $\delta^{13}C$ profiles, as the examined individuals did not exhibit appreciable seasonal change.

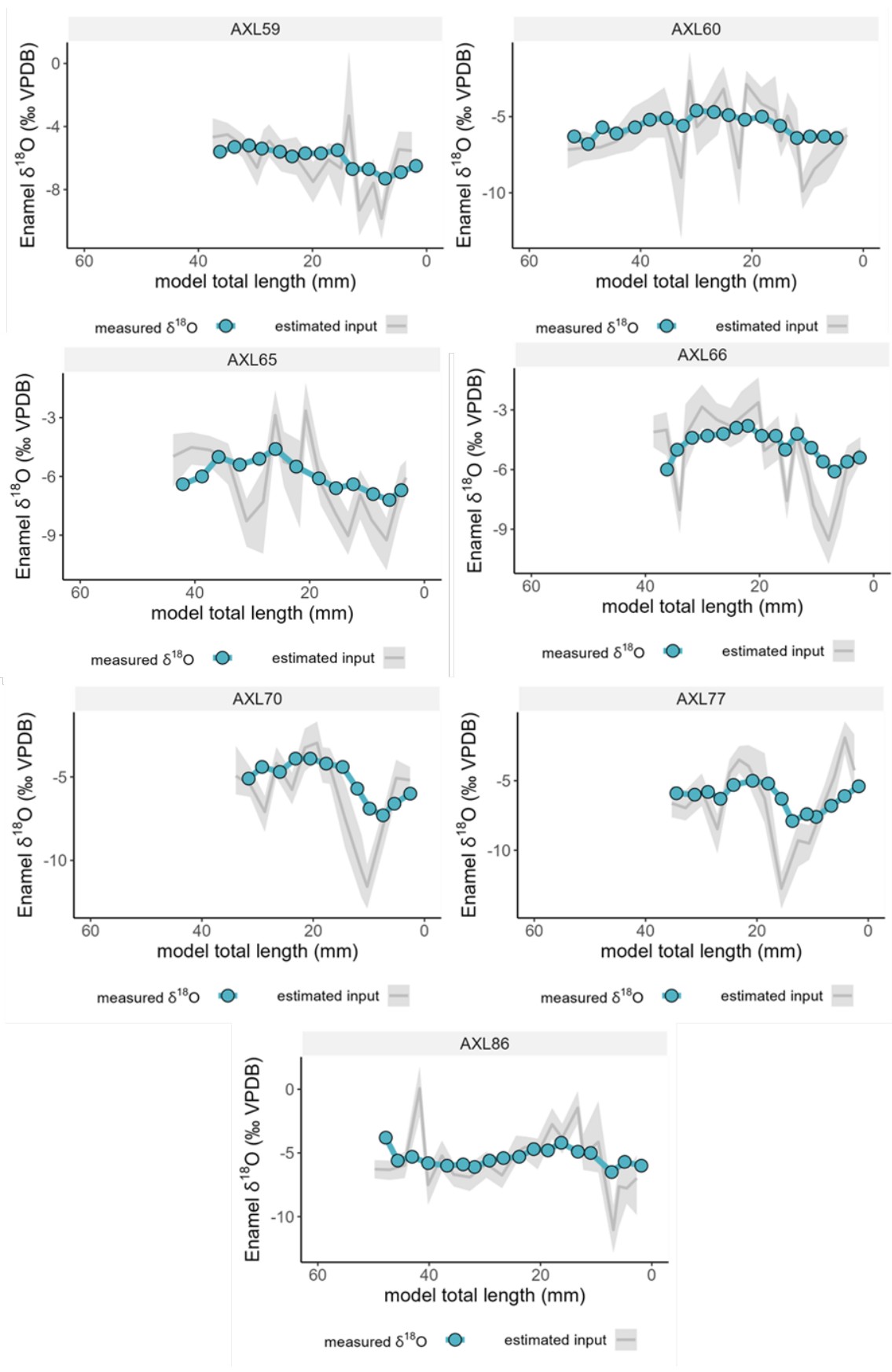

**Figure E1.** Inverse models for oxygen isotope composition (δ[18]O) from teeth from Axlor, considering distance from enamel root junction. The blue line and points correspond to original data and grey line the most likely model solution, with the 95% confidence interval shown in shaded areas.

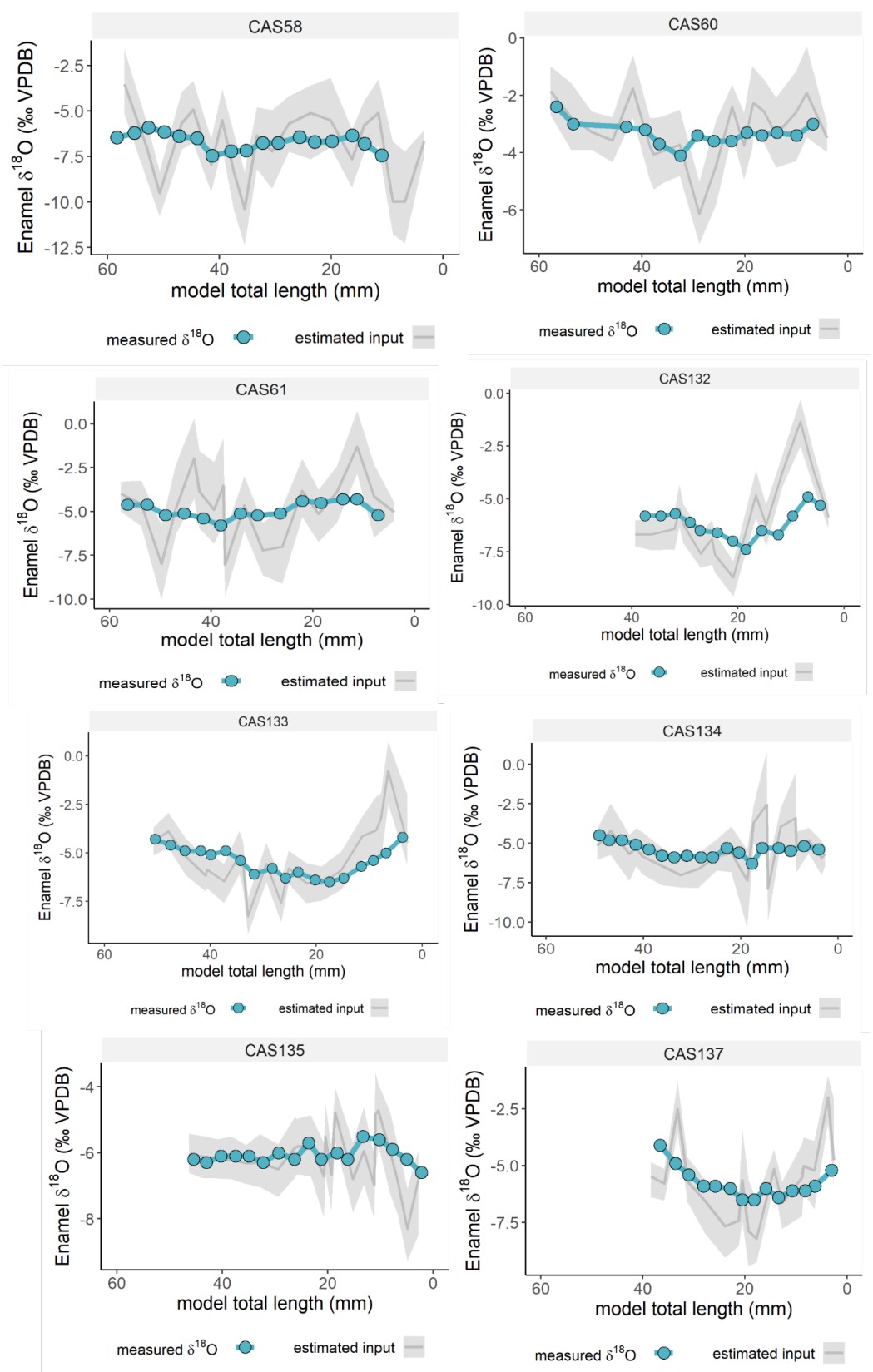

**Figure E2.** Inverse models for oxygen isotope composition (δ18O) from teeth from El Castillo, considering distance from enamel root junction. The blue line and points correspond to original data and grey line the most likely model solution, with the 95% confidence interval shown in shaded areas.

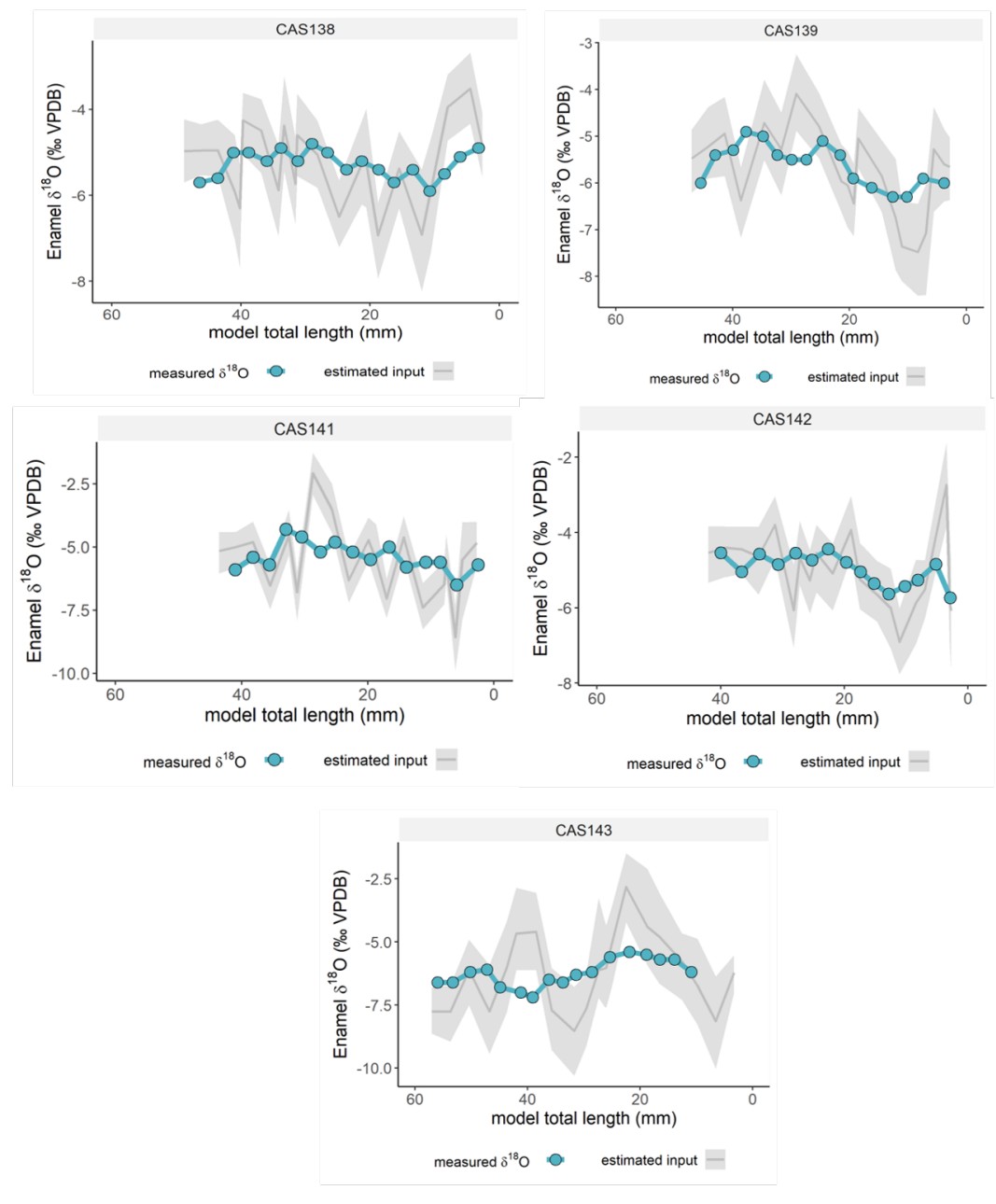

**Figure E3.** Inverse models for oxygen isotope composition (δ18O) from teeth from El Castillo, considering distance from enamel root junction. The blue line and points correspond to original data and grey line the most likely model solution, with the 95% confidence interval shown in shaded areas.

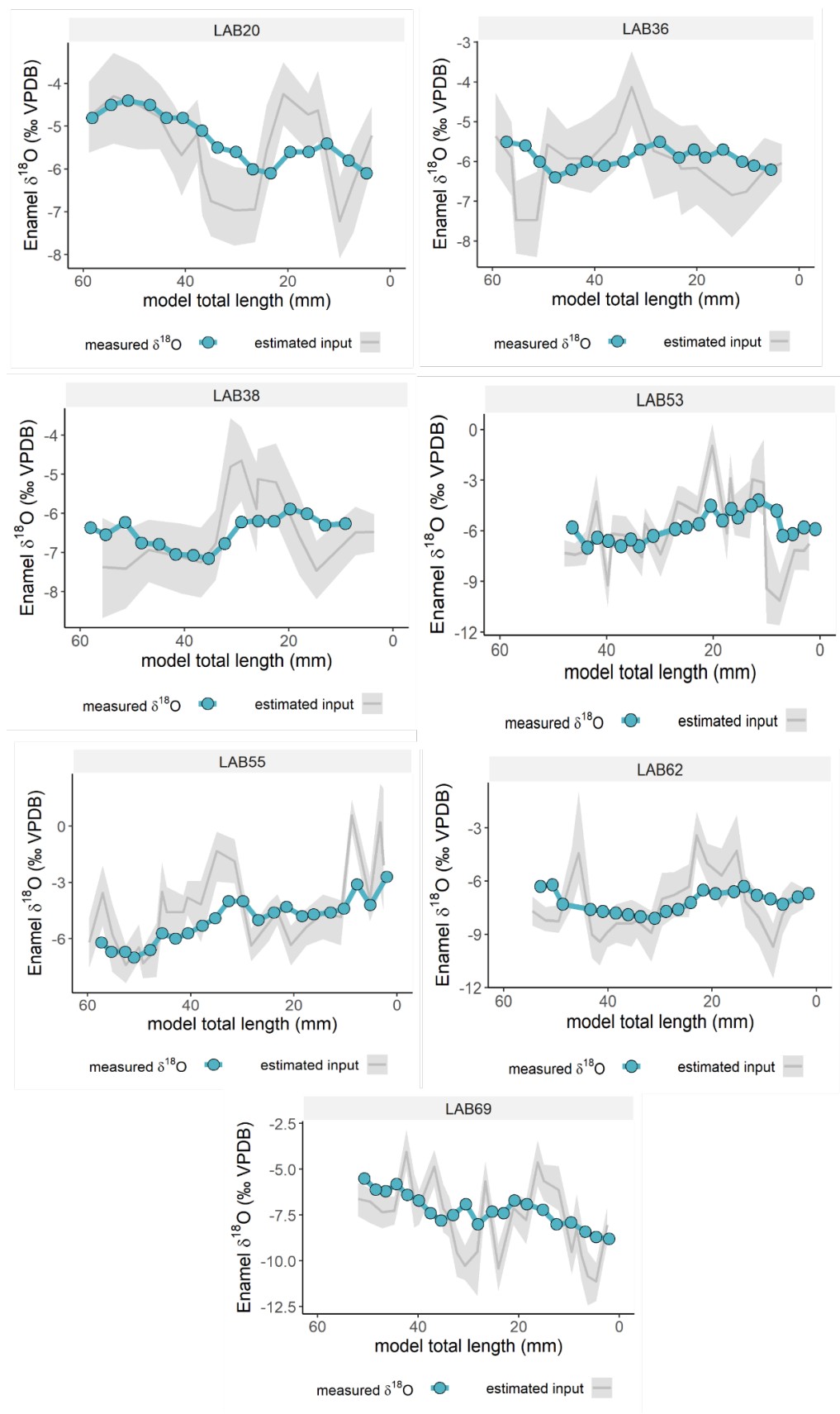

**Figure E4.** Inverse models for oxygen isotope composition (δ18O) from teeth from Labeko Koba, considering distance from enamel root junction. The blue line and points correspond to original data and grey line the most likely model solution, with the 95% confidence interval shown in shaded areas.

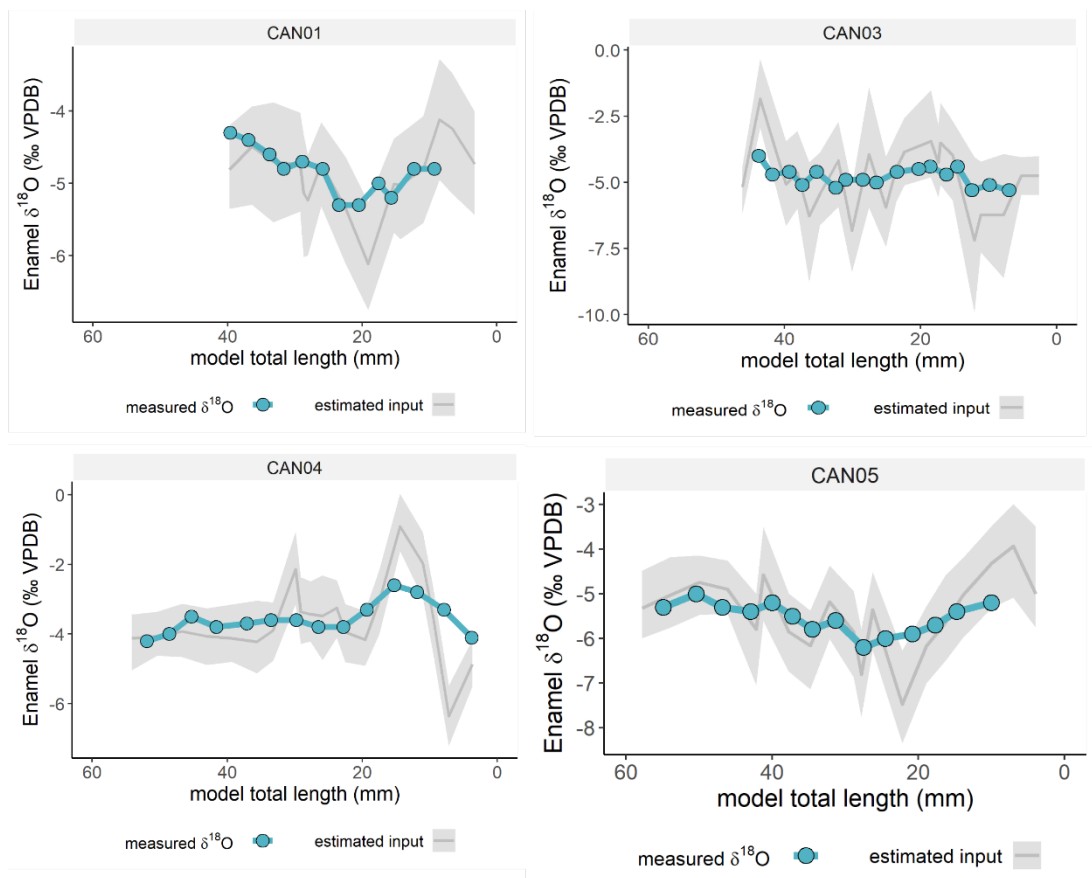

**Figure E5.** Inverse models for oxygen isotope composition (δ18O) from teeth from Canyars considering distance from enamel root junction. The blue line and points correspond to original data and grey line the most likely model solution, with the 95% confidence interval shown in shaded areas.

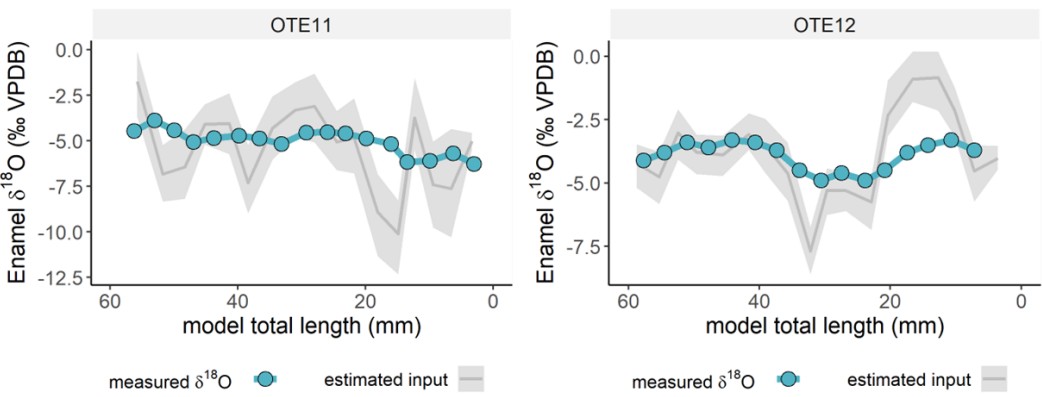

**Figure E6.** Inverse models for oxygen isotope composition (δ18O) from teeth from El Otero, considering distance from enamel root junction. The blue line and points correspond to original data and grey line the most likely model solution, with the 95% confidence interval shown in shaded areas.

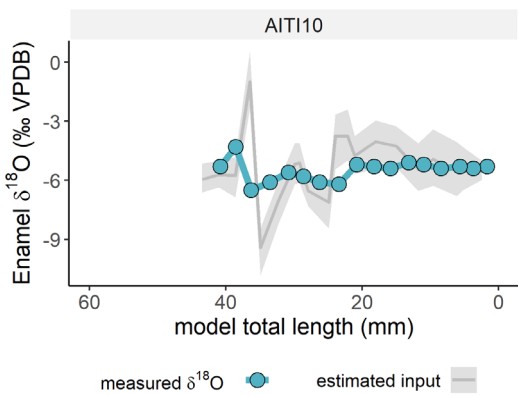

**Figure E7.** Inverse models for oxygen isotope composition (δ18O) from teeth from Aitzbitarte III interior, considering distance from enamel root junction. The blue line and points correspond to original data and grey line the most likely model solution, with the 95% confidence interval shown in shaded areas.

## References Appendix E

Bendrey, R., Vella, D., Zazzo, A., Balasse, M., Lepetz, S., 2015. Exponentially decreasing tooth growth rate in horse teeth: implications for isotopic analyses. Archaeometry 57, 1104–1124. https://doi.org/10.1111/arcm.12151

Blumenthal, S.A., Cerling, T.E., Chritz, K.L., Bromage, T.G., Kozdon, R., Valley, J.W., 2014. Stable isotope time-series in mammalian teeth: In situ δ18O from the innermost enamel layer. Geochimica et Cosmochimica Acta 124, 223–236. https://doi.org/10.1016/j.gca.2013.09.032

Kohn, M.J., 2004. Comment: Tooth Enamel Mineralization in Ungulates: Implications for Recovering a Primary Isotopic Time-Series, by B. H. Passey and T. E. Cerling (2002). Geochimica et Cosmochimica Acta 68, 403–405. https://doi.org/10.1016/S0016-7037(03)00443-5

Passey, B.H., Cerling, T.E., 2002. Tooth enamel mineralization in ungulates: implications for recovering a primary isotopic time-series. Geochimica et Cosmochimica Acta 66, 3225–3234. https://doi.org/10.1016/S0016-7037(02)00933-X

Passey, B.H., Cerling, T.E., Schuster, G.T., Robinson, T.F., Roeder, B.L., Krueger, S.K., 2005. Inverse methods for estimating primary input signals from time-averaged isotope profiles. Geochimica et Cosmochimica Acta 69, 4101–4116. https://doi.org/10.1016/j.gca.2004.12.002

Zazzo, A., Bendrey, R., Vella, D., Moloney, A.P., Monahan, F.J., Schmidt, O., 2012. A refined sampling strategy for intra-tooth stable isotope analysis of mammalian enamel. Geochimica et Cosmochimica Acta 84, 1–13. https://doi.org/10.1016/j.gca.2012.01.012