# Peer review of "Palaeoecology of ungulates in northern Iberia during the Late Pleistocene through isotopic analysis of teeth"

_Biogeosciences, 2023_

## Author Comment (AC2)

**REPLY TO REVIEWER 2**

*This is a remarkable manuscript that deserves publication after major changes.*

*From my point of view there are some overinterpretations on the one hand, and some oversimplifications on the other hand. An important problem of paleoenvironmental reconstructions based only on archaeological remains, like this manuscript, is that the obtained data are not in an objective paleoclimatic and paleoenvironmental frame, in this case (northern) Iberia. This study based the paleoenvironmental reconstructions mainly on the archaeological data instead of comparing them with regional continuous paleoclimatic and paleoenvironmental records. I mean, authors suggested that this type of paleoenvironmental studies is key to understand past climate and human interactions. See for example abstract lines 18-19 or the introduction. Authors must keep in mind that the paleoenvironmental reconstruction that they have performed in these archaeological sites are "discontinuous points" in the paleoclimatic record of the Iberian Peninsula (see for example the chronologies in Fig.2). I mean, the most accurate climatic records for the studied periods are marine, lacustrine and speleothem records, where one can observe the "objective" fluctuations of past climate. Authors should acknowledge this point in the manuscript as well as compare/discuss with these records (there are many for the IP, for example Martrat et al., 2004; 2007; Pérez Mejías et al., 2019; Moreno et al., 2012; González-Sampériz et al., 2020; Camuera et al., 2019; 2022, among others). Some of them also discuss about vegetation changes in NE Iberia, close to the study areas, which would help authors contextualize and discuss their interpretations about the animal diets. Similarly, the speleothem records from the north of Iberia would help constrain temperature/precipitation patterns; there are many of these records in the Vasco-Cantabrian area. Other records from the Iberian Peninsula, the Mediterranean coast or the Iberian margin would show the general climatic patters for this period. Authors should discuss their data (agreement or disagreement) according to these continuous records in order to have a big picture of the paleoclimate and paleoenvironments in the studied period. Otherwise, authors are only comparing the archaeological sites from their two study areas and the work would only be of local interest. In fig. 2 authors included the d18O record of Greenland, but they did not discuss their data according to this record. As I mentioned before, they should compare and discuss the Iberian records instead, since there can be some temporal offsets between some of the events of the NGRIP curve and the Iberian/mediterranean records. Taking all this into account, objective 3 (line 96) is not totally achieved. In any case, my main concerns about this work come from the methodological approach.*

Dear reviewer,

We highly appreciate all the suggestions and reflections provided for the manuscript, as they will significantly enhance the final publication. We have addressed them point by point as follows.

Regarding the discussion or data in connection with other regional proxies, we will include further discussion involving regional continuous paleoclimatic and paleoenvironmental terrestrial records of Iberia, as recommended (Martrat et al., 2004; 2007; Pérez Mejías et al., 2019; Moreno et al., 2012; González-Sampériz et al., 2020; Camuera et al., 2019; 2022), as well as marine ones (Naughton et al., 2007; Roucoux et al., 2001; Sánchez-Goñi et al., 1999; Sánchez-Goñi et al., 2008). We addressed this in our recent paper (Fernández-García et al., 2023). However, it's worth noting at a regional scale that limited continuous paleoclimatic records can be validated today. Nevertheless, we would like to emphasise that our primary focus is based on reconstructing climate that can be directly or indirectly linked to human presence. This is the main reason archaeological data takes priority in this paper, even though we know that they represent "discontinuous points" within the MIS3 stadial-interstadial fluctuations. We will make this perspective more straightforward in the introduction and incorporate further discussion regarding the regional environmental records.

*Chronology: This a very important part of the study and should be presented in a subsection in section 2 or in the methodology (section 3). Please, explain deeply the absolute chronology of the levels where the ungulated remains were collected. Please, specify the dating method: ESR, OSL or 14C in Fig 2. Regarding 14C samples: it is not clear what radiocarbon curve was used to calibrate the 14C samples. Please, re-calibrate all the radiocarbon data with the latest 14C curve from 2020; there are important age shifts after the Holocene comparing the intCal2020 curve and previous calibration curves. Takin into account the confidence intervals of the ages (I suppose 2 sigma for 14C)->Line 98-99: "The chronological resolution in the study areas for this period allows us to correlate regional paleoenvironmental changes with global records": this would be only true for the two sites younger than 30 ka, since the dates of the other sites might overlap stadial and interstadials (and their probability distributions are very large). What are the grey bands in Fig. 2, stadials? Please explain. In any case the chronology of the samples, according to Fig. 2, would only allow to differentiate between stadials/interstadials in the two youngest samples. What is the meaning of the green colour in one date at around 40 ka in Fig. 2 (Canyars)? What is the meaning of the dates (dots and bars) in Fig 2: Do they represent a single dating event, or a sum of distribution of various dates?, please explain this in the caption and in the main text. If it is the sum of distribution, did you use any statistic approach to obtain it (such as the ones that can be obtained from OxCAI software?) Do they represent the ages of the whole archeological sites or of the levels where the ungulated teeth were taken? This is very important to specify in the main text since the age of the remains could vary. Therefore, the caption of Fig. 2 should explain these details and there might be a (sub)section in the main text explaining the chronology of the levels where the remains were taken. This is crucial to validate the discussion of the manuscript*

We have comprehensively described all sites in Appendix A, opting to conserve space within the main manuscript. However, we acknowledge the reviewer's suggestion to include a synthesis in Section 2 ("Archaeological sites and sampled material"), providing more details on chronology, cultural affiliations, and dating methods. It is important to note that our focus does not directly address specific dates in this work; hence, they are not discussed in the methods section. Date calibrations were done using the OxCAL software with the IntCal2020 curve, as outlined in previous publications by our team (e.g., Vidal-Cordasco et al., 2022; Fernández-García et al., 2023). Dates provided represent an approximate age for each level, incorporating either multiple dates or a single date. Further details on dating methods and selected dates for each level are in Appendix B ("B1-Dates").

Additionally, detailed information on date calibration will be included in Appendix B (B1-Dates): "Single radiocarbon dates for sites were calibrated using OxCal4.4 software (Ramsey, 2009), considering the INTCAL20 calibration curve (Reimer et al., 2020). For sites with multiple dates, Bayesian chronology modeling was performed using OxCal4.4. Dates were organised into a '*Sequence*,' and chronological information for each level was grouped into a single '*Phase*' with start and end '*boundaries*' to bracket each archaeological level."

Considering the error chronological margins and the limitation into a straightforward correlation with a single climatic stadial (GS) or interstadial (GI) we will delete the sentence "The chronological resolution in the study areas for this period allows us to correlate regional paleoenvironmental changes with global records".

Concerning formal corrections for Figure 2, we will specify that grey bands indicate Greenland Stadials (GS), and, as previously addressed with reviewer 1, the green color denotes the Mediterranean area. Furthermore, it will be emphasised that detailed chronological information shown in thiis figure is available in Appendix B. In the figure, dots represent the mean value of the dates, while bars indicate the minimum and maximum values, considering a 95% interval. This clarification will be included in the figure caption.

*Methodology.*

*1. Authors mentioned that they did not carry out **any pre-treatment to remove secondary carbonates**, but did authors check the **potential presence of secondary carbonates or the preservation of carbonates**? This is very important since all the results are based on these values (there is no data of d18O in phosphates). The physical cleaning that was carried out would not remove all the potential secondary carbonates. Secondary carbonates are very common in archeological contexts such as karstic caves (like the ones studied here), and would modify the isotopic composition of carbonates if they are not eliminated. **This must be double checked before stating the sample preparation. Authors did not mention methods to double checked that the isotopic signal was the pristine one.** They only mentioned in line 503-504: "The carbonate content in our samples, ranging from 3.9% to 8.9%, is similar to the proportion found in modern tooth enamel, suggesting no immediate indication of diagenetic alteration". **However, there is no explanation about the methodology used to calculate this percentage of carbonate.** Authors should explain this in the methodology section and add the % of carbonate in a table (e.g. Table 2).*

Based on our experience, we have taken into account that the potential side effects resulting from pretreatment and the removal of carbonates can be more problematic than the risks associated with the presence of secondary carbonates. Calcium carbonate content, as briefly stated in the original manuscript, does not indicate a substantial presence of secondary carbonates, and we used this as a diagenetic test in line with recently published recommendations (Chesson et al. 2021). Additionally, phosphate data has previously been generated for a subset of samples at the Axlor site; these data show d18Ocarb-d18Ophos offsets within the expected range for all samples (Pederzani et al. 2023). We, therefore, consider the risk posed by secondary carbonates to be low. And while additional analyses such as FTIR may give more clarity on this, we believe it essential to acknowledge that no pretreatment method can guarantee their complete removal and that all removal methods, in turn, have the potential to introduce significant undesirable side effects, primarily on d18O (Pellegrini and Snoeck, 2016; Snoeck and Pellegrini, 2015). In addition, phosphate analyses were already in progress during the submission of this paper to double-check this issue and the results will be discussed in future works.

We recognised that secondary carbonates and diagenesis tests, to inform the choice of pretreatment methods, were not discussed as clearly as they could have been. In the methods section we will add a more explicit discussion of these points to justify better the methodological approach taken in the study. The carbonate content (%) was already included in Appendix B (B3. Samples raw), but the mean value will be included in Table 2 in the manuscript, as suggested. In agreement with the explanation of Iso-Analytical laboratory," the equivalent calcium carbonate content values (%) were derived by comparing the total ion beam data for the samples against the pure calcium carbonate references". We will also include this explanation in Appendix B (B3-Samples-Raw).

*2. Regarding the potential treatment to remove the organic matter, authors said: Lines 145-151: "For this reason, in this work, most of the samples were not pretreated, except for the equid samples from Labeko Koba and Aitzbitarte III, and the cervids and equids from El Otero that were sampled and pretreated in the context of the initial project. Pretreatment followed was established by Balasse et al. (2002), where around 7 mg of powdered enamel was prepared and pretreated with 3% of sodium hypochlorite (NaOCl) at room temperature for 24 h (0.1 ml/mg sample), and thoroughly rinsed with deionised water, before a reaction with 0.1M acetic acid for 4 h (0.1 ml/mg sample) (equivalent protocol in Jones et al., 2019)." And afterwards in lines 453-456: "In the case of equid samples from the Vasco-Cantabrian region, it should be considered that they have been pretreated with a combination of NaClO and acetic acid, which could potentially affect the isotopic values. Samples after organic removal pretreatment can potentially show either higher or lower δ13C values and higher δ18O values based on previous experiments (Pellegrini and Snoeck, 2016; Snoeck and Pellegrini, 2015)". So, my doubt is:*

*why did authors treat whole batches of samples to evaluate these "side effects" instead of applying both protocols (with pretreatment and without pretreatment) to aliquots of the same samples? Although they finally ended up that "the influence of the pre-treatment on our samples is deemed to be limited.", this was a risk, and now they cannot be 100% sure about this potential influence. Was there any reason to measure the samples where the organic matter was removed in a different IRMS. If yes, please, explain.*

The samples presented in this work have been collected over extensive research conducted by the EvoAdapta group. In 2013 we established a collaboration with the University of Cambridge, where pretreatment and IRMS measurements on the horse teeth from Cantabrian sites were achieved. At that time, pretreatment methods were not as methodologically questioned as today.

Some bovine tooth enamel powder samples were also collected in the respective museums during the initial research phase. However, the availability of these bovine teeth leftovers was limited then. Within the scope of the new ERC SUBSILIENCE project, given the new body of knowledge, it was determined that pretreatment initially done at Cambridge might significantly compromise the final results and thus, we decided not to pretreat the samples.

*3. Line 182-196 and throughout the calculations and the manuscript: Authors referred to E\* as the fractionation factor. The symbol E is traditionally the enrichment factor, not the fractionation factor (alpha). E= (alpha-1)x1000‰. So, what factor have authors applied eventually: fractionation or enrichment factors? Are these factors mixed in the text and in the calculations? This has to be clear, and if the factors are wrongly applied (enrichment instead of fractionation factor), correct the calculations. Although both factor are related (E= (alpha-1)x1000‰.) the obtained results would differ, and thus, the derived potential interpretations.*

This confusion is strictly terminological between fractionation and enrichment factors. As the reviewer suggests and in agreement with the specified references (Merceron et al., 2021; Passey et al., 2005; Tejada-Lara et al., 2018), "ℇ\*" should refer to isotope enrichment or enrichment factor, hence not causing any issue in interpretation or calculations. This term will be rectified throughout the manuscript.

*4. Lecuyer et al. (2021) performed the calculations to correct the effect of atmospheric CO2 (difference of 1‰ and CO2 concentration) for the LGM; so, these specific CO2 corrections can only be applied for the LGM, but in the present manuscript there are no samples for the LGM (23-19 ka). In addition, the correction for the isotopic composition of atmospheric CO2 should be done specifically for each age of the studied samples, instead of using a general average -7‰: as authors mentioned, a variation of a ca. 1‰ would imply a change in 150mm of precipitation. Check for example the isotopic composition of atmospheric CO2 reconstructed from ice records for the Late Pleistocene that ranges from -7 to -6.5 ‰ (Eggleston et al., 2016: Paleoceanography, among others). So, please, apply age-specific CO2 corrections. I mean, when you are quantifying climatic variables, you should reduce the potential error sources. These errors increase by applying a general unspecific correction for all the data (the same affirmation could be applied to all corrections of the isotopic data in the manuscript).*

We agree with the reviewer that atmospheric $CO_2$ levels varied throughout the Late Pleistocene, and ideally, adjustments should be more specific for each age. In our study, the primary aim of this correction is to mitigate the bias introduced by modern atmospheric $CO_2$ levels in the correlation. We are not entirely certain that applying the same average correction to all samples implies a higher risk than using a specific value based on chronology. This is because inaccuracies in these calculations could potentially introduce more unpredictable effects into the final interpretation, especially when dealing with imprecise chronological information.

We should assume, as Lécuyer et al. (2021) expressed, "that only broad estimates may be obtained" or "our estimates of MAP from d13C-data must be seen as a first approximation that needs to be confronted with estimates from other approaches". However, as the reviewer argues, 0.5‰ variations in $CO_2$ levels will suppose less than -100mm. In Table 1 from Leunberger et al. (1992) or in Eggleston et al. (2016) it can be appreciated that fluctuations are less than 0.5‰. Uncertainties associated with the estimated MAP values are on the order of±100-200 mm, and this associated error will be included.

In any case, our intention is not to propose an absolute MAP value, but to obtain comparative values on an understandable scale for assessing variations along a sequence or between deposits. It is important to acknowledge that uncertainties, as suggested by the reviewer, must be considered in the final value. We will introduce these appreciations in the text and the associated error.

*5. Authors proposed the **above-mentioned (oversimplified) correction for the change in the isotopic composition atmospheric CO2**, but they did not apply any **correction related to the change of the isotopic composition of the sea-water during the Late Pleistocene**, which is the main moisture source for rainfall (I would not mention the moisture and precipitation due to inland evaporation/recycling in the Iberian Peninsula, even during cold periods (Krklec and Domínguez-Villar, 2014), in order to simplify this interpretation). The global isotopic composition of the rain during colder/warmer periods (glaciar/interglaciar, stadial/interstadial) differs, not only due to the isotopic fractionation caused by temperatures, but also due to the **accumulation/release of the lighter water isotopes in the ice sheets/glaciers during cold/warmer periods** (Dansgaard, 1964), among others factors affecting the global isotopic composition of sea waters. Therefore, in order to obtain a reliable isotopic data related to precipitation, the obtained dO18 values has to be corrected to remove this effect. See for example, **Niedermeyer et al. (2010) or Garcia-Alix et al. (2021) approach to correct past hydrogen isotopes from vegetation, or even Fernández-García et al (2020) for fossil mammals** in the studied period of this paper.*

We are aware of this possible bias derived from variations in d18O, mainly between glacial and interglacial periods. However, as previously stated, considering the high chronological uncertainty within the dates, we considered it riskier to correct the results in one or another direction randomly. Nevertheless, as this can be a potential source of uncertainty, this explanation will be incorporated in the manuscript at the end of the section "3.4 Oxygen stable isotope compositions as environmental tracers". From our point of view, ideally, maintaining a consistent value for both corrections would be preferable to address the significant modern bias from the Holocene period. Therefore, we concur with the reviewer for a correction. However, considering the chronological uncertainties, we find it more consistent to apply a general correction specifically for the MIS3 period of -0.6‰ from the obtained d18Omw, as previously proposed by Fernández-García et al. (2019).

*6. Where did these obligate drinkers drink (water source)? Directly from the rain? Ponds? Lakes? Unless they directly drink precipitation waters (oversimplification), this would imply more isotopic fractionation and would also mask the temperature signal. This is even more important in the studied glacial period, and especially in the stadials? Apart from the potential enhanced rain evaporation due to low atmospheric moisture in glacial times (Dansgaard 1964), an increasing evaporation in lakes, ponds (and even in vegetation->enhanced evapotranspiration) during dry periods have been demonstrated by different isotopic studies in carbonates from freshwater gastropods, bivalves, or ostracods and even from leaf wax isotopes of freshwater and terrestrial plants. Please clarify this issue, and **explain this constrain in the methodology** since it would affect the reconstructed temperatures.*

Animals will likely acquire water from various types of water sources, with some sources, such as small streams, better reflecting seasonal isotopic oscillations than others, like large lakes or evaporating ponds. Therefore, we agree with the reviewer that certain types of drinking behaviours can impact d18O of the animals, and we have considered these concerns in our interpretation. We will provide a more detailed explanation in the methods and discussion sections, particularly about the state-of-the-art insights outlined by Pederzani and Britton, (2019).

In our study, we argue that, in animal teeth where the obtained profile is sinusoidal and has a sufficiently large range (<1.5-2), predominant seasonal water information is recorded, related to the predominant consumption of water sources that reflects seasonal precipitation reasonably well. However, it is evident that systematic consumption of certain highly buffered water sources can significantly attenuate the final signal recorded. We will discuss this possibility in more depth in the paper.

Regarding enhanced evaporation during more arid stadials, we consider this effect relatively small regarding oxygen isotope data from large bovids sampled in this study. These animals heavily rely on consuming large amounts of liquid drinking water, and the relative contribution of water from plants, which, as the reviewer mentions, is most susceptible to isotopic enrichment through evapotranspiration, is negligible (Hoppe, 2006; Maloiy, 1973). Other water sources strongly affected by evaporative enrichment are large rivers and lakes with long water residence times where a systematic consumption of such waters would lead to an absence of a seasonal sinusoidal d18O signal. However, this is not systematically observed in conjunction with the higher d18O that would be expected in such evaporated water bodies.

Finally, if aridity effects significantly drove d18O, we would expect a positive correlation between d18O and d13C, which also responds to aridity effects, but we do not see such correlation in our data. However, we agree that non-temperature effects on d18O are critical, so we will add the points raised here to the paper's discussion.

*7. **Precipitation source** (North Atlantic Oscillation modes and Mediterranean dynamics). In the Iberian Peninsula, the isotopic composition of precipitation is **highly affected by the moisture source in the present** - and in the past - (Araguás y Díaz Teijeiro, 2005, Celle-Jeanton et al., 2001; Domínguez-Villar et al., 2013; Krklec and Domínguez-Villar, 2014; Moreno et al., 2010; 2012; 2014; 2021, Toney et al., 2020; García-Alix et al., 2021; Schirrmacher et al., 2020, among others). Therefore, temperature-isotope fractionation equations would not work that well to reconstruct past temperatures. When proper analysis of the isotopic signal of precipitation are performed in N Iberia, **there are sampling stations (precipitation, temperatures, isotopes, and moisture sources) where the isotopes from precipitation do not correlate well with temperatures** (Moreno et al., 2021). This thoroughly study of the isotopic composition of precipitation in northern Spain ended up with "although important, air temperature only explains part of the observed $\delta^{18}O_p$ variability and is therefore not the only control." This issue is even more important in coastal areas, as the ones studied in this paper, and even more in the Mediterranean coast (at present there is some influence of amount effect in the Mediterranean areas of Catalonia; Moreno et al., 2021). Therefore, in the best oversimplified case-scenario (not considering previous comments 5 and 6, and admitting large errors due to these potential source and amount effects) **if we would want to calculate temperatures from the isotopic values of precipitation we would need an equation for the Vasco-Cantabrian region, and another one for NE-Mediterranean Iberia** (different precipitation pattern and forcing). And even more, since atmospheric patterns, and therefore, moisture sources, are not the same during the warm and cold seasons (see the above-mentioned studies), specific equations for cold and warm seasons should be applied to reconstruct "summer" and "winter" temperatures.*

We appreciate the reviewer's concern about differences in the slope of the d18Oprecip-temperature relationship, which is known to vary geographically, seasonally and through time due to differences in moisture source and atmospheric circulation patterns (Dansgaard, 1964; Gat, 1980; Rozanski et al., 1993).

There are essentially two approaches to account for these differences. The first approach, as described by the reviewer, attempts to specify as accurately as possible the particular d18Oprecip-temperature relationship for the study area. This approach usually yields more precise estimates. The second approach instead posits that creating a precisely accurate d18Oprecip-temperature equation for a specific location is extremely difficult as data is not directly available for the areas in question, but often for rather distant measurement stations and because we are additionally projecting into the past where atmospheric circulation patterns were not identical to what they are now.

In this second approach, the inherent variability and uncertainty of the d18Oprecip-temperature relationship is instead represented statistically, and the resulting uncertainty is incorporated into the error range of the temperature estimate by including a large number of data from different measurement stations into the d18Oprecip-temperature calibration data set. This approach yields less precise data but reduces the risk of systematically inaccurate results. In this study, we consider that due to the projection of equations into the past, the second approach is preferable. Still, we acknowledge that the decision between different methods remains debated in the field, with some researchers choosing the closest measurement station, a mixture of stations from one country, or even globally combined data sets (see e.g. Skrzypek et al. 2011; Pryor et al. 2014; Tütken et al. 2007; Lécuyer et al. 2021). We will justify this decision in more detail concerning the literature cited here in the methods section of our paper.

 *8. "MAT was calculated from the d18O mean value between summer and winter in each tooth before modeling to reduce associated error"; However in caption Table 4: "For some profiles with an unclear seasonal shape, MATs were deduced from the original average of teeth without a seasonal profile". So, what is the correct methodology?, In any case, according to the methodology section, MAT was calculated before modeling to reduce associated error, but summer and winter temperatures after the inverse modelling?. This reasoning is not clear to me: Is there no associated error in the transformation for summer and winter temperatures? This is why there* **are some odd values, for example, sample AXL60 MAT 12ºC, ST 20.4ºC and WT 10.8ºC (only 1.2ºC colder than the MAT).**

We appreciate that the reason for treating MAT differently from summer and winter temperature estimation was not as clearly explained as it could have been. Due to the influence of the prolonged enamel mineralisation process, the seasonal amplitude of d18O derived from sequentially sampled teeth necessarily has a lower measured amplitude than the real d18O seasonal variation in the drinking water consumed by the animal. This effect is corrected as much as possible by the inverse modeling procedure. Therefore it is imperative that summer and winter temperature estimations can be made on the output values of the inverse model rather than directly on measure d18O data, to avoid systematic underestimation of the seasonal temperature change. However, the same is not necessarily true for mean annual values, as the inverse model will essentially symmetrically expand the d18O seasonal curve, introducing no substantial changes to the data annual means. A systematic comparison of d18O annual means derived from unmodeled vs modeled curves in Pederzani et al. 2021 showed that annual means do not change between these different ways of obtaining d18O annual means for input into a temperature estimation. In this case, it is preferable to derive mean annual temperature estimation directly from d18Oenamel annual means, because the inverse model introduces significant uncertainty, which would also impact any derived temperature estimation. We therefore consider the approach of using unmodeled annual means, but modeled summer and winter d18O as the input for temperature

estimation to be the more robust method. We will clarify this difference in the text of the paper with reference to Pederzani et al. 2021 where the two approaches are compared in detail.

Regarding the use of different averaging methods for obtaining d18O annual averages, an in-depth comparison in Pederzani et al. 2021 also showed that the results of either using a summer-winter two-point average or an average of all data points across an entire year yield indistinguishable results. Using a two-point average commonly has the advantage that it allows the calculation of an annual mean for teeth that do not record complete annual cycles but do record both a summer peak and a winter trough in d18O. This method, therefore, allows us to maximise the number of data points that can be obtained for MAT estimation. Still, it is not possible to apply this method to specimens that do not show a clear sinusoidal d18O signal, so we chose to average all data points for those specimens instead. This last option, however, may be less reliable, and we will review the text to clarify these uncertainties.

*9. The reconstructed meteoric waters are different depending on the species, even in the same level, and therefore, reconstructed temperatures also differ. I'm aware of the different ecological behaviors of the different species, but the MAT should be, at least close. There are also some discrepancies in specimens of the same species in the same archaeological levels.*

The interspecies variability in temperature reconstructions is not significantly higher than the intraspecies variability, as demonstrated in Table 4, where estimations are within error for all species. Multiple factors could account for differences for both within the same species and between species. Variability in d18O, and consequently in derived temperature estimations, is expected even within the same species or among individuals from the same population. This variability is influenced not only by minor differences in foraging and drinking behavior but also by slight metabolic and physiological variations. The relationship between d18Odw and d18O of body water and enamel is affected by factors such as body size, metabolic rate, breathing rate, moisture content of food, and feces, among others (Kohn et al. 1996; Magozzi et al. 2019). It is important to note that appreciable variations in d18O have been observed even within small groups of animals living in the same constrained habitat (e.g., Magozzi et al. 2020; Hoppe et al. 2004). Additionally, it should be considered that archaeological levels represent palimpsests of occupations from different periods, which further increases the expected inter-individual variability observed in d18O. To provide more context for interpreting the slight differences observed among other species and individuals, we will include a note in the manuscript to convey the sentiments expressed in this response.

*10. The general comparison between the isotopic composition of the faunas of the archaeological sites of both areas is not objective since in the Vasco-Cantabrian area there are remains from 80 ka to 18 ka approx. (14 sections from 5 archaeological sites), but in the Catalonian area there is only one site at around 40 ka (it could have been coeval to the HS4, a period especially cold and dry). Thus, the general comparisons that appeared in some parts of the manuscript between two areas (not the ones specific at ca. 40 ka) are not balanced.*

We agree with the reviewer that there is a disbalance between both northern Iberian regions. The lack of data in the Mediterranean region produce this difference. So far, it is the only site with these type of carbonate enamel analysis in the contemporaneous archaeological sites of the Mediterranean. Nevertheselss, we considered interesting to include the Canyars results as it is well-dated archaeological site in a precise climatic event such as the Heinrich 4. For this reason, we make this evaluation to contrast with the wide-studied cantabrian region.

*11. What is the **error associated** with the different equations Eq. 1-9? Authors have to take into account the different errors (not only the standard deviations) that are being accumulated in each equation and also plot and mention them in the tables and in the text. For example, a MAT of 15ºC +/- 0.5 ºC would be accurate, but if the error was +/- 4ºC the interpretation would be more open. This is even more important **in this work since the oxygen isotopes were measured in carbonates, not in phosphates, and two more equations were used to converter the d18O data of the carbonates to d18O of phosphates.** Authors said: lines 277-278: "In these estimations, the associated error from converting δ18Ophos to MAT is enlarged by the uncertainty derived from the transformation of δ18Ocarb (VPDB) to δ18Ophos (VSMOW)". Therefore, all the errors should be calculated (for both O and C), included in the plots, tables and when describing the results.*

The authors fully agree with the reviewer and this issue will be solved. Associated errors will be calculated following the methodology outlined by Pryor et al. (2014), as done by one of us in Pederzani et al. (2021). We will add this information to the manuscript with a short explanation of the error calculation method, which integrates uncertainties from the consecutive conversion steps involved in the temperature estimation process. The error calculation will be included in Appendix B and the associated error will be placed in TAble 4 and within the text.

*12. Inverse Modelling. I think that the correct reference for the inverse modeling (main text and appendix D) is Passey et al. (2005) but this one "Passey, B.H., Cerling, T.E., Schuster, G.T., Robinson, T.F., Roeder, B.L., Krueger, S.K., 2005. Inverse methods for estimating primary input signals from time-averaged isotope profiles. Geochim. Cosmochim. Acta 69, 4101–4116." Reviewing this paper, I noted that the **reconstructed profiles showed mostly the same trends/changes as the original isotopic data. However, in the case of the reviewed manuscript sometimes these reconstructed profiles exhibited opposite patterns/trends from the original isotopic values. I'm not familiar with these kinds of transformations, but: is this common?** This is an extra transformation for the data that would be used to calculate absolute climatic parameters, and thus, and extra potential error source. Could authors double check these calculations?. In addition, in the original paper from Pasey et al. (2005), this inverse modeling was also applied to the carbon isotopes. **Why authors did not apply this correction also to the carbon isotopes of the sequential sampling?***

Firstly, the reviewer is correct about the reference; it was mistakenly listed in the final reference section. It will be edited in the manuscript. Secondly, some of the teeth included in the manuscript, as elaborated within the methods and also discussed in the section "5.2 Seasonality, mobility, and water acquisition: oxygen ratios and intratooth profiles," exhibit flat and less sinusoidal profiles, which are less suitable for the application of this model as it presupposes an approximately sinusoidal shape of the seasonal d18O curve. This may result in non-straightforward interpretation in the model outcomes for non-sinusoidal curves. Additionally, small but abrupt 'noise' spikes in the original d18O curves tend to be amplified in the model output. They can sometimes represent the highest or lowest values globally in the modelled d18O curves. For this reason, we consistently selected peaks and troughs as summer and winter modelled d18O based on where these seasonal regions are located in the original profile. It should be noted that the model reproduces the lag between d18O change in the input to the animal compared to the expression in tooth enamel. Therefore, the modelled d18O curve of the model is constantly shifted in x-direction relative to the d18Oenamel input time series. This can sometimes lead to seemingly opposing isotopic changes being located in the same x-axis region, but this does not represent an actual reversal of the isotopic trends between the model and the original. Regarding, the uncertainty associated with these models, this is represented as a 95% confidence interval, which is presented in each figure in Appendix D as shaded grey areas, and they can also be found in Appendix B ("B6-Model extrema").

Additionally, as previously discussed concerning error calculation, the accumulated errors for winter and summer will be expressed in Table 4. Finally, regarding the question about applying the model to carbon values, we considered this to add little valuable insight in our particular case as the d13C time series of our specimens do not show appreciable seasonal change and do not conform to an approximately sinusoidal shape. This is due to the absence of a marked seasonal change in diet, or at least one that cannot be detected through carbon isotopic values, which is common in Eurasian individuals consuming C3 plants. In the absence of such seasonal d13C change, modelling of amplitude damping using the Passey et al. 2005 model would not be genuinely appropriate and would likely lead to issues with model convergence, so we did not attempt this. We will add a note to the manuscript to justify this decision.

***Minor comments:***

*The first paragraph in the introduction and the first lines in the abstract deal with the importance of these kinds of studies to understand the human evolution in this region, but eventually, this is not discussed in the manuscript according to the obtained data.*

When we refered to understand human evolution, we meant to reconstruct the climatic and environmental conditions that late Neanderthals and early modern human groups had to face in the regions of study during the Middle to Upper Paleolithic. We aimed with this study to propose the local conditions to later evaluate their adaptation strategies and resilience. We will clarify this aspect.

*First sentences of the introduction (lines 38-43): please add references. There are interesting papers dealing with this issue in the Iberian Peninsula and in Europe: Neanderthal-AMH-climate change.*

Relevant works such as D'Errico and Sánchez Goñi (2003), Finlayson and Carrión (2007), Sepulchre et al. (2007), and Staubwasser et al. (2018) will be cited but they will be updated with recent reviews such as Sánchez Goñi (2020), Timmermman (2020), Klein et al. (2023) or Vidal-Cordasco et al. (2022; 2023).

*Lines 49-50. It is ok, but authors are not including some information (and significant climate-related references) in the area dealing with continuous paleoclimatic records, and they only focused on the data obtained from archeological sites (whose paleoenvironmental record is not that continuous); so, line 75 is not summarizing the multiproxy studies in this area.*

The regional non-archaeological studies mentioned by the reviewer, as well as references to other non-archaeological regional records, will be incorporated into the introduction, as previously discussed.

*Lines 80-87: add references*

The references of Passey et al. (2002), Kohn et al. (2004), and Zazzo et al. (2005) will be included to support the discussion regarding isotopic signal preservation and enamel mineralization. On the other hand, references Pederzani & Britton (2019), Ambrose and Norr (1993), Luz et al. (1984), and Pryor et al. (2014) will be used as examples to justify the general interpretation of carbon and oxygen signals.

*Regarding the fossil sites that authors call mediterranean". Since the rest of the fossil sites are in "northeastern Iberia", the term "Mediterranean area" is very open and do not specifically identify the studied site, I would say NE Iberia?*

The authors will make the adjustment, replacing the term "Mediterranean area" with "northeastern Iberia" as suggested.

*There are some issues with the chronology/dates. For example, line 21 abstract: 80 to 15,000 cal BP. Do you mean 80 ka or ky, right? Taking into account the study period, and the accuracy of the dates, I would not use "yr", I would use ky or ka. In addition, authors should round the dates in the text to the nearest hundreds (eg, line 601: 41,136 to 38,570 cal yr BP: 41.1 ka to 38.6 ka). This accuracy does not make sense in the studied period due to the uncertainty of the measurements.*

To prevent any potential confusion, as recommended by the reviewer, all dates throughout the paper will be revised for consistency and expressed in thousands of years ago (ka cal BP).

*Results: please, add the ages (in ka and the different technocomplexes) to the subsection headings of the different archaeological sites, otherwise one has to check Fig. 2 for each site.*

Ages will be incorporated into each site subsection to establish a general chronological framework and facilitate reader comprehension: Axlor (ca. 80-50 cal ka BP), El Castillo (ca. 75-48 cal ka BP), Labeko Koba (ca. 42-35 ka cal BP), Canyars (ca. 40 ka cal BP), Aitzbitarte III (ca. 25 ka cal BP), and, El Otero (ca. 19 ka cal BP). Nevertheless, these time periods are approximate and serve as a general reference and it is advisable for readers to refer to Figure 2 for detailed information on each level.

*I understand the structure of the result description, but the mixed description of the isotopes from different levels of the same archaeological sites, which sometimes have 10 ka of difference between them, is rare to me.*

We believe that this is the most efficient way to present the results, as it takes into consideration the specific characteristics of each individual site, which at times can be influenced by baseline isotopic values.

*Line 325: MATAs=1-8/-2.1ºC? do you mean 1.8?*

Yes, we mean 1.8ºC.

*Eq 10: P value <?*

Yes, it was a mistake. We will correct it to "<".

*All figures with the data plotted according to the chronology. El Castillo 21A appears after Axlor III; but according to the chronology this site is previous (Fig. 2). The same would happen between some levels of Labeko Kova and Canyars-I. Is this correct? In this case, arrange the sites with the real chronological order even though they belong to different archaeological sections.*

We understand that this assessment primarily pertains to Figure 5, where our priority was to arrange the sequences in accordance with an internal diachronic order. While we acknowledge the importance of chronological coherence, for a better understanting, we will better maintain the separation by sites.

*When speaking about range of values (temperatures, isotopes, precipitation), sometimes the lowest values are mentioned before and other times the other way around. Be consistent and cite the lowest values first (Eg, line 404 MATAs).*

Thanks for the comments. We will be consistent in the final version.

*Table 3: I suppose that the different group of data (rows) are related to the three groups of specimens. Right? Please, specify the taxon groups in the tables.*

As the reviewer has pointed out, there is an error, and it is necessary to designate each taxon in the table, from top to bottom, including total, bovines, and equids.

*I do not understand this sentence. Line 510-511: "Based on these arguments, it is suggested that the non-sinusoidal $\delta 18O$ signal observed in some individuals is likely attributed to the preservation of the original isotopic signature from water input." This sentence would suggest a bad preservation of the original water composition when a sinusoidal pattern is present?. However, authors explained afterwards some reasons related to the ecology of the specimens. Please, clarify this.*

We argue that these non-sinusoidal profiles do not indicate poor preservation in the studied specimens. Instead, we propose that they may be linked to somehow ethological factors of the individuals, capturing an isotopic signature originating from non-seasonal water sources. Further elaboration on this point has been provided in response to previous questions 6 and 9.

*Section 5.4. Regarding the d18O of meteoric waters. Authors compare (as a whole) their reconstructed d18Omw with the current values in the area and they ended up with similar ranges. Keep in mind that the reconstructed values correspond to the last 80 ka under glacial conditions (different temperatures, precipitation amount, and in some cases, moisture source)*

Our comparisons usually concentrate on assessing the range of values without drawing broader inferences. We agree with this consideration, and as discussed in our previous response to the reviewer, we will incorporate these points into the text.

*Sometimes the different d13C and d18O have subscripts with the meaning of the isotopic values (eg, $\delta 13Ccarb$, $\delta 18Ocarb$), but another times this is not indicated. This is confusing. Please, add always the subscripts explaining the meaning of the isotopic values.*

We will correct the text to ensure that "d13C" and "d18O" always include subscripts to maintain consistency.

*I miss a figure summarizing the chronological evolution of reconstructed temperatures and precipitation (with the associated accumulated errors) and comparing them with other continuous records of precipitation or temperatures or vegetation from the Iberian Peninsula, Mediterranean coast or Iberian margin.*

We agree that such a figure would be valuable for this study. However, we are presently in the process of refining and expanding the dataset for a more comprehensive regional comparison with phosphate analysis. As suggested in the paper and, as previously noted by the reviewer, the estimation of paleotemperatures was approached tentatively taking into account the uncertainties arising from starting with the carbonate fraction. For this reason, more reliable correlations between our snapshots and continuous regional records will be provided in future works.

*Title: ecological evolution of what? This is very general: I would say ecological evolution of ungulate fauna….*

Considering suggestion of the reviewer, we propose: "Ecological evolution of ungulates in northern Iberia during the Late Pleistocene through isotopic analysis on teeth"

*Please, use always the symbol delta instead of d, there are some "ds" throughout the manuscript.*

Thanks for the edits. We have identified this error occurring on three occasions and it will be addressed.

*Cited Literature:*

*Araguas-Araguas, L.J., Diaz Teijeiro, M.F., 2005. Isotope composition of precipitation and water vapour in the Iberian Peninsula. First results of the Spanish Network of Isotopes in Precipitation, in: Isotopic Composition of Precipitation in the Mediterranean Basin in Relation to Air Circulation Patterns and Climate. IAEA-TECDOC-1453, Vienna, pp. 173–190.*
*Camuera, J., Jim*
*énez-Moreno, G., Ramos-Román, M.J., García-Alix, A., Toney, J.L., Anderson, R.S., Jiménez-Espejo, F., Bright, J., Webster, C., Yanes, Y., Carrión, J.S., 2019. Vegetation and climate changes during the last two glacial-interglacial cycles in the western Mediterranean: A new long pollen record from Padul (southern Iberian Peninsula). Quat Sci Rev 205, 86–105.*
*Camuera, J., Ramos-Román, M.J., Jiménez-Moreno, G., García-Alix, A., Ilvonen, L., Ruha, L., Gil-Romera, G., González-Sampériz, P., Seppä, H., 2022. Past 200 kyr hydroclimate variability in the western Mediterranean and its connection to the African Humid Periods. Sci Rep 12, 9050.*
*Celle-Jeanton, H., Travi, Y., Blavoux, B., 2001. Isotopic typology of the precipitation in the Western Mediterranean Region at three different time scales. Geophys Res Lett 28, 1215–1218.*
*Dansgaard, W., 1964. Stable isotopes in precipitation. Tellus 16, 436–468.*
*Domínguez-Villar, D., Carrasco, R.M., Pedraza, J., Cheng, H., Edwards, R.L., Willenbring, J.K., 2013. Early maximum extent of paleoglaciers from Mediterranean mountains during the last glaciation. Sci Rep 3, 2034.*
*Eggleston, S., Schmitt, J., Bereiter, B., Schneider, R., Fischer, H., 2016. Evolution of the stable carbon isotope composition of atmospheric CO2 over the last glacial cycle. Paleoceanography 31, 434–452. https://doi.org/https://doi.org/10.1002/2015PA002874*
*Fernández-García, M., López-García, J. M., Royer, A., Lécuyer, C., Allué, E., Burjachs, F., Chacón, M. G., Saladié, P., Vallverdú, J., and Carbonell, E.,2020. Combined palaeoecological methods using small-mammal assemblages*

to decipher environmental context of a long-term Neanderthal settlement in northeastern Iberia, Quat. Sci. Rev., 228, 106072

García-Alix, A., Camuera, J., Ramos-Román, M.J., Toney, J.L., Sachse, D., Schefuß, E., Jiménez-Moreno, G., Jiménez-Espejo, F.J., López-Avilés, A., Anderson, R.S., Yanes, Y., 2021. Paleohydrological dynamics in the Western Mediterranean during the last glacial cycle. Glob Planet Change 202, 103527.

González-Sampériz, P., Gil-Romera, G., García-Prieto, E., Aranbarri, J., Moreno, A., Morellón, M., Sevilla-Callejo, M., Leunda, M., Santos, L., Franco-Múgica, F., Andrade, A., Carrión, J.S., Valero-Garcés, B.L., 2020. Strong continentality and effective moisture drove unforeseen vegetation dynamics since the last interglacial at inland Mediterranean areas: The Villarquemado sequence in NE Iberia. Quat. Sci. Rev. 242, 106425.

Krklec, K., Domínguez-Villar, D., 2014. Quantification of the impact of moisture source regions on the oxygen isotope composition of precipitation over Eagle Cave, central Spain. Geochim Cosmochim Acta 134, 39–54.

Lécuyer, C., Hillaire-Marcel, C., Burke, A., Julien, M.-A., and Hélie, J.-F., 2021.Temperature and precipitation regime in LGM human refugia of southwestern Europe inferred from δ13C and δ18O of large mammal remains, Quat. Sci. Rev., 255, 106796,

Martrat, B., Grimalt, J.O., Lopez-Martinez, C., Cacho, I., Sierro, F.J., Flores, J.A., Zahn, R., Canals, M., Curtis, J.H., Hodell, D.A., 2004. Abrupt Temperature Changes in the Western Mediterranean over the past 250,000 Years. Science (1979) 306, 1762–1765.

Martrat, B., Grimalt, J.O., Shackleton, N.J., Abreu Lucia de, Hutterli, M.A., Stocker, T.F., 2007. Four Climate Cycles of Recurring Deep and Surface Water Destabilizations on the Iberian Margin. Science (1979) 317, 502–507.

Moreno, A., Stoll, H., Jiménez-Sánchez, M., Cacho, I., Valero-Garcés, B., Ito, E., Edwards, R.L., 2010. A speleothem record of glacial (25–11.6kyr BP) rapid climatic changes from northern Iberian Peninsula. Global and Planetary Change 71, 218-231.

Moreno, A., González-Sampériz, P., Morellón, M., Valero-Garcés, B.L., Fletcher, W.J., 2012. Northern Iberian abrupt climate change dynamics during the last glacial cycle: A view from lacustrine sediments. Quat Sci Rev 36, 139–153.

Moreno, A., Sancho, C., Bartolomé, M., Oliva-Urcia, B., Delgado-Huertas, A., Estrela, M.J., Corell, D., López-Moreno, J.I., Cacho, I., 2014. Climate controls on rainfall isotopes and their effects on cave drip water and speleothem growth: the case of Molinos cave (Teruel, NE Spain). Clim Dyn 43, 221–241.

Moreno, A., Iglesias, M., Azorin-Molina, C., Pérez-Mejias, C., Bartolomé, M., Sancho, C., Stoll, H., Cacho, I., Frigola, J., Osácar, C., Muñoz, A., Delgado-Huertas, A., Bladé, I., Vimeux, F., 2021. Measurement report: Spatial variability of northern Iberian rainfall stable isotope values -- investigating atmospheric controls on daily and monthly timescales. Atmos. Chem. Phys. 21, 10159–10177.

Niedermeyer, E.M., Schefuß, E., Sessions, A.L., Mulitza, S., Mollenhauer, G., Schulz, M., Wefer, G., 2010. Orbital- and millennial-scale changes in the hydrologic cycle and vegetation in the western African Sahel: insights from individual plant wax δD and δ13C. Quat Sci Rev 29, 2996–3005.

Pérez-Mejías, C., Moreno, A., Sancho, C., Martín-García, R., Spötl, C., Cacho, I., Cheng, H., Edwards, R.L., 2019. Orbital-to-millennial scale climate variability during Marine Isotope Stages 5 to 3 in northeast Iberia. Quat. Sci. Rev. 224, 105946.

Schirrmacher, J., Andersen, N., Schneider, R.R., Weinelt, M., 2020. Fossil leaf wax hydrogen isotopes reveal variability of Atlantic and Mediterranean climate forcing on the southeast Iberian Peninsula between 6000 to 3000 cal. BP. PLoS One 15, e0243662.

Toney, J.L., García-Alix, A., Jiménez-Moreno, G., Anderson, R.S., Moossen, H., Seki, O., 2020. New insights into Holocene hydrology and temperature from lipid biomarkers in western Mediterranean alpine wetlands. Quat Sci Rev 240.

References for reviews (different from those indicated by the reviewer):

Ambrose, S.H., Norr, L., 1993. Experimental Evidence for the Relationship of the Carbon Isotope Ratios of Whole Diet and Dietary Protein to Those of Bone Collagen and Carbonate, in: Prehistoric Human Bone. Springer Berlin Heidelberg, Berlin, Heidelberg, pp. 1–37. https://doi.org/10.1007/978-3-662-02894-0_1

Chesson, L.A., Beasley, M.M., Bartelink, E.J., Jans, M.M.E., Berg, G.E., 2021. Using bone bioapatite yield for quality control in stable isotope analysis applications. Journal of Archaeological Science: Reports 35, 102749. https://doi.org/10.1016/j.jasrep.2020.102749

D'Errico, F., Sánchez Goñi, M.F., 2003. Neandertal extinction and the millennial scale climatic variability of OIS 3. Quaternary Science Reviews 22, 769–788. https://doi.org/10.1016/S0277-3791(03)00009-X

Dansgaard, W., 1964. Stable isotopes in precipitation. Tellus XVI, 436–468.

Gat, J.R., 1980. The relationship between surface and subsurface waters: water quality aspects in areas of low precipitation. Hydrological Sciences-Butlletin des Sciences Hydrologiques 25, 257–267.

Fernández-García, M., Vidal-Cordasco, M., Jones, J.R., Marín-Arroyo, A.B., 2023. Reassessing palaeoenvironmental conditions during the Middle to Upper Palaeolithic transition in the Cantabrian region (Southwestern Europe). Quaternary Science Reviews 301, 107928. https://doi.org/10.1016/j.quascirev.2022.107928

Finlayson, C., Carrión, J.S., 2007. Rapid ecological turnover and its impact on Neanderthal and other human populations. Trends in Ecology and Evolution 22, 213–222. https://doi.org/10.1016/j.tree.2007.02.001

Hoppe, K.A., Amundson, R., Vavra, M., McClaran, M.P., Anderson, D.L., 2004. Isotopic analysis of tooth enamel carbonate from modern North American feral horses: implications for paleoenvironmental reconstructions. Palaeogeography, Palaeoclimatology, Palaeoecology 203, 299–311. https://doi.org/10.1016/S0031-0182(03)00688-6

Hoppe, K.A., 2006. Correlation between the oxygen isotope ratio of North American bison teeth and local waters: Implication for paleoclimatic reconstructions. Earth and Planetary Science Letters 244, 408–417. https://doi.org/10.1016/j.epsl.2006.01.062

Kohn, M.J., 2004. Comment: tooth enamel mineralization in ungulates: implications for recovering a primary isotopic time-series. In: Passey, B.H., Cerling, T.E. (Eds.), (2002). Geochim. Cosmochim. Acta. vol. 68. pp. 403–405. https://doi.org/10.1016/ S0016-7037(03)00443-5.

Kohn, M.J., 1996. Predicting animal $\delta^{18}O$: Accounting for diet and physiological adaptation. Geochimica et Cosmochimica Acta 60, 4811–4829. https://doi.org/10.1016/S0016-7037(96)00240-2

Klein, K., Weniger, G., Ludwig, P., Stepanek, C., Zhang, X., Wegener, C., Shao, Y. 2023.Assessing climatic impact on transition from Neanderthal to anatomically modern human population on Iberian Peninsula: a macroscopic perspective. Science Bulletin, 68 (11), 1176-1186.

Lécuyer, C., Hillaire-Marcel, C., Burke, A., Julien, M.-A., Hélie, J.-F., 2021. Temperature and precipitation regime in LGM human refugia of southwestern Europe inferred from $\delta^{13}C$ and $\delta^{18}O$ of large mammal remains. Quaternary Science Reviews 255, 106796. https://doi.org/10.1016/j.quascirev.2021.106796

Luz, B., Kolodny, Y., Horowitz, M., 1984. Fractionation of oxygen isotopes between mammalian. Geochimica et Cosmochimica Acta 48, 1689–1693.

Magozzi, S., Vander Zanden, H.B., Wunder, M.B., Bowen, G.J., 2019. Mechanistic model predicts tissue–environment relationships and trophic shifts in animal hydrogen and oxygen isotope ratios. Oecologia 191, 777–789. https://doi.org/10.1007/s00442-019-04532-8

Magozzi, S., Vander Zanden, H.B., Wunder, M.B., Trueman, C.N., Pinney, K., Peers, D., Dennison, P.E., Horns, J.J., Şekercioğlu, Ç.H., Bowen, G.J., 2020. Combining Models of Environment, Behavior, and Physiology to Predict Tissue Hydrogen and Oxygen Isotope Variance Among Individual Terrestrial Animals. Frontiers in Ecology and Evolution 8. https://doi.org/10.3389/fevo.2020.536109

Maloiy, G.M.O., 1973. Water metabolism of East African ruminants in arid and semi-arid regions 1. Zeitschrift für Tierzüchtung und Züchtungsbiologie 90, 219–228. https://doi.org/10.1111/j.1439-0388.1973.tb01443.x

Merceron, G., Berlioz, E., Vonhof, H., Green, D., Garel, M., Tütken, T., 2021. Tooth tales told by dental diet proxies: An alpine community of sympatric ruminants as a model to decipher the ecology of fossil fauna. Palaeogeography, Palaeoclimatology, Palaeoecology 562, 110077. https://doi.org/10.1016/j.palaeo.2020.110077

Naughton, F., Sánchez-Goñi, M.F., Desprat, S., Turon, J.-L., Duprat, J., 2007. Present-day and past (last 25 000 years) marine pollen signal off western Iberia. Marine micropaleontology 62, 91–114. https://doi.org/10.1016/j.marmicro.2006.07.006

Passey, B.H., Robinson, T.F., Ayliffe, L.K., Cerling, T.E., Sponheimer, M., Dearing, M.D., Roeder, B.L., Ehleringer, J.R., 2005. Carbon isotope fractionation between diet, breath CO2, and bioapatite in different mammals. Journal of Archaeological Science 32, 1459–1470. https://doi.org/10.1016/j.jas.2005.03.015

Passey, B.H., Cerling, T.E., 2002. Tooth enamel mineralization in ungulates: implications for recovering a primary isotopic time-series. Geochim. Cosmochim. Acta 66, 3225–3234

Pellegrini, M., Snoeck, C., 2016. Comparing bioapatite carbonate pre-treatments for isotopic measurements: Part 2 — Impact on carbon and oxygen isotope compositions. Chemical Geology 420, 88–96. https://doi.org/10.1016/j.chemgeo.2015.10.038

Pederzani, S., Britton, K., 2019. Oxygen isotopes in bioarchaeology: Principles and applications, challenges and opportunities. Earth-Science Reviews 188, 77–107. https://doi.org/10.1016/j.earscirev.2018.11.005

Pederzani, S., Britton, K., Aldeias, V., Bourgon, N., Fewlass, H., Lauer, T., McPherron, S.P., Rezek, Z., Sirakov, N., Smith, G.M., Spasov, R., Tran, N.H., Tsanova, T., Hublin, J.J., 2021. Subarctic climate for the earliest Homo sapiens in Europe. Science Advances 7, 1–11. https://doi.org/10.1126/sciadv.abi4642

Pederzani, S., Britton, K., Jones, J.R., Agudo Pérez, L., Geiling, J.M., Marín-Arroyo, A.B., 2023. Late Pleistocene Neanderthal exploitation of stable and mosaic ecosystems in northern Iberia shown by multi-isotope evidence. Quaternary Research 1–25. https://doi.org/10.1017/qua.2023.32

Pryor, A.J.E., Stevens, R.E., Connell, T.C.O., Lister, J.R., 2014. Quantification and propagation of errors when converting vertebrate biomineral oxygen isotope data to temperature for palaeoclimate reconstruction. Palaeogeography, Palaeoclimatology, Palaeoecology 412, 99–107. https://doi.org/10.1016/j.palaeo.2014.07.003

Ramsey, C.B., 2009. Bayesian Analysis of Radiocarbon Dates. Radiocarbon 51, 337–360. https://doi.org/10.1017/S0033822200033865

Reimer, P.J., Austin, W.E.N., Bard, E., Bayliss, A., Blackwell, P.G., Bronk Ramsey, C., Butzin, M., Cheng, H., Edwards, R.L., Friedrich, M., Grootes, P.M., Guilderson, T.P., Hajdas, I., Heaton, T.J., Hogg, A.G., Hughen, K.A., Kromer, B., Manning, S.W., Muscheler, R., Palmer, J.G., Pearson, C., van der Plicht, J., Reimer, R.W.,

Richards, D.A., Scott, E.M., Southon, J.R., Turney, C.S.M., Wacker, L., Adolphi, F., Büntgen, U., Capano, M., Fahrni, S.M., Fogtmann-Schulz, A., Friedrich, R., Köhler, P., Kudsk, S., Miyake, F., Olsen, J., Reinig, F., Sakamoto, M., Sookdeo, A., Talamo, S., 2020. The IntCal20 Northern Hemisphere Radiocarbon Age Calibration Curve (0–55 cal kBP). Radiocarbon 62, 725–757. https://doi.org/10.1017/RDC.2020.41

Roucoux, K.H., Shackleton, N.J., Abreu, L. De, Schönfeld, J., Tzedakis, P.C., 2001. Combined marine proxy and pollen analyses reveal rapid Iberian vegetation response to North Atlantic millennial-scale climate oscillations. Quaternary Research 56, 128–132. https://doi.org/10.1006/qres.2001.2218

Rozanski, K., Araguas-araguas, L., Gonfiantini, R., 1993. Isotopic Patterns in Modern Global Precipitation, in: Swart, P.K., Lohman, K.C., McKenzie, J., Savin, S. (Eds.), Climate Change in Continental Isotopic Records. Washington, pp. 1–36. https://doi.org/10.1029/GM078p0001Snoeck, C., Pellegrini, M., 2015. Comparing bioapatite carbonate pre-treatments for isotopic measurements: Part 1—Impact on structure and chemical composition. Chemical Geology 417, 394–403. https://doi.org/10.1016/j.chemgeo.2015.10.004

Sánchez-Goñi, M.F., Eynaud, F., Turon, J.-L., Shackleton, N.J., 1999. High resolution palynological record off the Iberian margin: direct land-sea correlation for the Last Interglacial complex. Earth and Planetary Science Letters 171, 123–137.

Sánchez-Goñi, M.F., Landais, A., Cacho, I., Duprat, J., Rossignol, L., 2009. Contrasting intrainterstadial climatic evolution between high and middle North Atlantic latitudes: A close-up of Greenland Interstadials 8 and 12. Geochemistry, Geophysics, Geosystems 10, 1–16. https://doi.org/10.1029/2008GC002369

Sánchez-Goñi, M.F. 2020. Regional impacts of climate change and its relevance to human evolution. Evolutionary Human Sciences (2020), 2, e55, page 1 of 27

Sepulchre, P., Ramstein, G., Kageyama, M., Vanhaeren, M., Krinner, G., Sánchez-Goñi, M.F., d'Errico, F., 2007. H4 abrupt event and late Neanderthal presence in Iberia. Earth and Planetary Science Letters 258, 283–292. https://doi.org/10.1016/j.epsl.2007.03.041

Skrzypek, G., Wiśniewski, A., Grierson, P.F., 2011. How cold was it for Neanderthals moving to Central Europe during warm phases of the last glaciation? Quaternary Science Reviews 30, 481–487. https://doi.org/10.1016/j.quascirev.2010.12.018

Staubwasser, M., Drăgușin, V., Onac, B.P., Assonov, S., Ersek, V., Hoffmann, D.L., Veres, D., 2018. Impact of climate change on the transition of Neanderthals to modern humans in Europe. Proceedings of the National Academy of Sciences 115, 9116–9121. https://doi.org/10.1073/pnas.1808647115

Tejada-Lara, J. V., MacFadden, B.J., Bermudez, L., Rojas, G., Salas-Gismondi, R., Flynn, J.J., 2018. Body mass predicts isotope enrichment in herbivorous mammals. Proceedings of the Royal Society B: Biological Sciences 285, 20181020. https://doi.org/10.1098/rspb.2018.1020

Tütken, T., Furrer, H., Vennemann, T.W., 2007. Stable isotope compositions of mammoth teeth from Niederweningen, Switzerland: Implications for the Late Pleistocene climate, environment, and diet. Quaternary International 164–165, 139–150. https://doi.org/10.1016/j.quaint.2006.09.004

Vidal-Cordasco, M., Ocio, D., Hickler, T., Marín-Arroyo, A.B., 2022. Ecosystem productivity affected the spatiotemporal disappearance of Neanderthals in Iberia. Nature Ecology & Evolution 6, 1644–1657. https://doi.org/10.1038/s41559-022-01861-5

Vidal-Cordasco, M., Terlato, G., Ocio, D., Marín-Arroyo, A.B. 2023. Neanderthal coexistence with Homo sapiens in Europe was affected by herbivore carrying capacity. Sci. Adv.9,eadi4099(2023).DOI:10.1126/sciadv.adi4099

Other comments:

The name of one of the coauthors should be read as follows: Montserrat Sanz. Her edited affiliation is 4 Grup de Recerca del Quaternari (GRQ-SERP), Department of History and Archaeology, Universitat de Barcelona, C/Montalegre 6-8, 08001 Barcelona, Spain.

---

## Author Response (AR2)

Dear Editor,

First, I would like to explain the delay in our reply. Due to severe medical issues in my family, as a senior author and PI of the project where this research was carried out, I could not work as regularly, and this situation prevented me from an earlier answer on this work. We hope you and the reviewers can understand this exceptional situation.

Kind regards

Ana B. Marín-Arroyo

Dear reviewers,

As authors, we deeply value your perspective and the time you have invested in improving the quality of our contribution to the Biogeosciences journal. We sincerely apologize if any misunderstandings led you to believe we did not appreciate your suggestions. Upon thoroughly reviewing the manuscript, we have diligently followed the reviewer's instructions. In instances where we have not, we have provided an adequate justification. As the reviewer suggested, we noted inconsistencies in our justifications and have now addressed them accordingly. We regret our mistake, and we hope everything is now satisfactory.

Our detailed reply involves a coloured file (made in Excel) where we have 1) enumerated all changes made in response to the reviewers' suggestions, 2) summarized our reasons for some suggestions that were not fully implemented, and 3) described some partially implemented changes and their subsequent enhancements. Moreover, we have specified changes not initially addressed in the reviewer responses but included in the latest draft submitted. The coloured document will facilitate this new reviewer process, where answers are provided to each individual suggestion provided by reviewer 2. In summary, from the 35 individualized suggestions from reviewer 2, we have accepted and justified 30 of them; 2 were not implemented, and 3 were partly implemented and justified to the reviewer and within the paper.

Regarding the comments provided by the editor to reviewer 2 comments, we precise the following issues:

- Title (R2-34): We acknowledge that we were unclear in our decision regarding the title, with changes from the online response to reviewers. Initially, we agreed to change the title, but upon further review, we realised that the suggested title was inappropriate in English. We believe the current proposal aligns with the reviewer's advice and provides a closer idea to the original.

- Chronologies adjustment (R2-3): The chronologies were subsequently adjusted by discussion among authors through the reviewing process. We agree that we didn't justify this change adequately in our previous response to reviewers. We have reviewed the chronological methods, which are explained in a specific section in Methods (3.1. Dating methods). We hope this explanation is more precise now.

- Northeastern Iberia (R2-21): Regarding the use of "northeastern Iberian Peninsula" instead of "Mediterranean," we found relevant this suggestion, and it was implemented throughout the document, except in cases where we referred to the Mediterranean area in general and not specifically to the Canyars site. We acknowledge that the reviewer was correct.

Regarding the 30 accepted changes from reviewer 2, we identified seven cases in which changes were already implemented but are now being improved in the draft submitted today (R2-1, R2-

3, R2-4, R2-10, R2-21, R2-23, R2-31). In our view, these are minor changes, but we believe they now better fit the reviewer's expectations. Expect the site chronologies (R2-3) explained in detail above. Please refer to the attached document for further details.

Only two suggestions were not accepted: the new title proposal (previously explained; R2-34) and a suggested figure for climatic estimation evolutions (R2-33). We justified the second case in our previous response to the editor. In short, we chose not to include the figure because the estimation of paleotemperatures was approached tentatively, and we preferred not to focus on this discussion in this paper.

Afterwards, there are three suggestions from reviewer 2 that were only partly implemented (R2-8, R2-9, R2-11). All three are related to temperature or precipitation estimations, probably the most complex part of this manuscript. Our primary focus for this paper was not to delve deeply into these aspects, as explained. These decisions were extensively explained to the reviewer and justified considering the reviewer's argumentation within subsection 3.4. Specifically, it was suggested to introduce some corrections regarding temporal isotopic composition and age-specific correlations for d13C (R2-8) and d18O (R2-9), considering fluctuations experienced in these elements throughout the Pleistocene. In both cases, we justified our decision not to apply age-specific correlations based on the uncertainty of the dates, which was also pointed out by the reviewer. Nonetheless, we applied a general correction for both d18O and d13C. Furthermore, the reviewer suggested correlations for temperature estimations, differentiating between the Atlantic and Mediterranean regions and between cold and warm seasons in R2-11. We chose to maintain a wide-geographic correlation considering unknown past atmospheric circulation patterns and the limited data derived from IAEA stations. However, as suggested, we decided to include different equations for summer, winter, and mean annual temperatures, and we opted to apply the linear regression models proposed by Pederzani et al. (2021). This last aspect was changed from the initial online response and modified after the reviews were implemented in the text, as we noticed that this change substantially improved the quality of the data provided.

Finally, we detected an error in climatic estimations when implementing reviewer suggestions related to error calculations, which led us to explore alternative solutions and necessitated a significant investment of time. As explained in our last draft, responding to the editor: "During the calculation of errors, it was identified that the correlations utilized for the conversion from d18Ophosp to d18Omw do not correspond to the most updated version. The equations now chosen are the same as those employed in the Axlor site study (Pederzani et al., 2023), which includes a larger number of specimens and is more comprehensive. This, however, implies the modification of Figures 4 and 5. Numbers have also been updated in Tables 3 and 4, in the text and the Supplementary Information (SI). No significant implications have been detected, and the general interpretation aligns with the previous findings.".

We believe that these explanations will help the editor and reviewer appreciate our time carefully implementing their suggestions to improve the quality of our manuscript.

Kind regards,

The authors

| REVIEW CODE | IMPLEMENTED? | Reviewer comment complete | Summary of reviewer comment | How was implemented? | Examples | Changes from online reviews? | How is implemented in the new draft (05/2024)? |
|---|---|---|---|---|---|---|---|
| R2-1 | yes, and now improved | Authors suggested that this type of paleoenvironmental studies is key to understand past climate and human interactions. See for example abstract lines 18-19 or the introduction. Authors must keep in mind that the palaeoenvironmental reconstruction that they have performed in these archaeological sites are "discontinuous points" in the paleoclimatic record of the Iberian Peninsula (see for example the chronologies in Fig.2). | Highlight that the palaeoenvironmental reconstruction provided represent discontinuos points in the paleoclimatic record | In the introduction, we reflected on this idea and insisted that our primary focus is climate reconstruction linked to human presence at the sites. (e.g. "These analyses provide high-resolution snapshots of ecological information from animals accumulated during human occupations at the caves.") | In lines 88-90 | | The implementation of this perspective has been improved through the modification of some sentences in the introduction. Even if some sentences were already corrected to reflect this idea, we have included this perspective with new changes along the introduction (lines 32-34, 56-57, 77, 105). |
| R2-2 | Yes | (...) the most accurate climatic records for the studied periods are marine, lacustrine and speleothem records, where one can observe the "objective" fluctuations of past climate. Authors should acknowledge this point in the manuscript as well as compare/discuss with these records (there are many for the IP, for example Martrat et al., 2004; 2007; Pérez Mejías et al., 2019; Moreno et al., 2012; González-Sampériz et al., 2020; Camuera et al., 2019; 2022, among others). Some of them also discuss about vegetation changes in NE Iberia, close to the study areas, which would help authors contextualize and discuss their interpretations about the animal diets. In fig. 2 authors included the d18O record of Greenland, but they did not discuss their data according to this record. As I mentioned before, they should compare and discuss the Iberian records instead, since there can be some temporal offsets between some of the events of the NGRIP curve and the Iberian/mediterranean records. Taking all this into account, objective 3 (line 96) is not totally achieved. Similarly, the speleothem records from the north of Iberia would help constrain temperature/precipitation patterns; there are many of these records in the Vasco-Cantabrian area. Other records from the Iberian Peninsula, the Mediterranean coast or the Iberian margin would show the general climatic patters for this period. Authors should discuss their data (agreement or disagreement) according to these continuous records in order to have a big picture of the paleoclimate and paleoenvironments in the studied period. | Discuss out data (agreement/disagreement) with other local-regional climatic records (some references provided) and also with continous records (marine, NGRIP) in terms of vegetation/animal diet and climate/environment changes. | All these references have been incorporated, mentioned in the introduction and discussed in the discussion section. Indeed, this suggestion has notably improved the Discussion | Section 1 (lines 81-82) and Section 5.4 | | |
| R2-3 | yes, and now improved | Chronology: This a very important part of the study and should be presented in a subsection in section 2 or in the methodology (section 3). Please, explain deeply the absolute chronology of the levels where the ungulated remains were collected.(...) Please, specify the dating method: ESR, OSL or 14C in Fig 2. What are the grey bands in Fig. 2, stadials? Please explain. (...) What is the meaning of the green colour in one date at around 40 ka in Fig 2. (Canyars)? What is the meaning of the dates (dots and bars) in Fig 2: Do they represent a single dating event, or a sum of distribution at various dates?, please explain this in the caption and in the main text. If it is the sum of distribution, did you use any statistic approach to obtain it (such as the ones that can be obtained from OxCal software?) Do they represent the ages of the whole archeological sites or of the levels where the ungulated teeth were taken? This is very important to specify in the main text since the age of the remains could vary. Therefore, the caption of Fig. 2 should explain these details and there might be a (sub)section in the main text explaining the chronology of the levels where the remains were taken. This is crucial to validate the discussion of the manuscript | Explain the absolute chronology methods and calibration methods (radiocarbon curve intCal2020) in a subsection within the manuscript and include details in figure 2: dating methods, green colour and grey bands meaning, dots and bars meaning, statistical approch and software. Review incosistences between explanation provide to reviewer and manuscript implementation in Fig. 2, section 2 and Appendix B. | We have specified the methods of calibration and date origin for each level in section "2. Archaeological sites and sampled material." Appendix B includes all the original ESR, OSL, and 14C dates for each level and 14C calibration, as well as an explanation of average estimation by levels. All formal changes indicated in Figure 2 are included. (*) This review changed from the initial online response. | Appendix B (B1_Dates) + New Appendix C | | This issue has been largely improved in this last draft from the previous one because we detected some inconsistent results derived from the calibration method during the reviewing process (*). This includes new changes in the Figure 2 caption, Section 2, and Appendix B. A new section 3.1 and a new Appendix C, providing details on calibration dates, have also been created. |
| R2-4 | yes, and now improved | Takin into account the confidence intervals of the ages (I suppose 2 sigma for 14C)->Line 98-99: "The chronological resolution in the study areas for this period allows us to correlate regional paleoenvironmental changes with global records" this would be only true for the two sites younger than 30 ka, since the dates of the other sites might overlap stadial and interstadials (and their probability distributions are very large). (...) In any case the chronology of the samples | The chronological resolution do not allow to correlate our levels to global climatic changes | We have modified this sentence: Considering the error chronological margins and the limitation into a straightforward correlation with a single climatic stadial (GS) or interstadial (GI) we have modified the sentence "The chronological resolution in the study areas for this period allows us to correlate regional paleoenvironmental changes with global records". | Lines 111-113 | | In this last draft, we finally decided to remove this sentence |
| R2-5 | Yes | Authors mentioned that they did not carry out any pre-treatment to remove secondary carbonates, but did authors check the potential presence of secondary carbonates or the preservation of carbonates? This is very important since all the results are based on these values (there is no data of d18O in phosphates). The physical cleaning that was carried out would not remove all the potential secondary carbonates. Secondary carbonates are very common in archaeological contexts such as karstic caves (like the ones studied here), and would modify the isotopic composition of carbonates if they are not eliminated. This must be double checked before stating the sample preparation. Authors did not mention methods to double checked that the isotopic signal was the pristine one. They only mentioned in line 503-504: "The carbonate content in our samples, ranging from 3.9% to 8.9%, is similar to the proportion found in modern tooth enamel, suggesting no immediate indication of diagenetic alteration". However, there is no explanation about the methodology used to calculate this percentage of carbonate. Authors should explain this in the methodology section and add the % of carbonate in a table (e.g. Table 2). | Explain if some methods are use to double-checked absence of secondary carbonates, explain the method of carbonate content calculation and include it in Table 2 | We answered the reviewer with all the methods employed (calcium carbonate content, d18O in phosphates from Axlor) and explained that any method or pretreatment can totally assure this issue. We include carbonate content in Table 2 and the explanation in Appendix B. | Table 2, Appendix B (B1_Samples-Raw), Section 3.3. and lines 603-604 | | |
| R2-6 | Yes | Regarding the potential treatment to remove the organic matter, authors said: Lines 145-151: "For this reason, in this work, most of the samples were not pretreated, except for the equid samples from Labeko Koba and Aitzbitarte III, and the cervids and equids from El Otero that were sampled and pretreated in the context of the initial project. Pretreatment followed was established by Balasse et al. (2002), where around 7 mg of powdered enamel was prepared and pretreated with 3% of sodium hypochlorite (NaClO) at room temperature for 24 h (0.1 ml/mg sample), and thoroughly rinsed with deionised water, before a reaction with 0.1M acetic acid for 4 h (0.1 ml/mg sample) (equivalent protocol in Jones et al., 2019)." And afterwards in lines 453-456: "In the case of equid samples from the Vasco-Cantabrian region, it should be considered that they have been pretreated with a combination of NaClO and acetic acid, which could potentially affect the isotopic values. Samples after organic removal pretreatment can potentially show either higher or lower δ13C values and higher δ18O values based on previous experiments (Pellegrini and Snoeck, 2016; Snoeck and Pellegrini, 2015)". So, my doubt is: why did authors treat whole batches of samples to evaluate these "side effects" instead of applying both protocols (with pretreatment and without pretreatment) to aliquots of the same samples? Although they finally ended up that "the influence of the pre-treatment on our samples is deemed to be limited.", this was a risk, and now they cannot be 100% sure about this potential influence. Was there any reason to measure the samples where the organic matter was removed in a different IRMS. If yes, please, explain. | Justification on pretrated some samples and non-pretreated others | We have explained that the cause is related to different research phases of the project within the EvoAdapta group. | Lines 182-183 | | |
| R2-7 | Yes | Line 182-196 and throughout the calculations and the manuscript: Authors referred to E* as the fractionation factor. The symbol E is traditionally the enrichment factor, not the fractionation factor (alpha). E=(alpha-1)x1000‰. So, what factor have authors applied eventually: fractionation or enrichment factors? Are these factors mixed in the text and in the calculations? This has to be clear, and if the factors are wrongly applied (enrichment instead of fractionation factor), correct the calculations. Although both factor are related (E= (alpha-1)x1000‰) the obtained results would differ, and thus, the derived potential interpretations. | Review if is the fractionation factor or enrichment factor was applied. | We detected it was a terminological confusion: fractionation was used instead of enrichment. This was reviewed in the paper | Different parts of the paper | | |
| R2-8 | Partly implemented | Lecuyer et al. (2021) performed the calculations to correct the effect of atmospheric CO2 (difference of 1‰ and CO2 concentration) for the LGM; so, these specific CO2 corrections can only be applied for the LGM, but in the present manuscript there are no samples for the LGM (23-19 ka). In addition, the correction for the isotopic composition of atmospheric CO2 should be done specifically for each age of the studied samples, instead of using a general average -7‰: as authors mentioned, a variation of a ca. 1‰ would imply a change in 150mm of precipitation. Check for example the isotopic composition of atmospheric CO2 reconstructed from ice records for the Late Pleistocene that ranges from -7 to -6.5 ‰ (Eggleston et al., 2016: Paleoceanography, among others). So, please, apply age-specific CO2 corrections. I mean, when you are quantifying climatic variables, you should reduce the potential error sources. These errors increase by applying a general unspecific correction for all the data (the same affirmation could be applied to all corrections of the isotopic data in the manuscript). | Considering variation on isotopic composition of CO2 during the Pleistocene apply age specific CO2 corrections for d13C and not a generic correction to avoid errors | We agree that, ideally, corrections should be age-specific. However, considering the chronological uncertainties of some of the older levels in this work, we believe this could complicate the final interpretation. We, therefore, decided not to implement it, but we explained these CO2 variations; we mentioned the identity of age-specific corrections (as well as provided references) and the uncertainties related to MAP estimations. | Section 3.4 (lines 251-253, 260-263) | | |

| REVIEW CODE | IMPLEMENTED? | Reviewer comment complete | Summary of reviewer comment | How was implemented? | Examples | Changes from online reviews? | How is implemented in the new draft (05/2024)? |
|---|---|---|---|---|---|---|---|
| R2-9 | Partly implemented | Authors proposed the above-mentioned (oversimplified) correction for the change in the isotopic composition atmospheric CO2, but they did not apply any correction related to the change of the isotopic composition of the sea-water during the Late Pleistocene, which is the main moisture source for rainfall (I would not mention the moisture and precipitation due to inland evaporation/recycling in the Iberian Peninsula, even during cold periods (Krklec and Domínguez-Villar, 2014), in order to simplify this interpretation). The global isotopic composition of the rain during colder/warmer periods (glacial/interglaciar, stadial/interstadial) differs, not only due to the isotopic fractionation caused by temperatures, but also due to the accumulation/release of the lighter water isotopes in the ice sheets/glaciers during cold/warmer periods (Dansgaard, 1964), among others factors affecting the global isotopic composition of sea waters. Therefore, in order to obtain a reliable isotopic data related to precipitation, the obtained dO18 values has to be corrected to remove this effect. See for example, Niedermeyer et al. (2010) or García-Alix et al. (2021) approach to correct past hydrogen isotopes from vegetation, or even Fernández-García (2020) for fossil mammals in the studied period of this paper. | It is proposed to apply a d18Omw correction considering d18O oscillations in sea-water. Preferentially, an age-specific correction, considering d18O glacial-interglacial fluctuations. | Considering the chronological uncertainties, we find it more consistent to apply a general correction in d18Omw (for the MIS3 period). It is explained in section 3.5. This supposed changes in temperatures estimations along the text and in some tables and figures. | Table 3, table 4, figure 4, figure 5, section 3.5 | | |
| R2-10 | yes, and now improved | Where did these obligate drinkers drink (water source)? Directly from the rain? Ponds? Lakes? Unless they directly drink precipitation waters (oversimplification), this would imply more isotopic fractionation and would also mask the temperature signal. This is even more important in the studied glacial period, and especially in the stadials? Apart from the potential enhanced rain evaporation due to low atmospheric moisture in glacial times (Dansgaard 1964), an increasing evaporation in lakes, ponds (and even in vegetation->enhanced evapotranspiration) during dry periods have been demonstrated by different isotopic studies in carbonates from freshwater gastropods, bivalves, or ostracods and even from leaf wax isotopes of freshwater and terrestrial plants. Please clarify this issue, and explain this constrain in the methodology since it would affect the reconstructed temperatures. | Water sources of the animals studied and implications in temperatures estimation. Justify this in the text | In response to the reviewer, we explain possible water sources and implications for d18O interpretation. In short, evaporation and aridity do not seem to impact our samples, and for some individuals, we justify a seasonal pattern reflecting seasonal rainfall. We included, however, explanations of the impact of the non-temperature effect in the manuscript. | Section 5.2 | | We have reconsidered this response and added some explanations in the current subsection 3.5 (lines 284-289). |
| R2-11 | Partly implemented | Precipitation source (North Atlantic Oscillation modes and Mediterranean dynamics). In the Iberian Peninsula, the isotopic composition of precipitation is highly affected by the moisture source in the present - and in the past - (Araguás y Díaz Teijeiro, 2005, Celle-Jeanton et al., 2001; Domínguez-Villar et al., 2013; Krklec and Domínguez-Villar, 2014; Moreno et al., 2010; 2012; 2014; 2021, Toney et al., 2020; García-Alix et al., 2021; Schirrmacher et al., 2020, among others). Therefore, temperature-isotope fractionation equations would not work that well to reconstruct past temperatures. When proper analysis of the isotopic signal of precipitation are performed in N Iberia, there are sampling stations (precipitation, temperatures, isotopes, and moisture sources) where the isotopes from precipitation do not correlate well with temperatures (Moreno et al., 2021). This thoroughly study of the isotopic composition of precipitation in northern Spain ended up with "although important, air temperature only explains part of the observed δ18Opvariability and is therefore not the only control." This issue is even more important in coastal areas, as the ones studied in this paper, and even more in the Mediterranean coast (at present there is some influence of amount effect in the Mediterranean areas of Catalonia; Moreno et al., 2021). Therefore, in the best oversimplified case-scenario (not considering previous comments 5 and 6, and admitting large errors due to these potential source and amount effects) if we would want to calculate temperatures from the isotopic values of precipitation we would need an equation for the Vasco-Cantabrian region, and another one for NE-Mediterranean Iberia (different precipitation pattern and forcing). And even more, since atmospheric patterns, and therefore, moisture sources, are not the same during the warm and cold seasons (see the above-mentioned studies), specific equations for cold and warm seasons should be applied to reconstruct "summer" and "winter" temperatures. | In temperatures estimations based on d18O, consider moisture sources and other effects different from temperature-effect is dominant. Develop specific equations for Atlantic and Mediterranean and for cold and warm seasons. | Considering this unknown past atmospheric circulation patterns and the limited data derived from IAEA stations, we preferred to apply a wide-geographic correlation. In the final reviews, we decided to include different equations for summer, winter and MAT finally. Considering the reviewers' argumentation in section 3.5, these decisions are largely explained and justified. (*) This review changed from the initial online response. | Section 3.5 | (*) During the online review, we did not follow the advice to adjust the correlation to cold and warm seasons, but during the review's implementation in the text, we noticed that this change significantly improved the quality of the data provided. | |
| R2-12 | Yes | "MAT was calculated from the d18O mean value between summer and winter in each tooth before modeling to reduce associated error"; However in caption Table 4: "For some profiles with an unclear seasonal shape, MATs were deduced from the original average of teeth without a seasonal profile". So, what is the correct methodology?, In any case, according to the methodology section, MAT was calculated before modeling to reduce associated error, but summer and winter temperatures after the inverse modelling?. This reasoning is not clear to me: Is there no associated error in the transformation for summer and winter temperatures? This is why there are some odd values, for example, sample AXL60 MAT 12ºC, ST 20.4ºC and WT 10.8ºC (only 1.2ºC colder than the MAT). | Explain how MAT is estimated (summer-winter or original teeth average) and why summer and winter after modelling. | MAT was estimated from summer-winter unmodelled data to reduce errors, whereas summer and winter can only be deduced after modelling because seasonal amplitude is otherwise attenuated. To maximize data, MAT from unmodelled teeth profiles, MAT was deduced from teeth d18O average, but it is less reliable. We detected that these explanations were not clearly explained, and we improved them in section 3.5. | Section 3.6 | | |
| R2-13 | Yes | The reconstructed meteoric waters are different depending on the species, even in the same level, and therefore, reconstructed temperatures also differ. I'm aware of the different ecological behaviors of the different species, but the MAT should be, at least close. There are also some discrepancies in specimens of the same species in the same archaeological levels. | Reasons on differences in d18Omw between species in the same level | We believe interspecific variability is not higher than intraspecific variability, and we argue multiple reasons that can explain this (ecological behaviour, physiological factors, levels as palimpsests) both in the reviewer response and within the manuscript in section 5.2. | Section 5.2 (lines 637-641) | | |
| R2-14 | Yes (justified) | The general comparison between the isotopic composition of the faunas of the archaeological sites of both areas is not objective since in the Vasco-Cantabrian area there are remains from 80 ka to 18 ka approx. (14 sections from 5 archaeological sites), but in the Catalonian area there is only one site at around 40 ka (it could have been coeval to the HS4, a period especially cold and dry). Thus, the general comparisons that appeared in some parts of the manuscript between two areas (not the ones specific at ca. 40 ka) are not balanced. | Comparision between NW and NE samples is not balanced | We find this site interesting enough by the period it represents to be included, even if we agree with the reviewer | No changes required | | |
| R2-15 | Yes | What is the error associated with the different equations Eq. 1-9? Authors have to take into account the different errors (not only the standard deviations) that are being accumulated in each equation and also plot and mention them in the tables and in the text. For example, a MAT of 15ºC +/- 0.5 ºC would be accurate, but if the error was +/- 4ºC the interpretation would be more open. This is even more important in this work since the oxygen isotopes were measured in carbonates, not in phosphates, and two more equations were used to converter the d18O data of carbonates to d18O of phosphates. Authors said: lines 277-278: "In this work estimations, the associated error from converting δ18Ophos to MAT is enlarged by the uncertainty derived from the transformation of δ18Ocarb (VPDB) to δ18Ophos (VSMOW)". Therefore, all the errors should be calculated (for both O and C), included in the plots, tables and when describing the results. | Provide error acummulated associated to equations | Associated errors were calculated following Pryor et al. (2014). Considering the advice provided by Pederzani et al. (2021), it has been argued that the uncertain repercussions are associated with each conversion step. This is mentioned in section 3.4. Errors are included in Appendix B (B6-Temperature-estimations) and mentioned in table 4 caption. Moreover, spreadsheets were provided in SI. | Section 3.5, Appendix B (B6-Temperature-estimations), Table 4 caption. Spreadsheets is SI | | |
| R2-16 | Yes | Inverse Modelling. I think that the correct reference for the inverse modeling (main text and appendix D) is Passey et al. (2005) but this one "Passey, B.H., Cerling, T.E., Schuster, G.T., Robinson, T.F., Roeder, B.L., Krueger, S.K., 2005. Inverse methods for estimating primary input signals from time-averaged isotope profiles. Geochim. Cosmochim. Acta 69, 4101–4116." Reviewing this paper, I noted that the reconstructed profiles showed mostly the same trends/changes as the original isotopic data. However, in the case of the reviewed manuscript sometimes these reconstructed profiles exhibited opposite patterns/trends from the original isotopic values. I'm not familiar with these kinds of transformations, but: is this common? This is an extra transformation for the data that would be used to calculate absolute climatic parameters, and thus, extra potential error source. Could authors double check these calculations?. In addition, in the original paper from Pasey et al. (2005), this inverse modeling was also applied to the carbon isotopes. Why authors did not apply this correction also to the carbon isotopes of the sequential sampling? | Doubts on modelling: 1) why some modelled teeth show opposite pattern from original; 2) why is not apply to d13C profiles. Error in the reference. | We provide different reasons that can explain the editor's feeling about "opposite patterns" after modelling, derived from a lag in the x-axis respecting the original signal and non-sinusoidal profiles. The reference was corrected. The absence of seasonal change does not allow model application. Details in Section 3.6 and Appendix E. | Section 3.6 (lines 334-338) Appendix E (lines 1567-1569, 1587-1590) | | |

| REVIEW CODE | IMPLEMENTED? | Reviewer comment complete | Summary of reviewer comment | How was implemented? | Examples | Changes from online reviews? | How is implemented in the new draft (05/2024)? |
|---|---|---|---|---|---|---|---|
| R2-17 | Yes | The first paragraph in the introduction and the first lines in the abstract deal with the importance of these kinds of studies to understand the human evolution in this region, but eventually, this is not discussed in the manuscript according to the obtained data. | The introduction explains the importance of the article for human evolution but this is not discussed | We mean the changes in human dynamics related to environmental conditions. We have clarified this in section 1 | Section 1 | | |
| R2-18 | Yes | First sentences of the introduction (lines 38-43): please add references. There are interesting papers dealing with this issue in the Iberian Peninsula and in Europe: Neanderthal-AMH-climate change. | Include references on Neanderthal-AMH-cimate changes | References were added in section 1 | Section 1 (lines 44-45, 52-53) | | |
| R2-19 | Yes | Lines 49-50. It is ok, but authors are not including some information (and significant climate-related references) in the area dealing with continuous paleoclimatic records, and they only focused on the data obtained from archeological sites (whose paleoenvironmental record is not that continuous); so, line 75 is not summarizing the multiproxy studies in this area. | Introduce information climatic related references | References were added in section 1 (link with comment R2-2) | Section 1 (lines 79-85) | | |
| R2-20 | Yes | Lines 80-87: add references | Add references | References were added in section 1 | Section 1 (lines 90-91, 93-94) | | |
| R2-21 | yes, and now improved | Regarding the fossil sites that authors call mediterranean". Since the rest of the fossil sites are in "northeastern Iberia", the term "Mediterranean area" is very open and do not specifically identify the studied site, I would say NE Iberia? | Change Mediterreanean by northeastern Iberia | We decided it to change this all along the text. | Appendix A. Different parts of the paper | | We detected an error that sometimes prevented completion during the reviewer process. Now, it is corrected everywhere, including in Appendix A. |
| R2-22 | Yes | There are some issues with the chronology/dates. For example, line 21 abstract: 80 to 15,000 cal BP. Do you mean 80 ka or ky, right? Taking into account the study period, and the accuracy of the dates, I would not use "yr", I would use ky or ka. In addition, authors should round the dates in the text to the nearest hundreds (eg, line 601: 41,136 to 38,570 cal yr BP: 41.1 ka to 38.6 ka). This accuracy does not make sense in the studied period due to the uncertainty of the measurements. | Correct dates format to ka/ky | All dates throughout the paper were revised and expressed in as ka BP or ka cal BP. | Different parts of the paper | | |
| R2-23 | yes, and now improved | Results: please, add the ages (in ka and the different technocomplexes) to the subsection headings of the different archaeological sites, otherwise one has to check Fig. 2 for each site. | Include dates and technocomplexes from archaeological sites in subsection headings | We included dates in the mention headings (*) This review changed from the initial online response. | Headings from subsections 4.1 to 4.6 | (*) Final dates are different from those indicated initially in online reviews due to the already explained change in calibration criteria (More details on R2-3) | In this last review, we noticed that it was suggested also to include technocomplexes. Now, it is implemented. |
| R2-24 | Yes (justified) | I understand the structure of the result description, but the mixed description of the isotopes from different levels of the same archaeological sites, which sometimes have 10 ka of difference between them, is rare to me. | In results, different levels are explain together | We believe that this is the most efficient way, considering the specific characteristics of each individual site and baseline isotopic values. | No changes required | | |
| R2-25 | Yes | Line 325: MATAs=1-8/-2.1°C? do you mean 1.8? | Error on MAT (line 325) | Corrected and all temperatures reviewed along the paper | Section 4 | (*) All temperature estimations were greatly modified and derived from new correlation adjustments for temperature estimations | |
| R2-26 | Yes | Eq 10: P value <? | Error on p-value | It was an error, but we finally Fernandez-Garcia et al. (2019) was removed from the text. (*) This review changed from the initial online response. | Subsection 3.5 | (*) We propose to correct this in the initial answer, but the p-value was associated with a correlation no longer used in the text (more details in R2-11). | |
| R2-27 | Yes | All figures with the data plotted according to the chronology. El Castillo 21A appears after Axlor III; but according to the chronology this site is in previous (Fig. 2). The same would happen between some levels of Labeko Kova and Canyars-I. Is this correct? In this case, arrange the sites with the real chronological order even though they belong to different archaeological sections. | Arrange the sites diachronically | We interpreted reviewer referred to Figure 5. We have rearranged the levels in chronological order. (*) This review changed from the initial online response. | Figure 5 | (*) Initially, we considered was more easy to undestand arranging the figure by sites, but we reconsidered this decision during reviews modifications in the manuscript and finally on organizing levels diachronically. | |
| R2-28 | Yes | When speaking about range of values (temperatures, isotopes, precipitation), sometimes the lowest values are mentioned before and other times the other way around. Be consistent and cite the lowest values first (Eg, line 404 MATAs). | Put the lowest value first in data | Corrected and review along the paper | Sections 4 and 5 | | |
| R2-29 | Yes | Table 3: I suppose that the different group of data (rows) are related to the three groups of specimens. Right? Please, specify the taxon groups in the tables. | Taxa missing in table 3 | Titles of the table were corrrected | Table 3 | | |
| R2-30 | Yes | I do not understand this sentence. Line 510-511: "Based on these arguments, it is suggested that the non-sinusoidal δ18O signal observed in some individuals is likely attributed to the preservation of the original isotopic signature from water input." This sentence would suggest a bad preservation of the original water composition when a sinusoidal pattern is present?. However, authors explained afterwards some reasons related to the ecology of the specimens. Please, clarify this. | You suggested that sinosoidal profiles indicate bad preservation or ethological factors? | These non-sinusoidal profiles do not indicate poor preservation and may be linked to the individuals' ethological factors (more details on R2-10 and R2-13). We clarified this in the text. | Subsection 5.2 and lines 607-620 | | |
| R2-31 | yes, and now improved | Section 5.4. Regarding the d18O of meteoric waters. Authors compare (as a whole) their reconstructed d18Omw with the current values in the area and they ended up with similar ranges. Keep in mind that the reconstructed values correspond to the last 80 ka under glacial conditions (different temperatures, precipitation amount, and in some cases, moisture source) | When comparing d18Omw with current values consider all factors that can be different in the past | We agree with this consideration and we were less determinant in our explanation. | Subsection 5.4 (first pragraph) | | We have included an extra explanation on this point |
| R2-32 | Yes | Sometimes the different d13C and d18O have subscripts with the meaning of the isotopic values (eg, δ13Ccarb, δ18Ocarb), but another times this is not indicated. This is confusing. Please, add always the subscripts explaining the meaning of the isotopic values. | Add always subscripts to d13C and d18O | We corrected the text to ensure that "d13C" and "d18O" always include subscripts, except when referred to general explanations. | Different parts of the paper | | |
| R2-33 | No implemented | I miss a figure summarizing the chronological evolution of reconstructed temperatures and precipitation (with the associated accumulated errors) and comparing them with other continuous records of precipitation or temperatures or vegetation from the Iberian Peninsula, Mediterranean coast or Iberian margin. | Include a figure summarizing temperatures and precipitations estimations | We chose not to include it because the estimation of paleotemperatures was approached tentatively, and we preferred not to focus on this discussion in this paper | None | | |
| R2-34 | No implemented | Title: ecological evolution of what? This is very general: I would say ecological evolution of ungulate fauna…. | Change the title | We agreed with the title initially in the online reviews, but after discussing it with coauthors we considered it not the most appropriated | None | | We provide a new proposal in this reviewed version |
| R2-35 | Yes | Please, use always the symbol delta instead of d, there are some "ds" throughout the manuscript. | Use delta symbol (not "d") | Corrected | Different parts of the paper | | |

---

## Author Response (AR3)

Dear Editor,

We are glad that reviewers consider our manuscript worthy of publication in *Biogeosciences* after our changes and detailed replies. All of the minor changes have been addressed, and we have provided our reply below.

Kind regards,

Ana B. Marín-Arroyo

Reviewers

I think that most of the major issues of the previous versions have been solved. I only have a couple of suggestions/comments:

Title: I think that "evolutionary ecology" is not the best choice for this paper, since the field of evolutionary ecology deals with topics relating ecology and evolutionary patterns, and there is no "evolutionary discussion" of the results in this manuscript. So, why not "Paleoecology of ungulates…."? This would be a good summary of the results.

Agree. We modified the title to "Palaeoecology of ungulates in northern Iberia during the Late Pleistocene through isotopic analysis of teeth"

Fig. 2, please, specify which dates are 14C, ESR and OSL in figure 2.

This data has been included in the figure caption.

New caption> Figure 2. Representation of the duration each archaeological level (dots represent the median values, bars represent 95% confidence intervals for 14C dates and 68% for ESR and OSL dates) related to techno-complexes in both northwestern (in black) and northeastern Iberia (in green) and the δ18O record from the NGRIP (North Greenland Ice Core Project members, 2004; Rasmussen et al., 2014). Grey bands indicate Greenland Stadials (GS). Dates from EL Castillo (C14 UF, ESR), El Otero (C14 UF), Axlor (C14 UF, OSL), Labeko Koba (C14 UF), Aitzbitarte III-interior (C14 AMS) and Canyars (C14 UF, ABA, ABOx-SC) are shown in Appendix B and C.

I think that the potential evaporation and amount effect interferences should be clarified. Line 354-355-> authors have discussed about the negligible effect of plant evapotranspiration. This is fine, but there are also other evaporative effects that have not been properly discussed. For example, the evaporation of rain drops when precipitation is scarce (amount effect), especially in dry periods, like the last glacial cycle (or even in summer precipitation). It should also be mentioned the evaporation that occurred in the water bodies (ponds, lakes) where these animals would drink. These effects would modify the isotopic value of the drinking water, and would also mask the temperature signal. In their response authors said that "evaporation and aridity do not seem to impact our samples, and for some individuals, we justify a seasonal pattern reflecting seasonal rainfall. ". However, this response is very vague. For example, the "amount effect" (Dansgaard 1964) affects precipitation when it is scarce (amount) and specially when the atmosphere is dry (glacial periods, summer periods, etc). It is very important at low latitudes. In mid latitudes such as the Iberian Peninsula, there would be a mixture between amount and temperature effect (specially in dry periods: summer, glacial…). Authors should discuss this potential effect in their samples. See the classic paper from Dansgaard (1964). In the case of water bodies (ponds, lakes, etc.), authors could argue that low temperatures during the last glacial cycle would prevent (or reduce) evaporation.

In section 5.2 (lines 590-656) we provide a detailed discussion about this issue. Regarding the influence of water sources susceptible to evaporation, this is mentioned in lines 651-656. We

concur with the reviewers and consider this to be a potential cause of the non-sinusoidal profiles observed (line 635). We have provided further clarification regarding the "*amount effect*"

Line 894 ATC (and throughout the manuscript)-> Pleases use "HS" for the Heinrich stadials. "HE" is only related to the specific episodes of IRD discharge in the North Atlantic.

This change has been addressed throughout the text.

---

## Author Response (AR4)

Dear Editor,

We are glad to hear that our paper has been accepted after the minor changes we submitted on 22nd July 2024. Now, we submit the final paper accepted, following the journal guidelines as requested.

Kind regards,

Ana B. Marín-Arroyo